# Model-Free Reinforcement Learning with the Decision-Estimation Coefficient

**Dylan J. Foster**
dylanfoster@microsoft.com

**Noah Golowich**
nzg@mit.edu

**Jian Qian**
jianqian@mit.edu

**Alexander Rakhlin**
rakhlin@mit.edu

**Ayush Sekhari**
sekhari@mit.edu

## Abstract

We consider the problem of interactive decision making, encompassing structured bandits and reinforcement learning with general function approximation. Recently, Foster et al. [12] introduced the Decision-Estimation Coefficient, a measure of statistical complexity that lower bounds the optimal regret for interactive decision making, as well as a meta-algorithm, Estimation-to-Decisions, which achieves upper bounds in terms of the same quantity. Estimation-to-Decisions is a *reduction*, which lifts algorithms for (supervised) online estimation into algorithms for decision making. In this paper, we show that by combining Estimation-to-Decisions with a specialized form of *optimistic estimation* introduced by Zhang [31], it is possible to obtain guarantees that improve upon those of Foster et al. [12] by accommodating more lenient notions of estimation error. We use this approach to derive regret bounds for model-free reinforcement learning with value function approximation, and give structural results showing when it can and cannot help more generally.

## 1 Introduction

The theory of interactive decision making—ranging from bandits to reinforcement learning with function approximation—contains a variety of sufficient conditions for sample-efficient learning, [25, 17, 26, 11, 19, 7, 29, 18, 23, 3, 20, 10, 22, 9, 33], but *necessary conditions* have been comparatively unexplored. Recently, however, Foster et al. [12] introduced the Decision-Estimation Coefficient (DEC), a measure of statistical complexity which leads to upper *and* lower bounds on the optimal sample complexity for interactive decision making.

Regret bounds based on the Decision-Estimation Coefficient are achieved by Estimation-to-Decisions (E2D), a meta-algorithm which reduces the problem of interactive decision making to supervised online estimation. While the Decision-Estimation Coefficient leads to tight lower bounds on regret for many problem settings, the upper bounds in Foster et al. [12] can be suboptimal in certain situations due to the need to perform estimation with respect to *Hellinger distance*, a stringent notion of estimation error. When specialized to reinforcement learning, the guarantees for the E2D meta-algorithm in Foster et al. [12] are only tight for model-based settings (where function approximation is employed to model and estimate transition probabilities), and do not lead to meaningful guarantees for model-free settings with value function approximation. In this paper, we explore the prospect of developing tighter regret bounds suitable for model-free settings.

**Contributions.** We show that by combining Estimation-to-Decisions with *optimistic online estimation*, an elegant technique recently introduced by Zhang [31], it is possible to obtain regret bounds that improve upon Foster et al. [12] by accommodating weaker notions of estimation error. Our main contributions are:

37th Conference on Neural Information Processing Systems (NeurIPS 2023).

- We introduce a new *optimistic* variant of the Decision-Estimation Coefficient, and show that a variant of Estimation-to-Decisions that incorporates optimistic estimation achieves regret bounds that scale with this quantity (Section 2). Using this approach, we derive the first regret bounds for Estimation-to-Decisions applied to model-free reinforcement learning with *bilinear classes* [11] (Section 2.3).

- We show that in general, whether or not optimistic estimation leads to improvement depends on the *divergence* with respect to which estimation is performed: For *symmetric* divergences, optimistic estimation offers no improvement, but for *asymmetric* divergences, including those found in reinforcement learning, the improvement can be drastic (Section 3). In addition, we highlight settings in which combining optimistic estimation with Estimation-to-Decisions offers provable improvement over previous approaches that apply the technique with posterior sampling [31].

Perhaps the most important aspect of our work is to elucidate the connection between the DEC framework and optimistic estimation, building foundations for further research into these techniques.

In what follows, we review the Decision-Estimation Coefficient and Estimation-to-Decisions meta-algorithm (Section 1.2), highlighting opportunities for improvement. In Section 2, we present our main results, including our application to model-free reinforcement learning. We close with discussion and structural results, highlighting situations in which optimistic estimation can and cannot help (Section 3).

## 1.1 Problem Setting

We adopt the *Decision Making with Structured Observations* (DMSO) framework of Foster et al. [12], which is a general setting for interactive decision making that encompasses bandit problems (structured, contextual, and so forth) and reinforcement learning with function approximation.

The protocol consists of $T$ rounds. For each round $t = 1, \ldots, T$:

1. The learner selects a *decision* $\pi^t \in \Pi$, where $\Pi$ is the *decision space*.

2. The learner receives a reward $r^t \in \mathcal{R} \subseteq \mathbb{R}$ and observation $o^t \in \mathcal{O}$ sampled via $(r^t, o^t) \sim M^\star(\pi^t)$, where $M^\star : \Pi \to \Delta(\mathcal{R} \times \mathcal{O})$ is the underlying *model*.

Above, $\mathcal{R}$ is the *reward space* and $\mathcal{O}$ is the *observation space*. The model (conditional distribution) $M^\star$ represents the underlying environment, and is unknown to the learner, but the learner is assumed to have access to a *model class* $\mathcal{M} \subset (\Pi \to \Delta(\mathcal{R} \times \mathcal{O}))$ that is flexible enough to capture $M^\star$.

**Assumption 1.1** (Realizability)**.** *The learner has access to a model class $\mathcal{M}$ containing the true model $M^\star$.*

The model class $\mathcal{M}$ represents the learner's prior knowledge about the decision making problem, and allows one to appeal to estimation and function approximation. For structured bandit problems, models correspond to reward distributions, and $\mathcal{M}$ encodes structure in the reward landscape. For reinforcement learning problems, models correspond to Markov decision processes (MDPs), and $\mathcal{M}$ typically encodes structure in value functions or transition probabilities. We refer to Foster et al. [12] for further background.

For a model $M \in \mathcal{M}$, $\mathbb{E}^{M,\pi}[\cdot]$ denotes the expectation under the process $(r, o) \sim M(\pi)$, $f^M(\pi) := \mathbb{E}^{M,\pi}[r]$ denotes the mean reward function, and $\pi_M := \arg\max_{\pi \in \Pi} f^M(\pi)$ denotes the optimal decision. We measure performance in terms of regret, which is given by $\mathbf{Reg}_{\mathsf{DM}} := \sum_{t=1}^T \mathbb{E}_{\pi^t \sim p^t}[f^{M^\star}(\pi_{M^\star}) - f^{M^\star}(\pi^t)]$, where $p^t$ is the learner's randomization distribution for round $t$.

**Additional notation.** For an integer $n \in \mathbb{N}$, we let $[n]$ denote the set $\{1, \ldots, n\}$. For a set $\mathcal{Z}$, we let $\Delta(\mathcal{Z})$ denote the set of all probability distributions over $\mathcal{Z}$. For a model class $\mathcal{M}$, $\mathrm{co}(\mathcal{M})$ denotes the convex hull. We write $f = \widetilde{O}(g)$ to denote that $f = O(g \cdot \max\{1, \mathrm{polylog}(g)\})$, and use $\lesssim$ as shorthand for $a = O(b)$.

## 1.2 Background: Estimation-to-Decisions and Decision-Estimation Coefficient

To motivate our results, this section provides a primer on the Estimation-to-Decisions meta-algorithm and the Decision-Estimation Coefficient. We refer to Foster et al. [12] for further background.

---

**Algorithm 1** Estimation-to-Decisions (E2D) for General Divergences

---

1: **parameters**: Estimation oracle $\mathbf{Alg}_{\mathsf{Est}}$, Exp. parameter $\gamma > 0$, divergence $D^\pi(\cdot \parallel \cdot)$.

2: **for** $t = 1, 2, \cdots, T$ **do**

3:     Compute estimate $\widehat{M}^t = \mathbf{Alg}_{\mathsf{Est}}^t\big(\{(\pi^i, r^i, o^i)\}_{i=1}^{t-1}\big)$.

4:     $p^t \leftarrow \arg\min_{p \in \Delta(\Pi)} \sup_{M \in \mathcal{M}} \mathbb{E}_{\pi \sim p}\big[f^M(\pi_M) - f^M(\pi) - \gamma \cdot D^\pi(\widehat{M}^t \parallel M)\big]$. // Eq. (2).

5:     Sample decision $\pi^t \sim p^t$ and update estimation oracle with $(\pi^t, r^t, o^t)$.

---

**Online estimation.** Estimation-to-Decisions (Algorithm 1) is a reduction that lifts algorithms for on-line estimation into algorithms for decision making. An online estimation oracle, denoted by $\mathbf{Alg}_{\mathsf{Est}}$, is an algorithm that, using knowledge of the class $\mathcal{M}$, estimates the underlying model $M^\star$ from data in a sequential fashion. At each round $t$, given the data $\mathcal{H}^{t-1} = (\pi^1, r^1, o^1), \ldots, (\pi^{t-1}, r^{t-1}, o^{t-1})$ observed so far, the estimation oracle computes an estimate $\widehat{M}^t = \mathbf{Alg}_{\mathsf{Est}}^t\Big(\{(\pi^i, r^i, o^i)\}_{i=1}^{t-1}\Big)$ for the true model $M^\star$.

To measure the estimation oracle's performance, we make use of a user-specified *divergence-like function*, which quantifies the discrepancy between models. Formally, we define a divergence-like function (henceforth, "divergence") as any function $D : \Pi \times \mathrm{co}(\mathcal{M}) \times \mathrm{co}(\mathcal{M}) \to \mathbb{R}_+$, with $D^\pi(M \parallel M')$ representing the discrepancy between the models $M$ and $M'$ at the decision $\pi$. Standard choices used in past work [12, 13, 5, 15, 14, 27] include the squared error $D_{\mathsf{sq}}^\pi(M, M') := (f^M(\pi) - f^{M'}(\pi))^2$ for bandit problems, and squared Hellinger distance[1] $D_{\mathsf{H}}^\pi(M, M') := D_{\mathsf{H}}^2(M(\pi), M'(\pi))$ for RL, where for distributions $\mathbb{P}$ and $\mathbb{Q}$, $D_{\mathsf{H}}^2(\mathbb{P}, \mathbb{Q}) := \int (\sqrt{d\mathbb{P}} - \sqrt{d\mathbb{Q}})^2$. We then measure the estimation oracle's performance in terms of *cumulative estimation error* with respect to $D$, defined as

$$\mathbf{Est}^D := \sum_{t=1}^T \mathbb{E}_{\pi^t \sim p^t}\Big[D^{\pi^t}\Big(\widehat{M}^t \parallel M^\star\Big)\Big], \tag{1}$$

where $p^t$ is the conditional distribution over $\pi^t$ given $\mathcal{H}^{t-1}$. We make the following assumption on the algorithm's performance.

**Assumption 1.2.** *At each time $t \in [T]$, the* online estimation oracle $\mathbf{Alg}_{\mathsf{Est}}$ *returns, given* $(\pi^1, r^1, o^1), \ldots, (\pi^{t-1}, r^{t-1}, o^{t-1})$ *with* $(r^i, o^i) \sim M^\star(\pi^i)$ *and* $\pi^i \sim p^i$, *an estimator* $\widehat{M}^t : \Pi \to \Delta(\mathcal{R} \times \mathcal{O})$ *such that* $\mathbf{Est}^D \leq \mathbf{Est}^D(T, \delta)$, *with probability at least* $1 - \delta$, *where* $\mathbf{Est}^D(T, \delta)$ *is a known upper bound.*

For the squared error, one can obtain $\mathbf{Est}^{\mathsf{sq}}(T, \delta) := \mathbf{Est}^{D_{\mathsf{sq}}}(T, \delta) \lesssim \log(|\mathcal{F}_{\mathcal{M}}|/\delta)$, where $\mathcal{F}_{\mathcal{M}} := \{f^M \mid M \in \mathcal{M}\}$, and for Hellinger distance, it is possible to obtain $\mathbf{Est}^{\mathsf{H}}(T, \delta) := \mathbf{Est}^{D_{\mathsf{H}}}(T, \delta) \lesssim \log(|\mathcal{M}|/\delta)$.

**Estimation-to-Decisions.** A general version of the E2D meta-algorithm is displayed in Algorithm 1. At each timestep $t$, the algorithm queries estimation oracle to obtain an estimator $\widehat{M}^t$ using the data $(\pi^1, r^1, o^1), \ldots, (\pi^{t-1}, r^{t-1}, o^{t-1})$ observed so far. The algorithm then computes the decision distribution $p^t$ by solving a min-max optimization problem involving $\widehat{M}^t$ and $\mathcal{M}$ (as well as the divergence $D$), and then samples the decision $\pi^t$ from this distribution.

**The Decision-Estimation Coefficient.** The min-max optimization problem in Algorithm 1 is derived from the *Decision-Estimation Coefficient* (DEC), a complexity measure whose value, for a given scale parameter $\gamma > 0$ and reference model $\overline{M} : \Pi \to \Delta(\mathcal{R} \times \mathcal{O})$, is given by

$$\mathsf{dec}_\gamma^D(\mathcal{M}, \overline{M}) = \inf_{p \in \Delta(\Pi)} \sup_{M \in \mathcal{M}} \mathbb{E}_{\pi \sim p}\Big[f^M(\pi_M) - f^M(\pi) - \gamma \cdot D^\pi(\overline{M} \parallel M)\Big], \tag{2}$$

with $\mathsf{dec}_\gamma^D(\mathcal{M}) := \sup_{\overline{M} \in \mathrm{co}(\mathcal{M})} \mathsf{dec}_\gamma^D(\mathcal{M}, \overline{M})$. Informally, the DEC measures the best tradeoff between suboptimality $(f^M(\pi_M) - f^M(\pi))$ and information gain (measured by $D^\pi(\overline{M} \parallel M)$) that can be achieved by a decision distribution $p$ in the face of a worst-case model $M \in \mathcal{M}$.

---

[1] When $D^\pi(\cdot \parallel \cdot)$ is symmetric, we write $D^\pi(\cdot, \cdot)$ to make this explicit.

The main result of Foster et al. [12] shows that the regret of E2D is controlled by the DEC and the estimation oracle's cumulative error $\mathbf{Est}^D$. Let $\widehat{\mathcal{M}}$ be any set for which $\widehat{M}^t \in \widehat{\mathcal{M}}$ for all $t$ almost surely. We have the following guarantee.

**Theorem 1.1** (Foster et al. [12]). *Algorithm 1 with exploration parameter $\gamma > 0$ guarantees that* $\mathbf{Reg}_{\mathsf{DM}} \leq \sup_{\overline{M} \in \widehat{\mathcal{M}}} \mathsf{dec}_\gamma^D(\mathcal{M}, \overline{M}) \cdot T + \gamma \cdot \mathbf{Est}^D$ *almost surely.*

For the special case of Hellinger distance, standard algorithms (exponential weights) achieve $\mathbf{Est}^{\mathsf{H}}(T, \delta) \lesssim \log(|\mathcal{M}|/\delta)$ with $\widehat{\mathcal{M}} = \mathrm{co}(\mathcal{M})$, so that Theorem 1.1 gives (abbreviating $\mathsf{dec}_\gamma^{\mathsf{H}} \equiv \mathsf{dec}_\gamma^{D_{\mathsf{H}}}$):

$$\mathbf{Reg}_{\mathsf{DM}} \lesssim \mathsf{dec}_\gamma^{\mathsf{H}}(\mathcal{M}) \cdot T + \gamma \cdot \log(|\mathcal{M}|/\delta). \tag{3}$$

**Opportunities for improvement.** Foster et al. [12, 15] provide *lower bounds* on the regret for any decision making problem that have a similar expression to (3), showing that the Decision-Estimation Coefficient with Hellinger distance ($\mathsf{dec}_\gamma^{\mathsf{H}}(\mathcal{M})$) plays a fundamental role in determining the statistical complexity of decision making. However, these lower bounds do contain the estimation term $\mathbf{Est}^{\mathsf{H}}(T, \delta) = \log(|\mathcal{M}|/\delta)$) appearing in (3), and thus capture the price of moving from estimation to decision making (as characterized by the DEC), but not the complexity of estimation itself.

In general, the dependence on $\mathbf{Est}^{\mathsf{H}}(T, \delta) = \log(|\mathcal{M}|/\delta)$ in the upper bound (3) can render the bound loose. In reinforcement learning, working with Hellinger distance necessitates modeling transition probabilities. While this leads to optimal results in some settings, in general the optimal rates for Hellinger estimation error can be prohibitively large, even in settings where model-free (or, *value-based*) methods, which directly model value functions, are known to succeed; this drawback is shared by all subsequent work based on the DEC [13, 5, 15, 14, 27]. A natural solution is to replace Hellinger distance with a divergence (e.g., based on Bellman error) tailored to value function approximation, but naive choices for $D$ along these lines render $\mathsf{dec}_\gamma^D(\mathcal{M}, \overline{M})$ too large to give meaningful guarantees, and Foster et al. [12, 15] left this as an open problem.[2]

## 2 Estimation-to-Decisions with Optimistic Estimation

To derive improved regret bounds that address the shortcomings described in the prequel, we combine Estimation-to-Decisions with a specialized estimation approach introduced by Zhang [31] (see also Dann et al. [6], Agarwal and Zhang [1, 2], Zhong et al. [32]), which we refer to as *optimistic estimation*. We then use this approach to derive regret bounds for model-free reinforcement learning.

### 2.1 Optimistic Estimation

The idea of optimistic estimation is to augment the estimation objective with a bonus that biases the estimator toward models $M \in \mathcal{M}$ for which the value $f^M(\pi_M)$ is large. We present a general version of the technique.

Let a divergence $D^\pi(\cdot \| \cdot)$ be fixed. Following the development in Section 1.2, an *optimistic estimation oracle* $\mathbf{Alg}_{\mathsf{Est}}$ is an algorithm which, at each step $t$, given the observations and rewards collected so far, computes an estimate for the underlying model. For technical reasons, it will be useful to consider *randomized estimators* (Foster et al. [12], Chen et al. [5]) that, at each round, produce a distribution $\mu^t \in \Delta(\mathcal{M})$ over models. Such estimators take the form $\mu^t = \mathbf{Alg}_{\mathsf{Est}}^t \left( \{(\pi^i, r^i, o^i)\}_{i=1}^{t-1} \right)$. where $\mu^t \in \Delta(\mathcal{M})$. For a parameter $\gamma > 0$, we define the *optimistic estimation error* as

$$\mathbf{OptEst}_\gamma^D := \sum_{t=1}^T \mathbb{E}_{\pi^t \sim p^t} \mathbb{E}_{\widehat{M}^t \sim \mu^t} \left[ D^\pi \left( \widehat{M}^t \| M^\star \right) + \gamma^{-1}(f^{M^\star}(\pi_{M^\star}) - f^{\widehat{M}^t}(\pi_{\widehat{M}^t})) \right]. \tag{4}$$

This quantity is similar to (1), but incorporates a bonus term $\gamma^{-1}(f^{M^\star}(\pi_{M^\star}) - f^{\widehat{M}^t}(\pi_{\widehat{M}^t}))$, which encourages the estimation algorithm to *over-estimate* the optimal value $f^{M^\star}(\pi_{M^\star})$ for the underlying model $M^\star$.

---

[2]For simpler model classes, it *is* possible to improve upon (3) by moving from Hellinger distance to lenient notions of estimation error: In bandit problems with Gaussian rewards, it suffices to consider the $D_{\mathsf{sq}}^\pi(M, \overline{M}) := (f^M(\pi) - f^{\overline{M}}(\pi))^2$, which leads to upper bounds that scale with $\log|\mathcal{F}_{\mathcal{M}}| \ll \log|\mathcal{M}|$ [12].

---

**Algorithm 2** Optimistic Estimation-to-Decisions (E2D.Opt)

---

1: **params**: Estimation oracle $\mathbf{Alg}_{\mathsf{Est}}$, Exp. param. $\gamma > 0$, divergence $D$ with suff. stat. space $\Psi$.

2: **for** $t = 1, 2, \cdots, T$ **do**

3:      Receive randomized estimator $\mu^t \in \Delta(\Psi) = \mathbf{Alg}_{\mathsf{Est}}^t\big(\{(\pi^i, r^i, o^i)\}_{i=1}^{t-1}\big)$.

4:      Compute

$$p^t \leftarrow \underset{p \in \Delta(\Pi)}{\arg\min} \ \sup_{M \in \mathcal{M}} \ \mathbb{E}_{\pi \sim p} \, \mathbb{E}_{\widehat{\psi} \sim \mu^t} \Big[ f^{\widehat{\psi}}(\pi_{\widehat{\psi}}) - f^M(\pi) - \gamma \cdot D^\pi\Big(\widehat{\psi} \parallel M\Big) \Big] \ \texttt{// Eq. (6).}$$

5:      Sample decision $\pi^t \sim p^t$ and update estimation oracle with $(\pi^t, r^t, o^t)$.

---

**Assumption 2.1.** *At each time $t \in [T]$, the* optimistic estimation oracle $\mathbf{Alg}_{\mathsf{Est}}$ *returns, given* $(\pi^1, r^1, o^1), \ldots, (\pi^{t-1}, r^{t-1}, o^{t-1})$ *with* $(r^i, o^i) \sim M^\star(\pi^i)$ *and* $\pi^i \sim p^i$, *a randomized estimator* $\mu^t \in \Delta(\mathcal{M})$ *such that* $\mathbf{OptEst}_\gamma^D \leq \mathbf{OptEst}_\gamma^D(T, \delta)$, *with w.p.* $1 - \delta$, *where* $\mathbf{OptEst}_\gamma^D(T, \delta)$ *is a known upper bound.*

For the case of contextual bandits, Zhang [31] proposes an augmented version of the exponential weights algorithm which, for a learning rate parameter $\eta > 0$, sets $\mu(M) \propto \exp\big(-\eta\big(L^t(f^M) - \gamma^{-1} f^M(\pi_M)\big)\big)$, where $L^t(f^M)$ is the squared prediction error for the rewards observed so far. This method achieves $\mathbb{E}\big[\mathbf{OptEst}_\gamma^{\mathsf{sq}}\big] \lesssim \log(|\mathcal{F}_\mathcal{M}|) + \sqrt{T \log|\mathcal{F}_\mathcal{M}|}/\gamma$, and Zhang [31] combines this estimator with posterior sampling to achieve optimal contextual bandit regret. Agarwal and Zhang [1], Zhong et al. [32], Agarwal and Zhang [2] extend this development to reinforcement learning, also using posterior sampling as the exploration mechanism.[3] In what follows, we combine optimistic estimation with Estimation-to-Decisions, which provides a universal mechanism for exploration. Beyond giving guarantees which were previously out of reach for E2D (Section 2.3), this approach generalizes and subsumes posterior sampling, and can succeed in situations where posterior sampling fails (Section 3).

**Remark 2.1.** For the non-optimistic estimation error $\mathbf{Est}^D$, it is possible to obtain low error for well-behaved losses such as the square loss and Hellinger distance without the use of randomization by appealing to improper mixture estimators (e.g., Foster et al. [12]). We show in Section 3 that for such divergences, randomization does not lead to statistical improvements. For the optimistic estimation error (4), randomization is essential due to the presence of the term $\gamma^{-1}\big(f^{M^\star}(\pi_{M^\star}) - f^{\widehat{M}^t}(\pi_{\widehat{M}^t})\big)$. $\triangleleft$

**Sufficient statistics.** Before proceeding, we note that many divergences of interest have the useful property that they depend on the estimated model $\widehat{M}$ only through a "sufficient statistic" for the model class under consideration. Formally, there exists a *sufficient statistic space* $\Psi$ and *sufficient statistic* $\psi : \mathcal{M} \to \Psi$ with the property that we can write (overloading notation)

$$D^\pi(M \parallel M') = D^\pi(\psi(M) \parallel M'), \quad f^M(\pi) = f^{\psi(M)}(\pi), \quad \text{and} \quad \pi_M = \pi_{\psi(M)}$$

for all models $M, M'$. In this case, it suffices for the online estimation oracle to directly estimate the sufficient statistic by producing a randomized estimator $\mu^t \in \Delta(\Psi)$. We measure performance via

$$\mathbf{OptEst}_\gamma^D := \sum_{t=1}^T \mathbb{E}_{\pi^t \sim p^t} \, \mathbb{E}_{\widehat{\psi}^t \sim \mu^t} \Big[ D^{\pi^t}\Big(\widehat{\psi}^t \parallel M^\star\Big) + \gamma^{-1}(f^{M^\star}(\pi_{M^\star}) - f^{\widehat{\psi}^t}(\pi_{\widehat{\psi}^t})) \Big] \qquad (5)$$

Examples include bandit problems, where one may use squared estimation error $D_{\mathsf{sq}}^\pi(\cdot, \cdot)$ and take $\psi(\mathcal{M}) = f^M$, and model-free reinforcement learning, where we show that by choosing the divergence $D$ appropriately, one can use *Q-value functions* as a sufficient statistic. Note that we only focus on sufficient statistics for the first argument to $D^\pi(\cdot \parallel \cdot)$, since this is the quantity we wish to estimate.

## 2.2 Algorithm and Main Result

We provide an *optimistic* variant of the E2D meta-algorithm (E2D.Opt) in Algorithm 2. At each timestep $t$, the algorithm calls the estimation oracle to obtain a randomized estimator $\mu^t$ using the data

---

[3]Dann et al. [6] also apply the optimistic estimation idea to model-free reinforcement learning, but do not provide *online* estimation guarantees.

$(\pi^1, r^1, o^1), \ldots, (\pi^{t-1}, r^{t-1}, o^{t-1})$ collected so far. The algorithm then uses the estimator to compute a distribution $p^t \in \Delta(\Pi)$ and samples $\pi^t$ from this distribution, with the main change relative to Algorithm 1 being that the minimax problem in Algorithm 2 is derived from an "optimistic" variant of the DEC, which we refer to as the *Optimistic Decision-Estimation Coefficient*. For $\mu \in \Delta(\mathcal{M})$, define

$$\mathsf{o\text{-}dec}_\gamma^D(\mathcal{M}, \mu) = \inf_{p \in \Delta(\Pi)} \sup_{M \in \mathcal{M}} \mathbb{E}_{\pi \sim p} \mathbb{E}_{\overline{M} \sim \mu} \left[ f^{\overline{M}}(\pi_{\overline{M}}) - f^M(\pi) - \gamma \cdot D^\pi(\overline{M} \parallel M) \right]. \quad (6)$$

and $\mathsf{o\text{-}dec}_\gamma^D(\mathcal{M}) = \sup_{\mu \in \Delta(\mathcal{M})} \mathsf{o\text{-}dec}_\gamma^D(\mathcal{M}, \mu)$. The Optimistic DEC has two difference from the original DEC. First, it is parameterized by a distribution $\mu \in \Delta(\mathcal{M})$ rather than a reference model $\overline{M} : \Pi \to \Delta(\mathcal{R} \times \mathcal{O})$, which reflects the use of randomized estimators; the value in (6) takes the expectation over a reference model $\overline{M}$ drawn from this distribution (this modification also appears in the randomized DEC introduced in Foster et al. [12]). Second, and more critically, the optimal value $f^M(\pi_M)$ in (2) is replaced by the optimal value $f^{\overline{M}}(\pi_{\overline{M}})$ for the (randomized) reference model. This seemingly small change is the main advantage of incorporating optimistic estimation, and makes it possible to bound the Optimistic DEC for certain divergences $D$ for which the value of the unmodified DEC would otherwise be unbounded (cf. Section 3).

**Remark 2.2.** When the divergence $D$ admits a sufficient statistic $\psi : \mathcal{M} \to \Psi$, for any distribution $\mu \in \Delta(\mathcal{M})$, if we define $\nu \in \Delta(\Psi)$ via $\nu(\psi) = \mu(\{M \in \mathcal{M} : \psi(M) = \psi\})$, we have

$$\mathsf{o\text{-}dec}_\gamma^D(\mathcal{M}, \mu) = \inf_{p \in \Delta(\Pi)} \sup_{M \in \mathcal{M}} \mathbb{E}_{\pi \sim p} \mathbb{E}_{\psi \sim \nu} \left[ f^\psi(\pi_\psi) - f^M(\pi) - \gamma \cdot D^\pi(\psi \parallel M) \right].$$

In this case, by overloading notation slightly, we may write $\mathsf{o\text{-}dec}_\gamma^D(\mathcal{M}) = \sup_{\nu \in \Delta(\Psi)} \mathsf{o\text{-}dec}_\gamma^D(\mathcal{M}, \nu)$. ◁

**Main result.** Our main result shows that the regret of Optimistic Estimation-to-Decisions is controlled by the Optimistic DEC and the optimistic estimation error for the oracle $\mathbf{Alg}_{\mathsf{Est}}$.

**Theorem 2.1.** *For any $\delta > 0$, Algorithm 2 ensures that with probability at least $1 - \delta$,*

$$\mathbf{Reg}_{\mathsf{DM}} \leq \mathsf{o\text{-}dec}_\gamma^D(\mathcal{M}) \cdot T + \gamma \cdot \mathbf{OptEst}_\gamma^D(T, \delta). \quad (7)$$

This regret bound has the same structure as Theorem 1.1, with the DEC and estimation error replaced by their optimistic counterparts. In the remainder of the paper, we show that 1) by adopting *asymmetric* divergences specialized to reinforcement learning, this result leads to the first guarantees for model-free RL with E2D, but 2) for symmetric divergences such as Hellinger distance, the result never improves upon Theorem 1.1.

**Estimation with batching.** For our application to reinforcement learning, we generalize the results above to accomodate estimation algorithms that draw *batches* of multiple samples from each distribution $p^t$. Given a *batch size* $n$, we break the $T$ rounds of the decision making protocol into $K := T/n$ contiguous epochs (or, "iterations"). Within each epoch, the learner's distribution $p^k$ is unchanged (we index by $k$ rather than $t$ to reflect this), and we create a *batch* $B^k = \{(\pi^{k,l}, r^{k,l}, o^{k,l})\}_{l=1}^n$ by sampling $\pi^{k,l} \sim p^k$ independently and observing $(r^{k,l}, o^{k,l}) \sim M^\star(\pi^{k,l})$ for each $l \in [n]$. We can then appeal to estimation algorithms of the form $\mu^k = \mathbf{Alg}_{\mathsf{Est}}^k(\{B^k\}_{i=1}^{k-1})$. Regret bounds for a variant of E2D.Opt with batching are given in Appendix B.1.

### 2.3 Application to Model-Free Reinforcement Learning

In this section, we use Optimistic Estimation-to-Decisions to provide sample-efficient guarantees for model-free reinforcement learning with *bilinear classes* [11], a general class of tractable reinforcement learning problems which encompasses many settings [17, 26, 7, 29, 18, 23, 3, 20, 10, 22, 9, 33].

**Reinforcement learning preliminaries.** To state our results, let us recall how reinforcement learning fits into the DMSO framework. We consider an episodic, finite-horizon reinforcement learning setting. With $H$ denoting the horizon, each model $M \in \mathcal{M}$ specifies a non-stationary Markov decision process $M = \{\mathcal{S}, \mathcal{A}, \{P_h^M\}_{h=1}^H, \{R_h^M\}_{h=1}^H, d_1\}$, where $\mathcal{S}$ is the state space, $\mathcal{A}$ is the action space, $P_h^M : \mathcal{S} \times \mathcal{A} \to \Delta(\mathcal{S})$ is the probability transition distribution at step $h$, $R_h^M : \mathcal{S} \times \mathcal{A} \to \Delta(\mathbb{R})$ is the reward distribution, and $d_1 \in \Delta(\mathcal{S}_1)$ is the initial state distribution. We allow the reward distribution and transition kernel to vary across models in $\mathcal{M}$, but assume that the initial state distribution is fixed.

For a fixed MDP $M \in \mathcal{M}$, each episode proceeds under the following protocol. At the beginning of the episode, the learner selects a randomized, non-stationary *policy* $\pi = (\pi_1, \ldots, \pi_H)$, where $\pi_h : \mathcal{S} \to \Delta(\mathcal{A})$; we let $\Pi_{\mathrm{RNS}}$ (for "randomized, non-stationary") denote the set of all such policies. The episode then evolves through the following process, beginning from $s_1 \sim d_1$: For $h = 1, \ldots, H$: $a_h \sim \pi_h(s_h)$, $r_h \sim R_h^M(s_h, a_h)$, and $s_{h+1} \sim P_h^M(\cdot \mid s_h, a_h)$. For notational convenience, we take $s_{H+1}$ to be a deterministic terminal state. We assume for simplicity that $\mathcal{R} \subseteq [0, 1]$ (that is, $\sum_{h=1}^H r_h \in [0, 1]$ almost surely). Within the DMSO framework, at each time $t$, the learning agent chooses $\pi^t \in \Pi_{\mathrm{RNS}}$, then observes the cumulative reward $r^t = \sum_{h=1}^H r_h^t$ and trajectory $o^t := (s_1^t, a_1^t, r_1^t), \ldots, (s_H^t, a_H^t, r_H^t)$ that results from executing $\pi^t$.

*Value functions.* The value for a policy $\pi$ under $M$ is given by $f^M(\pi) := \mathbb{E}^{M,\pi}\left[\sum_{h=1}^H r_h\right]$, where $\mathbb{E}^{M,\pi}[\cdot]$ denotes expectation under the process above. For a given model $M$ and policy $\pi$, we define the state-action value function and state value functions via $Q_h^{M,\pi}(s, a) = \mathbb{E}^{M,\pi}\left[\sum_{h'=h}^H r_{h'} \mid s_h = s, a_h = a\right]$, and $V_h^{M,\pi}(s) = \mathbb{E}^{M,\pi}\left[\sum_{h'=h}^H r_{h'} \mid s_h = s\right]$. We define $\pi_M$ as the optimal policy, which maximizes $Q_h^{M,\pi_M}(s, a)$ for all states simultaneously. We abbreviate $Q^{M,\star} \equiv Q^{M,\pi_M}$.

*Value function approximation.* To apply our results to reinforcement learning, we take a model-free (or, value function approximation) approach, and estimate value functions for the underlying MDP $M^\star$; this contrasts with model-based methods, such as those considered in Foster et al. [12], which estimate transition probabilities for $M^\star$ directly. We assume access to a class $\mathcal{Q}$ of value functions of the form $Q = (Q_1, \ldots, Q_H)$.

**Assumption 2.2.** *The value function class $\mathcal{Q}$ has $Q^{M^\star,\star} \in \mathcal{Q}$, where $M^\star$ is the underlying model.*

For $Q = (Q_1, \ldots, Q_H) \in \mathcal{Q}$, we define $\pi_Q = (\pi_{Q,1}, \ldots, \pi_{Q,H})$ via $\pi_{Q,h}(s) = \arg\max_{a \in \mathcal{A}} Q_h(s, a)$. We define $\Pi_{\mathcal{Q}} = \{\pi_Q \mid Q \in \mathcal{Q}\}$ as the induced policy class. While is not necessary for our results, we mention in passing that the class $\mathcal{Q}$, under Assumption 2.2, implicitly induces a model class via $\mathcal{M}_{\mathcal{Q}} := \{M \mid Q^{M,\star} \in \mathcal{Q}\}$.

*Bilinear classes.* The bilinear class framework [11] gives structural conditions for sample-efficient reinforcement learning that capture most known settings where tractable guarantees are possible. The following is an adaptation of the definition from Du et al. [11].[4]

**Definition 2.1** (Bilinear class). *An MDP $M$ is said to be bilinear with dimension $d$ relative to a class $\mathcal{Q}$ if:*

(i) *There exist functions $W_h(\cdot; M) : \mathcal{Q} \to \mathbb{R}^d$, $X_h(\cdot; M) : \mathcal{Q} \to \mathbb{R}^d$ such that for all $Q \in \mathcal{Q}$ and $h \in [H]$,*

$$\left|\mathbb{E}^{M,\pi_Q}\left[Q_h(s_h, a_h) - r_h - \max_{a' \in \mathcal{A}} Q_{h+1}(s_{h+1}, a')\right]\right| \leq |\langle X_h(Q; M), W_h(Q; M)\rangle|. \quad (8)$$

(ii) *Let $z_h := (s_h, a_h, r_h, s_{h+1})$. There exists a collection of estimation policies $\{\pi_Q^{\mathrm{est}}\}_{Q \in \mathcal{Q}}$ and a discrepancy function $\ell_h^{\mathrm{est}}(\cdot; \cdot) : \mathcal{Q} \times \mathcal{Z} \to \mathbb{R}$ such that for all $Q, Q' \in \mathcal{Q}$ and $h \in [H]$,[5]*

$$|\langle X_h(Q; M), W_h(Q'; M)\rangle| = \left|\mathbb{E}^{M,\pi_Q \circ_h \pi_Q^{\mathrm{est}}}\left[\ell_h^{\mathrm{est}}(Q'; z_h)\right]\right|. \quad (9)$$

*If $\pi_Q^{\mathrm{est}} = \pi_Q$, we say that estimation is* on-policy. *We assume $|\mathbb{E}^{M,\pi}[\ell_h^{\mathrm{est}}(Q^{M,\star}; z_h)]| = 0$ for all $\pi$.*

We let $d_{\mathrm{bi}}(\mathcal{Q}; M)$ denote the minimal dimension $d$ for which the bilinear property holds for $M$. For a model class $\mathcal{M}$, we define $d_{\mathrm{bi}}(\mathcal{Q}; \mathcal{M}) = \sup_{M \in \mathcal{M}} d_{\mathrm{bi}}(\mathcal{Q}; M)$. We let $L_{\mathrm{bi}}(\mathcal{Q}; M) \geq 1$ denote any almost sure upper bound on $|\ell_h^{\mathrm{est}}(Q; z_h)|$ under $M$.

---

[4]For the sake of simplicity, we adopt a less general definition than Du et al. [11]: We 1) assume that the "hypothesis class" is parameterized by the $Q$-function class $\mathcal{Q}$, and 2) limit to discrepancy functions that do not explicitly depend on the function $Q$ indexing the factor $X_h(Q; M)$. The results here readily extend to the full definition.

[5]For $\pi$ and $\pi'$, $\pi \circ_h \pi'$ denotes the policy that follows $\pi$ for layers $1, \ldots, h-1$ and follows $\pi'$ for layers $h, \ldots, H$.

## 2.4 Guarantees for Bilinear Classes

We now apply our main results to derive regret bounds for bilinear classes. We first provide optimistic estimation guarantees, then bound the Optimistic DEC, and conclude by applying E2D.Opt.

We take $\Psi = \mathcal{Q}$ as the sufficient statistic space, with $\psi(M) := Q^{M,\star}$, and define $f^Q(\pi_Q) := \mathbb{E}_{s_1 \sim d_1}[Q_1(s_1, \pi_Q(s_1))]$. For the divergence $D$, we appeal to squared discrepancy, in the vein of Jiang et al. [17], Du et al. [11]:

$$D_{\mathsf{bi}}^\pi(Q \parallel M) = \sum_{h=1}^H \big(\mathbb{E}^{M,\pi}[\ell_h^{\mathrm{est}}(Q; z_h)]\big)^2. \tag{10}$$

We abbreviate $\mathbf{OptEst}_\gamma^{\mathsf{bi}} = \mathbf{OptEst}_\gamma^{D_{\mathsf{bi}}}$ and $\mathsf{o\text{-}dec}_\gamma^{\mathsf{bi}}(\mathcal{M}, \mu) = \mathsf{o\text{-}dec}_\gamma^{D_{\mathsf{bi}}}(\mathcal{M}, \mu)$.

**Estimation.** To perform estimation, we approximate the average discrepancy in (14) from samples (drawing a batch of $n$ samples at each step; cf. Algorithm 3), then appeal to the exponential weights method for online learning, with a bonus to enforce optimism. See Algorithm 6 (deferred to Appendix D for space).

**Proposition 2.1.** *For any batch size $n$ (with $K := T/n$) and parameter $\gamma \geq 1$, Algorithm 6, with an appropriate learning rate, ensures that with probability at least $1 - \delta$,*

$$\mathbf{OptEst}_\gamma^{\mathsf{bi}} \lesssim \frac{\sqrt{K \log|\mathcal{Q}|}}{\gamma} + H L_{\mathsf{bi}}^2(\mathcal{Q}; M^\star) \log(|\mathcal{Q}|KH/\delta)\left(1 + \frac{K}{n}\right), \tag{11}$$

*whenever $M^\star$ is bilinear relative to $\mathcal{Q}$ and Assumption 2.2 is satisfied.*

This result does not actually use the bilinear class structure, and gives a bound on (14) for any choice of $\ell_h^{\mathrm{est}}$. Similar to the optimistic estimation result given by Zhang [31] for contextual bandits, this guarantee consists of "slow" term $\frac{\sqrt{K \log|\mathcal{Q}|}}{\gamma}$ resulting from optimism (which decays as $\gamma$ grows), and a "fast" term. However, compared to the contextual bandit setting, the fast term, $\log(|\mathcal{Q}|KH/\delta)\left(1 + \frac{K}{n}\right)$, scales with the ratio $\frac{K}{n}$, which reflects sampling error in the estimated discrepancy, and necessitates a large batch size. Previous algorithms for bilinear classes [17, 11] require large batch sizes for similar reasons (specifically, because the expectation in (14) appears inside the square, it is not possible to form an unbiased estimate for $D_{\mathsf{bi}}$ directly).

**Bounding the DEC.** A bound on the Optimistic DEC follows by adapting arguments in Foster et al. [12]; our result has slightly improved dependence on $H$ compared to the bounds in that work.

**Proposition 2.2.** *Let $\mathcal{M}$ be any model class for which the bilinear class property holds relative to $\mathcal{Q}$.*
*(1) In the on-policy case where $\pi_Q^{\mathrm{est}} = \pi_Q$, we have that for all $\gamma > 0$, $\mathsf{o\text{-}dec}_\gamma^{\mathsf{bi}}(\mathcal{M}) \lesssim \frac{H \cdot d_{\mathsf{bi}}(\mathcal{Q};\mathcal{M})}{\gamma}$.*
*(2) In the general case ($\pi_Q^{\mathrm{est}} \neq \pi_Q$), we have that for all $\gamma \geq H^2 d_{\mathsf{bi}}(\mathcal{Q};\mathcal{M})$, $\mathsf{o\text{-}dec}_\gamma^{\mathsf{bi}}(\mathcal{M}) \lesssim \sqrt{\frac{H^2 \cdot d_{\mathsf{bi}}(\mathcal{Q};\mathcal{M})}{\gamma}}$.*

Combining Theorem B.1, Proposition 2.1, and Proposition 2.2, we obtain the following result.

**Corollary 2.1** (Regret bound for bilinear classes)**.** *Let $\mathcal{Q}$ be given. Assume that $M^\star \in \mathcal{M}$, where $\mathcal{M}$ is bilinear relative to $\mathcal{Q}$, and that Assumption 2.2 holds. Abbreviate $d \equiv d_{\mathsf{bi}}(\mathcal{Q};\mathcal{M})$ and $L \equiv L_{\mathsf{bi}}(\mathcal{Q};\mathcal{M}) := \sup_{M \in \mathcal{M}} L_{\mathsf{bi}}(\mathcal{Q};M)$. For an appropriate choice of $n$ and $\gamma$, Algorithm 3, using the algorithm from Proposition 2.1 as an oracle, enjoys the following guarantees with probability at least $1 - \delta$:*
*(1) In the on-policy case where $\pi_Q^{\mathrm{est}} = \pi_Q$: $\mathbf{Reg}_{\mathsf{DM}} \lesssim (H^2 d L^2 \log(|\mathcal{Q}|TH/\delta))^{1/2} T^{3/4}$.*
*(2) In the general case where $\pi_Q^{\mathrm{est}} \neq \pi_Q$: $\mathbf{Reg}_{\mathsf{DM}} \lesssim (H^6 d^2 L^4 \log^3(|\mathcal{Q}|TH/\delta))^{1/6} T^{5/6}$.*

This is the first regret bound for model-free reinforcement learning with the Estimation-to-Decisions meta-algorithm. Importantly, the result scales only with the horizon, the dimension $d$, and the capacity $\log|\mathcal{Q}|$. Improving the dependence on $T$ in Corollary 2.1 is an interesting question: currently, there are no algorithms for general bilinear classes that achieve $\sqrt{T}$ regret without additional assumptions. Let us emphasize that regret bounds for bilinear classes can already be achieved by a number of existing methods [11, 19]. The contribution here is to show that such guarantees can be achieved through the general DEC framework, thereby placing this line of research on stronger foundations.

**Tighter guarantees under Bellman completeness.** In Appendix B.2 (deferred for space), we adapt techniques from Agarwal and Zhang [2] to derive tighter estimation guarantees when $\mathcal{Q}$ satisfies a *Bellman completeness* property (e.g., Zanette et al. [30], Jin et al. [19]), by appealing to refined algorithms tailored to squared Bellman error. We use this to derive tighter regret bounds for bilinear classes ($T^{2/3}$ instead of $T^{3/4}$).

## 3 Understanding the Role of Optimistic Estimation

We close with discussion and interpretation of our results.

**When does optimistic estimation help?** Perhaps the most pressing question at this point is to understand when the regret bound for E2D.Opt (Theorem 2.1) improves upon the corresponding regret bound for vanilla E2D (Theorem 1.1). In what follows, we show that: (1) For any divergence $D$, the Optimistic DEC is equivalent to a variant of the original DEC which incorporates randomized estimators [12], but with the arguments to the divergence *flipped*; (2) For divergences $D$ that satisfy a triangle inequality, this randomized DEC is equivalent to the original DEC itself. Together these results show that the improvement given by the Optimistic DEC is limited to *asymmetric* divergences such as the bilinear divergence in Section 2.3; for more traditional divergences such as Hellinger distance and squared error, the optimistic approach offers no improvement. Our results use the following regularity assumption, satisfied by all standard divergences.

**Assumption 3.1.** *For all $M, \overline{M} \in \mathrm{co}(\mathcal{M})$, $(f^{\overline{M}}(\pi) - f^M(\pi))^2 \leq L_{\mathrm{lip}}^2 \cdot D^\pi(\overline{M} \parallel M)$ for a constant $L_{\mathrm{lip}} > 0$.*

Given a divergence $D$, we define the *flipped divergence*, which swaps the first and second arguments, by $\check{D}^\pi(\overline{M} \parallel M) := D^\pi(M \parallel \overline{M})$. We define the Decision-Estimation Coefficient for randomized estimators [12, 5] as $\underline{\mathsf{dec}}_\gamma^D(\mathcal{M}, \mu) = \inf_{p \in \Delta(\Pi)} \sup_{M \in \mathcal{M}} \mathbb{E}_{\pi \sim p}\left[f^M(\pi_M) - f^M(\pi) - \gamma \cdot \mathbb{E}_{\overline{M} \sim \mu}\left[D^\pi(\overline{M} \parallel M)\right]\right]$, with $\underline{\mathsf{dec}}_\gamma^D(\mathcal{M}) := \sup_{\mu \in \Delta(\mathcal{M})} \underline{\mathsf{dec}}_\gamma^D(\mathcal{M}, \mu)$. This definition is identical to (2), but allows $\overline{M}$ to be randomized.

**Proposition 3.1.** *Whenever Assumption 3.1 holds, we have that for all $\gamma > 0$,*

$$\underline{\mathsf{dec}}_{3\gamma/2}^{\check{D}}(\mathcal{M}) - \frac{L_{\mathrm{lip}}^2}{2\gamma} \leq \mathsf{o\text{-}dec}_\gamma^D(\mathcal{M}) \leq \underline{\mathsf{dec}}_{\gamma/2}^{\check{D}}(\mathcal{M}) + \frac{L_{\mathrm{lip}}^2}{2\gamma}. \tag{12}$$

For settings in which there exists an estimation oracle for which the flipped estimation error $\mathbf{Est}^{\check{D}} = \sum_{t=1}^T \mathbb{E}_{\pi \sim p^t} \mathbb{E}_{\widehat{M}^t \sim \mu^t}\left[D^{\pi^t}\left(M^\star \parallel \widehat{M}^t\right)\right]$ is controlled, this result shows that to match the guarantee in Theorem 2.1, optimism is not required, and it suffices to run a variant of vanilla E2D that incorporates randomized estimators (cf. Foster et al. [12], Section 4.3).

We now turn to the role of randomization. When $D$ is convex in the first argument, we have $\underline{\mathsf{dec}}_\gamma^D(\mathcal{M}) \leq \sup_{\overline{M} \in \mathrm{co}(\mathcal{M})} \mathsf{dec}_\gamma^D(\mathcal{M}, \overline{M}) = \mathsf{dec}_\gamma^D(\mathcal{M})$, but it is not immediately apparent whether the opposite direction of this inequality holds, and one might hope that working with the randomized DEC in Proposition 3.1 would lead to improvements over the non-randomized counterpart in Theorem 1.1. The next result shows that this is not the case: Under mild assumptions on the divergence $D$, randomization offers no improvement.

**Proposition 3.2.** *Let $D$ be any bounded divergence such that for all $M, M', \overline{M}$ and $\pi \in \Pi$, $D^\pi(M \parallel M') \leq C\left(D^\pi(\overline{M} \parallel M) + D^\pi(\overline{M} \parallel M')\right)$. Then for all $\gamma > 0$, $\sup_{\overline{M}} \mathsf{dec}_\gamma^D(\mathcal{M}, \overline{M}) \leq \underline{\mathsf{dec}}_{\gamma/(2C)}^D(\mathcal{M})$.*

**Implication for Hellinger distance.** Squared Hellinger distance is symmetric, satisfies Assumption 3.1 with $L_{\mathrm{lip}} = 1$ whenever $\mathcal{R} \subseteq [0, 1]$, and satisfies the condition in Proposition 3.2 with $C = 2$. Hence, by combining Proposition 3.1 with Proposition 3.2, we obtain the following corollary.

**Corollary 3.1.** *If $\mathcal{R} \subseteq [0, 1]$, then $\mathsf{o\text{-}dec}_{2\gamma}^{\mathsf{H}}(\mathcal{M}) - \frac{1}{\gamma} \leq \sup_{\overline{M}} \mathsf{dec}_\gamma^{\mathsf{H}}(\mathcal{M}, \overline{M}) \leq \mathsf{o\text{-}dec}_{\gamma/6}^{\mathsf{H}}(\mathcal{M}) + \frac{3}{\gamma} \; \forall \gamma > 0$.*

This shows that for Hellinger distance—at least from a statistical perspective—there is no benefit to using the Optimistic DEC or randomized DEC compared to the original version. In particular, this implies that regret bounds based on the randomized DEC with Hellinger distance (such as those found

in recent work of Chen et al. [5]) do not offer improvement over the guarantees for vanilla E2D in Foster et al. [12]. One caveat, though, is that working with the Optimistic DEC, as well as randomized estimators, has potential to give computational improvement, as computing a distribution $p \in \Delta(\Pi)$ that minimizes $\text{o-dec}_\gamma^H(\mathcal{M}, \mu)$ might be simpler than computing a corresponding distribution for $\text{dec}_\gamma^H(\mathcal{M}, \overline{M})$ with $\overline{M} \in \text{co}(\mathcal{M})$. We are not currently aware of any examples where such an improvement occurs, as even maintaining a distribution $\mu \in \Delta(\mathcal{M})$ requires intractably large memory for most classes of interest.

**Implication for model-free RL.** The bilinear divergence $D_{\text{bi}}^\pi(Q \parallel M) = \sum_{h=1}^H (\mathbb{E}^{M,\pi}[\ell_h^{\text{est}}(Q; z_h)])^2$ that we adopt in Section 2.3 is asymmetric, as are closely related divergences such as squared Bellman error. Here, there are two reasons why optimistic estimation offers meaningful advantages.

**(1)** By Proposition 3.1 ($D_{\text{bi}}$ satisfies Assumption 3.1 with $L_{\text{lip}} = O(H)$), a natural alternative to optimistic estimation is to estimate with respect to the flipped divergence $\check{D}_{\text{bi}}$, then appeal to Algorithm 3 of Foster et al. [12]. The issue with this approach is that minimizing the flipped estimation error, which takes the form

$$\textbf{Est}^{\check{D}_{\text{bi}}} = \sum_{t=1}^T \mathbb{E}_{\pi \sim p^t} \mathbb{E}_{\widehat{M}^t \sim \mu^t} \left[ D_{\text{bi}}^{\pi^t}\left(Q^{M^\star,\star} \parallel \widehat{M}^t\right)\right] = \sum_{t=1}^T \mathbb{E}_{\pi \sim p^t} \mathbb{E}_{\widehat{M}^t \sim \mu^t} \left[\sum_{h=1}^H \left(\mathbb{E}^{\widehat{M}^t, \pi^t}\left[\ell_h^{\text{est}}(Q^{M^\star,\star}; z_h)\right]\right)^2\right],$$

is challenging in model-free settings; we are not aware of any algorithms that accomplish this.[6]

**(2)** Alternatively, a second choice is to perform estimation with respect to the un-flipped divergence $D_{\text{bi}}$ (which can be accomplished with Proposition 2.1 by taking $\gamma \to \infty$), and appeal to vanilla E2D (either Algorithm 1, or Algorithm 3 of Foster et al. [12] if one wishes to incorporate randomized estimators). However, the following result shows that unlike the Optimistic DEC, the original DEC with the divergence $D_{\text{bi}}$ does not admit a favorable bound, even for tabular reinforcement learning.

**Proposition 3.3.** *Let $\mathcal{M}$ be the class of all horizon-$H$ tabular MDPs with $|\mathcal{S}| = 2$ and $|\mathcal{A}| = 2$. Consider the discrepancy function $\ell_h^{\text{est}}(Q; z_h) = (Q_h(s_h, a_h) - r_h - \max_{a' \in \mathcal{A}} Q_{h+1}(s_{h+1}, a'))$. Then we have $\text{o-dec}_\gamma^{\text{bi}}(\mathcal{M}) \lesssim \frac{H}{\gamma}$, yet there exists $\overline{M} \in \mathcal{M}$ for which $\text{dec}_\gamma^{\text{bi}}(\mathcal{M}, \overline{M}) \gtrsim \frac{2^H}{\gamma} \wedge 1$.*

**Insufficiency of posterior sampling.** For contextual bandits, where $\text{o-dec}_\gamma^{\text{bi}}(\mathcal{M}) \lesssim \frac{|\mathcal{A}|}{\gamma}$ [31], and bilinear classes, where $\text{o-dec}_\gamma^{\text{bi}}(\mathcal{M}) \lesssim \frac{H \cdot d_{\text{bi}}(\mathcal{M})}{\gamma}$ (Proposition 2.2), a strategy that achieves the bound on the Optimistic DEC is *posterior sampling* (this is also the approach taken in Agarwal and Zhang [1, 2], Zhong et al. [32]). That is, given a distribution $\mu \in \Delta(\mathcal{M})$, choosing $p(\pi) = \mu(\{M \in \mathcal{M} \mid \pi_M = \pi\})$ in (6) certifies the desired bound on $\text{o-dec}_\gamma^D(\mathcal{M}, \mu)$ for these examples. Optimistic Estimation-to-Decisions subsumes and generalizes posterior sampling, but in light of the fact that this simple strategy succeeds for large classes of problems, it is reasonable to ask if there is a sense in which posterior sampling is universal, and whether it can achieve the value of the Optimistic DEC for any model class. This would be desirable, since it would indicate that solving the minimax problem in Algorithm 2 is not necessary. The following sample shows that this is not the case: there are model classes (specifically, MDPs with a constant number of actions) for which the regret of posterior sampling is exponentially large compared to the regret of Algorithm 2.

**Proposition 3.4.** *Consider the divergence $D_H^\pi(\cdot, \cdot)$. For any $S \in \mathbb{N}$ and $H \geq \log_2(S)$, there exists a class of horizon-$H$ MDPs $\mathcal{M}$ with $|\mathcal{S}| = S$ and $|\mathcal{A}| = 3$ that satisfies the following properties:*
- *There exists an estimation oracle with $\textbf{OptEst}_\gamma^H \lesssim \log(S/\delta)$ w.p. at least $1 - \delta$ for all $\gamma > 0$.*
- *Posterior sampling, which sets $p^t(\pi) = \mu^t(\{M \mid \pi_M = \pi\})$, has $\mathbb{E}[\textbf{Reg}_{\text{DM}}] \gtrsim S \wedge 2^{\Omega(H)}$.*
- *Algorithm 2 with divergence $D = D_H^\pi(\cdot, \cdot)$ has $\mathbb{E}[\textbf{Reg}_{\text{DM}}] \leq \widetilde{O}(\sqrt{T \log(S)})$.*

This shows that posterior sampling does not provide a universal mechanism for exploration, and highlights the need for deliberate strategies such as E2D.

---

[6]While Foster et al. [12] do give regret bounds for model-free RL using this divergence, they only bound Bayesian regret, and do not provide an explicit algorithm for the frequentist setting.

## Acknowledgements

NG is supported by a Fannie & John Hertz Foundation Fellowship and an NSF Graduate Fellowship. AR acknowledges support from the NSF through award DMS-2031883, from the ARO through award W911NF-21-1-0328, and from the DOE through award DE-SC0022199.

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

# Contents of Appendix

## A   Additional Related Work

In this section we discuss additional related work not already covered.

Chen et al. [5] build on the DEC framework by giving regret bounds for a E2D that incorporates randomized estimators, but not optimism. For the case of finite classes, their guarantees scale as roughly

$$\mathbf{Reg}_{\mathsf{DM}} \lesssim \underline{\mathsf{dec}}_\gamma^{D_\mathsf{H}}(\mathcal{M}) \cdot T + \gamma \cdot \log(|\mathcal{M}|/\delta),$$

where $\underline{\mathsf{dec}}_\gamma^{D_\mathsf{H}}(\mathcal{M})$ is the DEC for randomized estimators defined in Section 3. As discussed in Section 3, this regret bound cannot improve upon the guarantees for the original E2D method in Foster et al. [12] beyond constants, as we have $\sup_{\overline{M}} \mathsf{dec}_{4\gamma}^\mathsf{H}(\mathcal{M}, \overline{M}) \le \underline{\mathsf{dec}}_\gamma^{D_\mathsf{H}}(\mathcal{M})$. In addition, since the algorithm does not incorporate optimism, it cannot be directly applied to model-free reinforcement learning settings.

Foster et al. [15] give upper and lower bounds on optimal regret based on a variant of the DEC called the *constrained Decision-Estimation Coefficient*. These results tighten the original regret bounds in Foster et al. [12], but the upper bounds still scale with $\mathbf{Est}^\mathsf{H}(T, \delta) = \log(|\mathcal{M}|/\delta)$, rendering them unsuitable for model-free RL. Nonetheless it would be interesting to explore whether the techniques in this work can be combined with optimistic estimation.

## B   Additional Results

### B.1   Optimistic Estimation-to-Decisions with Batching

---

**Algorithm 3** Optimistic Estimation-to-Decisions (E2D.Opt) with Batching

---

1: **parameters**:

      Online estimation oracle $\mathbf{Alg}_{\mathsf{Est}}$ with batch size $n$.

      Exploration parameter $\gamma > 0$.

      Divergence $D(\cdot \parallel \cdot)$ with sufficient statistic space $\Psi$.

2: Let $K := T/n$.

3: **for** $k = 1, 2, \cdots, K$ **do**

4:    Receive randomized estimator $\mu^k \in \Delta(\Psi) = \mathbf{Alg}_{\mathsf{Est}}^t\big((B^i)_{i=1}^{k-1}\big)$.

5:    Get $p^k \leftarrow \arg\min_{p \in \Delta(\Pi)} \sup_{M \in \mathcal{M}} \mathbb{E}_{\pi \sim p} \mathbb{E}_{\widehat{\psi} \sim \mu^k} \Big[ f^{\widehat{\psi}}(\pi_{\widehat{\psi}}) - f^M(\pi) - \gamma \cdot D^\pi\big(\widehat{\psi} \parallel M\big) \Big]$. // Eq. (6).

6:    Sample batch $B^k = \{(\pi^{k,l}, r^{k,l}, o^{k,l})\}_{l=1}^n$ where $\pi^{k,l} \sim p^k$ and $(r^{k,l}, o^{k,l}) \sim M^\star(\pi^{k,l})$, and update estimation oracle with $B^k$.

---

For our application to reinforcement learning, it will be useful to generalize E2D.Opt to accomodate estimation algorithms that draw *batches* of multiple samples from each distribution $p^t$. Given a *batch*

*size* $n$, we break the $T$ rounds of the decision making protocol into $K := T/n$ contiguous epochs (or, "iterations"). Within each epoch, the learner's distribution $p^k$ is unchanged (we index by $k$ rather than $t$ to reflect this), and we create a *batch* $B^k = \{(\pi^{k,l}, r^{k,l}, o^{k,l})\}_{l=1}^n$ by sampling $\pi^{k,l} \sim p^k$ independently and observing $(r^{k,l}, o^{k,l}) \sim M^\star(\pi^{k,l})$ for each $l \in [n]$. We consider estimation oracles of the form $\mu^k = \mathbf{Alg}_{\mathsf{Est}}^k\left(\{B^k\}_{i=1}^{k-1}\right)$, and measure estimation error via

$$\mathbf{OptEst}_\gamma^D := \sum_{k=1}^K \mathbb{E}_{\pi^k \sim p^k} \mathbb{E}_{\widehat{M}^k \sim \mu^k}\left[D^\pi\left(\widehat{M}^k \parallel M^\star\right) + \gamma^{-1}(f^{M^\star}(\pi_{M^\star}) - f^{\widehat{M}^t}(\pi_{\widehat{M}^k}))\right]. \quad (13)$$

We assume that the estimation oracle ensures that with probability at least $1 - \delta$, $\mathbf{OptEst}_\gamma^D \leq \mathbf{OptEst}_\gamma^D(K, n, \delta)$, where $\mathbf{OptEst}_\gamma^D(K, n, \delta)$ is a known upper bound.

Algorithm 3 is a variant of E2D.Opt that incorporates batching. The algorithm updates the distribution $p^k$ in the same fashion as its non-batched counterpart, but does so only at the beginning of each epoch. The main guarantee for this method as follows.

**Theorem B.1.** *Let $T \in \mathbb{N}$ be given, and let $n$ be the batch size. For any $\delta > 0$, Algorithm 3 ensures that with probability at least $1 - \delta$, $\mathbf{Reg}_{\mathsf{DM}} \leq \mathsf{o\text{-}dec}_\gamma^D(\mathcal{M}) \cdot T + \gamma n \cdot \mathbf{OptEst}_\gamma^D(T/n, n, \delta)$.*

See Appendix D.1 for the proof. When working with divergences for which unbiased estimates are unavailable, this approach can lead to stronger guarantees than Theorem 2.1. We refer to the proof of Proposition 2.1 for a concrete example.

## B.2   Model-Free RL: Tighter Guarantees under Bellman Completeness

In this section, we show how to derive tighter estimation guarantees (and consequently tighter regret bounds) when $\mathcal{Q}$ satisfies a *Bellman completeness* assumption [30, 19]. For a given MDP $M$, let $\mathcal{T}_h^M[f](s, a) := \mathbb{E}^M[r_h + \max_{a'} f(s_{h+1}, a') \mid s_h = s, a_h = a]$ denote the Bellman operator for layer $h$.

**Assumption B.1** (Completeness). *We assume that $\mathcal{Q} = \mathcal{Q}_1 \times \cdots \times \mathcal{Q}_H$ is a product class, and that for all $h$ and $Q_{h+1} \in \mathcal{Q}_{h+1}$, $[\mathcal{T}_h^{M^\star} Q_{h+1}] \in \mathcal{Q}_h$.*

As before, we take $\Psi = \mathcal{Q}$, $\psi(M) := Q^{M,\star}$, and $f^Q(\pi_Q) := \mathbb{E}_{s_1 \sim d_1}[Q_1(s_1, \pi_Q(s_1))]$. For the divergence $D$, we now appeal to *squared Bellman error* (e.g., [19, 28]):

$$D_{\mathsf{sbe}}^\pi(Q \parallel M) = \sum_{h=1}^H \mathbb{E}^{M,\pi}\left[(Q_h(s_h, a_h) - [\mathcal{T}_h^M Q_{h+1}](s_h, a_h))^2\right]. \quad (14)$$

We abbreviate $\mathbf{OptEst}_\gamma^{\mathsf{sbe}} = \mathbf{OptEst}_\gamma^{D_{\mathsf{sbe}}}$ and $\mathsf{o\text{-}dec}_\gamma^{\mathsf{sbe}}(\mathcal{M}, \mu) = \mathsf{o\text{-}dec}_\gamma^{D_{\mathsf{sbe}}}(\mathcal{M}, \mu)$.

**Estimation.**   Algorithm 4 performs optimistic online estimation with squared Bellman error. The algorithm is an adaptation of a two-timescale exponential weights strategy originally introduced by Agarwal and Zhang [2] within an optimistic posterior sampling algorithm referred to as TS3. We show that this technique leads to a self-contained online estimation guarantee outside the context of the TS3 algorithm.

**Proposition B.1** (Estimation for square Bellman error). *Assume that $\mathcal{Q}$ satisfies completeness relative to $M^\star$. Moreover assume $\sum_{h=1}^H \sup_{s,a} r_h(s, a) \leq 1$ and $\sup_{Q,h,s,a} Q_h(s, a) \leq 1$. Then for any $\gamma \geq 1$ and $\eta \leq 1/(2^{16}(\log(|\mathcal{Q}|K/\delta) + 1))$, with batch size $n = H$ ($K := T/n$), $\lambda = 1/8$, $\beta = (12\gamma H)^{-1}$ and $\delta > 0$, Algorithm 4 ensures that with probability at least $1 - \delta$,*

$$\mathbf{OptEst}_\gamma^{\mathsf{sbe}} \lesssim \frac{H \log|\mathcal{Q}|}{\eta} + \frac{\eta \log(|\mathcal{Q}|K/\delta)K}{\gamma} + \frac{K}{\gamma^2 H}. \quad (15)$$

*whenever $\mathcal{Q}$ satisfies completeness relative to $M^\star$.*[7]

Note that Proposition B.1 does not make use of the bilinear class assumption, and only requires that $\mathcal{Q}$ satisfies completeness. As such, we expect that this result will find use more broadly.

---

[7]Agarwal and Zhang [2] give a tighter estimation error bound of roughly $\mathbf{OptEst}_\gamma^{\mathsf{sbe}} \lesssim \log^2(|\mathcal{Q}|HK) + \frac{K}{\gamma^2}$, but this result takes advantage of a Bellman rank assumption on the underlying MDP. The estimation error bound we state here does not require any structural assumptions on the MDP under consideration, but gives a worse rate.

---

**Algorithm 4** Two-Timescale Exponential Weights for Bellman Complete Value Function Classes

---

1: Initialize $S^0 = \varnothing$.
2: **for** $k = 1, \ldots, K$ **do**
3:      For any $Q, Q' \in \mathcal{Q}$ and $h \in [H]$, define

$$\Delta_h^k(Q', Q) := Q'(s_h^{k,h}, a_h^{k,h}) - r_h^{k,h} - Q(s_{h+1}^{k,h}),$$

$$q^k(Q'|Q) := q^k(Q'|Q, S^{k-1}) \propto \exp\left( -\lambda \cdot \frac{1}{H} \sum_{s=1}^{k-1} \sum_{h=1}^{H} \Delta_h^s(Q', Q)^2 \right),$$

$$L^k(Q) := \frac{1}{H} \sum_{h=1}^{H} \Delta_h^k(Q, Q)^2 + \frac{1}{\lambda} \log \mathbb{E}_{Q' \sim q^k(\cdot|Q)} \left[ \exp\left( -\lambda \cdot \frac{1}{H} \sum_{h=1}^{H} \Delta_h^k(Q', Q)^2 \right) \right],$$

$$\mu^k(Q) := \mu^k(Q|S^{k-1}) \propto \exp\left( -\eta \sum_{s=1}^{k-1} \left( L^s(Q) - \beta \cdot \frac{1}{H} \sum_{h=1}^{H} \max_a Q(s_1^{s,h}, a) \right) \right).$$

4:      Predict $\mu^k$.
5:      **for** $l = 1, \ldots, H$ **do**
6:          Play $\pi^{k,l} \sim p^k$ and obtain the trajectory $o^{k,l} = (s_1^{k,l}, a_1^{k,l}, r_1^{k,l}), \ldots, (s_H^{k,l}, a_H^{k,l}, r_H^{k,l})$, where $p^k \in \Delta(\Pi)$ is a decision distribution produced by any batched algorithm (e.g., Algorithm 3) that selects a decision adaptively based on $\mu^k$.
7:      Update $S^k \leftarrow S^{k-1} \cup \bigcup_{l=1}^{H} \{s_l^{k,l}, a_l^{k,l}, r_l^{k,l}, s_{l+1}^{k,l}\} \cup \bigcup_{l=1}^{H} \{s_1^{k,l}\}$.

---

**Regret bound for bilinear classes.** To provide regret bounds, we assume that $M^\star$ satisfies the bilinear class property relative to $\mathcal{Q}$ as in Section 2.3. In addition to assuming that $M^\star$ is bilinear, we make the following restrictions: (1) $\pi_Q^{\mathrm{est}} = \pi_Q$, i.e. estimation is on policy, (2) $\ell_h^{\mathrm{est}}(Q; z_h) = Q_h(s_h, a_h) - r_h - \max_{a'} Q_{h+1}(s_{h+1}, a')$, so that $\mathbb{E}^{M,\pi}[\ell_h^{\mathrm{est}}(Q; z_h)]$ is the average Bellman error for $Q$ under $M$.[8] With this discrepancy function, Jensen's inequality implies that $\mathsf{o\text{-}dec}_\gamma^{\mathsf{sbe}}(\mathcal{M}) \leq \mathsf{o\text{-}dec}_\gamma^{\mathsf{bi}}(\mathcal{M})$, so combining Theorem B.1, Proposition B.1, and Proposition 2.2, we obtain the following result.

**Corollary B.1** (Regret bound under completeness). Let $\mathcal{Q}$ be given. Assume that $M^\star \in \mathcal{M}$, where $\mathcal{M}$ is bilinear relative to $\mathcal{Q}$, and that completeness holds. Moreover assume $\sum_{h=1}^{H} \sup_{s,a} r_h(s,a) \leq 1$ and $\sup_{Q,h,s,a} Q_h(s,a) \leq 1$. Abbreviate $d \equiv d_{\mathsf{bi}}(\mathcal{Q}; \mathcal{M})$. For an appropriate choice of $n$ and $\gamma$, Algorithm 3, using Algorithm 4 (with appropriate parameter choice) as an oracle, enjoys the following guarantees with probability at least $1 - \delta$:

$$\mathbf{Reg}_{\mathsf{DM}} \lesssim H d^{1/3} (\log(|\mathcal{Q}|K/\delta))^{4/5} T^{2/3}. \tag{16}$$

This improves upon the $T^{3/4}$-type rate in Corollary 2.1.

## B.3 Proofs from Appendix B.2

Proposition B.1 is an application of more general results given in Appendix C.3, which analyze a generalization of Algorithm 4 for a more general online learning setting. To Proposition B.1, we simply apply these results to the reinforcement learning framework.

**Proof of Proposition B.1.** Let the batch size $n = H$ be fixed, and let $K := T/n$ be the number of epochs. Recall that for each step $k \in [K]$, the estimation oracle is given a batch of examples $B^k = \{(\pi^{k,l}, r^{k,l}, o^{k,l})\}_{l=1}^n$ where $\pi^{k,l} \sim p^k$ and $(r^{k,l}, o^{k,l}) \sim M^\star(\pi^{k,l})$. Each observation (trajectory) takes the form $o^{k,l} = (s_1^{k,l}, a_1^{k,l}, r_1^{k,l}), \ldots, (s_H^{k,l}, a_H^{k,l}, r_H^{k,l})$. We abbreviate $Q^\star = Q^{M^\star, \star}$.

**Estimation algorithm.** For each step $k$, the randomized estimator $\mu^k$ selected as described in Algorithm 4. This algorithm is an instantiation of Algorithm 5 in the general online learning setting described in Appendix C.3, with $\mathcal{G} = \mathcal{Q}$ and for all $h \in [H]$, $\mathcal{X}_h = \mathcal{S} \times \mathcal{A}$, $\mathcal{Y}_h = \mathbb{R} \times \mathcal{S}$ and

---

[8]These restrictions correspond to restricting attention to $Q$-type *Bellman rank*, a special case of the bilinear class property [19, 12].

$\mathcal{W} = \mathcal{S}^H$. The unknown kernels are the transition distributions for the corresponding layers of the MDP $M^\star$, and the loss functions are

$$\ell_{h,1}((s_h, a_h), Q) := Q_h(s_h, a_h),$$
$$\ell_{h,2}((r_h, s_{h+1}), Q) := r_h + \max_a Q_h(s_{h+1}, a),$$

$$\ell_3(\{s_1^l\}_{l \in [H]}) := -\frac{1}{H} \sum_{l=1}^{H} \max_a Q_1(s_1^l, a)$$

Finally, take $x_h^k = (s_h^{k,h}, a_h^{k,h})$, $y_h^k = (r_h^{k,h}, s_{h+1}^{k,h})$ and $w^k = \{s_1^{k,h}\}_{h \in [H]}$. It is important to note that $s_h^{k,h}, a_h^{k,h}, r_h^{k,h}, s_{h+1}^{k,h}$ are taken from different trajectories for $h \in [H]$, so $y_h^k \mid x_h^k$ are independent from one other for $h \in [H]$. Moreover, note that the distributions $p^k \in \Delta(\Pi)$ play the role of nature: the distribution of the tuple $(x_h^k, y_h^k, w^k)$ for $h \in [H]$ is determined by running a policy $\pi^k \sim p^k$ in the ground-truth MDP $M^\star$. With this configuration, observe that in the notation of Appendix C.3, we have, for any $Q$,

$$\mathbb{E}_{x_{1:H}^k} \mathcal{E}(Q, Q, x_{1:H}^k)^2 = \frac{1}{H} \sum_{h=1}^{H} \mathbb{E}_{x_h^k} (\ell_{h,1}(x_h^k, Q) - \mathbb{E}[\ell_{h,2}(y_h, Q) \mid x_h^k])^2$$

$$= \frac{1}{H} \sum_{h=1}^{H} \mathbb{E}_{\pi^{k,h} \sim p^k} \mathbb{E}^{M^\star, \pi^{k,h}} \left[ \left( Q_h(s_h, a_h) - [\mathcal{T}_h^{M^\star} Q_{h+1}](s_h, a_h) \right)^2 \right]$$

$$= \frac{1}{H} \mathbb{E}_{\pi \sim p^k} \mathbb{E}^{M^\star, \pi} \left[ \sum_{h=1}^{H} \left( Q_h(s_h, a_h) - [\mathcal{T}_h^{M^\star} Q_{h+1}](s_h, a_h) \right)^2 \right]$$

$$= \frac{1}{H} \mathbb{E}_{\pi \sim p^k} D_{\mathsf{sbe}}^\pi(Q \parallel M^\star).$$

and

$$\mathbb{E}_{x_{1:H}^k, w^k} \iota^k(Q) = \frac{1}{H} \sum_{l=1}^{H} \mathbb{E}_{\pi^{k,h} \sim p^k} \mathbb{E}^{M^\star, \pi^{k,h}} \left[ \max_a Q_1^*(s_1^{k,l}, a) - \max_a Q_1(s_1^{k,l}, a) \right]$$

$$= f^{M^\star}(\pi_{M^\star}) - f^Q(\pi_Q).$$

**Estimation error bound.** We take $\alpha = 12\beta$, so that Theorem C.1 implies that with probability at least $1 - \delta$,

$$\sum_{k=1}^{K} \mathbb{E}_{Q \sim \mu^k} \left( \frac{1}{H} \mathbb{E}_{\pi \sim p^k} D_{\mathsf{sbe}}^\pi(Q \parallel M^\star) + \alpha(f^{M^\star}(\pi_{M^\star}) - f^Q(\pi_Q)) \right)$$

$$\lesssim \eta \alpha \log(|\mathcal{Q}|K/\delta)K + \log|\mathcal{Q}|/\eta + \alpha^2 K.$$

Then by taking $\alpha = \frac{1}{\gamma H}$, this further implies that with probability at least $1 - \delta$,

$$\mathbf{OptEst}_\gamma^{\mathsf{sbe}} = \sum_{k=1}^{K} \mathbb{E}_{\pi \sim p^k} \mathbb{E}_{Q \sim \mu^k} \left( D_{\mathsf{sbe}}^\pi(Q \parallel M^\star) + \frac{1}{\gamma}(f^{M^\star}(\pi_{M^\star}) - f^Q(\pi_Q)) \right)$$

$$\lesssim H(\eta \alpha \log(|\mathcal{Q}|K/\delta)K + \log|\mathcal{Q}|/\eta + \alpha^2 K)$$

$$\lesssim \frac{H \log|\mathcal{Q}|}{\eta} + \frac{\eta \log(|\mathcal{Q}|K/\delta)K}{\gamma} + \frac{K}{\gamma^2 H}.$$

$$\square$$

**Proof of Corollary B.1.** We choose $n = H$ and apply Algorithm 4 as the estimation oracle. We first consider the "trivial" parameter regime in which $Hd^{1/3}(\log(|\mathcal{Q}|K/\delta))^{-1/5}T^{-1/3} \geq 1/(2^{16}(\log(|\mathcal{Q}|K/\delta) + 1))$. Here, $T \lesssim Hd^{1/3}(\log(|\mathcal{Q}|K/\delta))^{4/5}T^{2/3}$, and thus

$$\mathbf{Reg}_{\mathsf{DM}} \lesssim Hd^{1/3}(\log(|\mathcal{Q}|K/\delta))^{4/5}T^{2/3}.$$

When the case above, does not hold, we proceed as in the theorem statement, choosing $\eta = Hd^{1/3}(\log(|\mathcal{Q}|K/\delta))^{-1/5}T^{-1/3} \leq 1/(2^{16}(\log(|\mathcal{Q}|K/\delta)+1))$. Combining Theorem B.1 and Proposition B.1 then gives

$$\mathbf{Reg}_{\mathsf{DM}} \lesssim \mathsf{o\text{-}dec}_\gamma^{\mathsf{sbe}}(\mathcal{M}) \cdot T + \gamma \frac{H^2 \log|\mathcal{Q}|}{\eta} + \eta \log(|\mathcal{Q}|K/\delta)T + \frac{K}{\gamma^2}$$

with probability at least $1 - \delta$. Next, using Proposition 2.2 to bound $\mathsf{o\text{-}dec}_\gamma^{\mathsf{sbe}}(\mathcal{M})$ in the above display, it follows that

$$\mathbf{Reg}_{\mathsf{DM}} \lesssim \frac{HdT}{\gamma} + \gamma \frac{H^2 \log|\mathcal{Q}|}{\eta} + \eta \log(|\mathcal{Q}|K/\delta)T + \frac{K}{\gamma^2}.$$

We choose $\gamma = d^{2/3}(\log(|\mathcal{Q}|K/\delta))^{-2/5}T^{1/3}$ to obtain

$$\mathbf{Reg}_{\mathsf{DM}} \lesssim Hd^{1/3}(\log(|\mathcal{Q}|K/\delta))^{4/5}T^{2/3}$$

with probability at least $1 - \delta$.

$\square$

## C  Technical Tools

### C.1  Preliminaries

**Lemma C.1.** *For all $x \in [0, 1]$, we have*

$$e^{-x} \leq 1 - (1 - 1/e)x \leq 1 - x/2, \quad \text{and} \quad e^x \leq 1 + (e-1)x \leq 1 + 2x.$$

**Lemma C.2.** *For all $x \geq -1/8$, we have $e^{-x} \leq 1 - x + x^2$.*

**Proof of Lemma C.2.** Let $f(x) = e^{-x} - 1 + x - x^2$. We have $f''(x) = e^{-x} - 2 < 0$ for $x \geq -1/8$. Thus, $f'$ is monotonically decreasing on $x \geq -1/8$, so for $x \in [-1/8, 0]$, $f'(x) \geq f'(0) = 0$. Hence, $f(x)$ is non-decreasing on $x \in [-1/8, 0]$. Furthermore, $f'(x) = -e^{-x} + 1 - 2x \leq 0$ for $x \geq 0$. Thus $f(x)$ obtains maximum value at $x = 0$, and $f(x) \leq f(0) = 0$. $\square$

### C.2  Basic Online Learning Results

In this section we state a technical lemma regarding the performance of the exponential weights algorithm for online learning. Let $\mathcal{G}$ be an abstract set of hypotheses. We consider the following online learning process.

For $t = 1, \ldots, T$:

- Learner predicts a (random) hypothesis $g^t \in \mathcal{G}$.

- Nature reveals $\ell^t \in \mathcal{L} := (\mathcal{G} \to \mathbb{R})$ and learner suffers loss $\ell^t(g^t)$.

We define regret to the class $\mathcal{G}$ via

$$\mathbf{Reg}_{\mathsf{OL}} = \sum_{t=1}^{T} \mathbb{E}_{g^t \sim \mu^t}[\ell^t(g^t)] - \inf_{g \in \mathcal{G}} \sum_{t=1}^{T} \ell^t(g), \tag{17}$$

where $\mu^t \in \Delta(\mathcal{G})$ is the learner's randomization distribution for step $t$.

**Lemma C.3.** *Consider the exponential weights update method with learning rate $\eta > 0$, which sets*

$$\mu^t(g) \propto \exp\left(-\eta \sum_{i < t} \ell^i(g)\right).$$

*For any sequence of non-negative loss functions $\ell^1, \ldots, \ell^T$, this algorithm satisfies*

$$\mathbf{Reg}_{\mathsf{OL}} \leq \frac{\eta}{2} \sum_{t=1}^{T} \mathbb{E}_{g^t \sim \mu^t}\left[(\ell^t(g^t))^2\right] + \frac{\log|\mathcal{G}|}{\eta}. \tag{18}$$

In addition, for any sequence of loss functions $\ell^1, \ldots, \ell^T$ with $\ell^t(g) \in [-L, L]$ for all $g \in \mathcal{G}$, if $\eta \leq (2L)^{-1}$, then

$$\mathbf{Reg}_{\mathsf{OL}} \leq 2\eta \sum_{t=1}^{T} \mathbb{E}_{g^t \sim \mu^t} \big[ (\ell^t(g^t) - \mathbb{E}_{g' \sim \mu^t}[\ell^t(g')])^2 \big] + \frac{\log|\mathcal{G}|}{\eta} \leq 4\eta \sum_{t=1}^{T} \mathbb{E}_{g^t \sim \mu^t} \big[ (\ell^t(g^t))^2 \big] + \frac{\log|\mathcal{G}|}{\eta} \tag{19}$$

**Proof of Lemma C.3.** A standard telescoping argument combined with the fact that $-\inf_{g \in \mathcal{G}} \sum_{t=1}^{T} \ell^t(g) \leq \frac{1}{\eta} \log \left( \exp \left( \sum_{g \in \mathcal{G}} -\eta \sum_{t=1}^{T} \ell^t(g) \right) \right)$ (e.g., Cesa-Bianchi and Lugosi [4]) gives that for any choice $\eta > 0$ and any sequence of loss functions, exponential weights has

$$\mathbf{Reg}_{\mathsf{OL}} \leq \sum_{t=1}^{T} \mathbb{E}_{g \sim \mu^t}[\ell^t(g)] + \frac{1}{\eta} \sum_{t=1}^{T} \log \left( \sum_{g \in \mathcal{G}} \mu^t(g) \exp(-\eta \ell^t(g)) \right) + \frac{\log|\mathcal{G}|}{\eta} \tag{20}$$

$$= \frac{1}{\eta} \sum_{t=1}^{T} \log \left( \sum_{g \in \mathcal{G}} \mu^t(g) \exp(-\eta(\ell^t(g) - \mathbb{E}_{g' \sim \mu^t}[\ell^t(g')])) \right) + \frac{\log|\mathcal{G}|}{\eta}. \tag{21}$$

We first prove (18). Using that $\log(x) \leq x - 1$ for $x \geq 0$ and $\exp(-x) \leq 1 - x + \frac{x^2}{2}$ for $x \geq 0$, we have

$$\log \left( \sum_{g \in \mathcal{G}} \mu^t(g) \exp(-\eta \ell^t(g)) \right) \leq -\eta \, \mathbb{E}_{g \sim \mu^t}[\ell^t(g)] + \frac{\eta^2}{2} \mathbb{E}_{g \sim \mu^t} \big[ (\ell^t(g))^2 \big],$$

so that

$$\mathbf{Reg}_{\mathsf{OL}} \leq \frac{\eta}{2} \sum_{t=1}^{T} \mathbb{E}_{g \sim \mu^t} \big[ (\ell^t(g))^2 \big] + \frac{\log|\mathcal{G}|}{\eta}.$$

To prove (19), we use that $\log(x) \leq x - 1$ for $x \geq 0$ and $\exp(-x) \leq 1 - x + 2x^2$ whenever $|x| \leq 1$ to get

$$\log \left( \sum_{g \in \mathcal{G}} \mu^t(g) \exp(-\eta(\ell^t(g) - \mathbb{E}_{g' \sim \mu^t}[\ell^t(g')])) \right) \leq 2\eta^2 \, \mathbb{E}_{g \sim \mu^t} \big[ (\ell^t(g) - \mathbb{E}_{g' \sim \mu^t}[\ell^t(g')])^2 \big],$$

so that

$$\mathbf{Reg}_{\mathsf{OL}} \leq 2\eta \sum_{t=1}^{T} \mathbb{E}_{g \sim \mu^t} \big[ (\ell^t(g) - \mathbb{E}_{g' \sim \mu^t}[\ell^t(g')])^2 \big] + \frac{\log|\mathcal{G}|}{\eta}.$$

$\square$

### C.3 Online Learning with Completeness

In this section, we give guarantees for an online learning algorithm Algorithm 5, which generalizes the two-timescale exponential weights algorithm (Algorithm 4) of Agarwal and Zhang [2]. We describe and analyze the algorithm in a general online learning framework, which abstracts away the core problem solved by Algorithm 4: value function estimation using a Bellman complete value function class.

Let $\mathcal{G}$ be an abstract set of hypotheses. We consider and online learning process parameterized by a positive integer $H$ and $\alpha \in [0, 1]$.

- There are $2H + 1$ outcome spaces, $\{\mathcal{X}_h\}_{h \in [H]}$, $\{\mathcal{Y}_h\}_{h \in [H]}$, and $\mathcal{W}$.

- There are $H$ unknown probability kernels $\{\mathcal{K}_h : \mathcal{X}_h \to \mathcal{Y}_h\}_{h \in [H]}$.

- There are $2H + 1$ known loss functions $\{\ell_{h,1} : \mathcal{X}_h \times \mathcal{G} \to [0, 1]\}_{h \in [H]}$, $\{\ell_{h,2} : \mathcal{Y}_h \times \mathcal{G} \to [0, 1]\}_{h \in H}$, and $l_3 : \mathcal{W} \times \mathcal{G} \to [0, 1]$.

Define $S^0 = \varnothing$. We consider the following process. For $t = 1, \ldots, T$:

- Learner predicts a (randomized) hypothesis $f^t \in \mathcal{G}$.

- Nature reveals $\{x_h^t, y_h^t\}_{h \in [H]}$ and $w^t$. The outcomes $\{x_h^t\}_{h \in [H]}$ can be chosen adaptively, but they are mutually independent given $S^{t-1}$. The outcome $w^t$ can be chosen adaptively based on $S^{t-1}$ and $\{x_h^t\}_{h \in [H]}$. Each outcome $y_h^t \sim \mathcal{K}_h(x_h^t)$ is drawn independently for each $h \in [H]$.

- The history is updated via $S^t \leftarrow S^{t-1} \bigcup \{x_h^t, y_h^t\}_{h \in [H]} \bigcup \{w^t\}$.

- The learner suffers loss

$$\frac{1}{H} \sum_{h=1}^{H} (\ell_{h,1}(x_h^t, f^t) - \mathbb{E}[\ell_{h,2}(y_h, f^t) \mid x_h^t])^2 + \alpha l_3(w^t, f^t). \tag{22}$$

The learner's goal is to minimize a form of regret for the cumulative loss given in (22). This loss function reflects two objective . The first objective involves the $H$ losses $\{\ell_{1,h}\}_{h \in [H]}$ with corresponding outcomes $\{x_h\}_{h \in [H]}$, as well as the $H$ losses $\{\ell_{2,h}\}_{h \in [H]}$ tied to outcomes $\{y_h\}_{h \in [H]}$, which are generated stochastically based on $\{x_h\}_{h \in [H]}$. The primary objective is to minimize the primary error

$$\frac{1}{H} \sum_{t=1}^{T} \sum_{h=1}^{H} (\ell_{h,1}(x_h^t, f^t) - \mathbb{E}[\ell_{h,2}(y_h, f^t) \mid x_h^t])^2$$

The secondary objective is to minimize $\sum_{t=1}^{T} l_3(w^t, f^t)$, and the ultimate goal is to minimize a weighted sum of the two objectives.

This online learning setup, adapted from Agarwal and Zhang [2], generalizes the reinforcement learning setting in which Algorithm 4 operates. The adaptively chosen outcome $x_h^t$ corresponds to the state-action pair at the $h$-th step, $(s_h^t, a_h^t)$, with the policy at time $t$ chosen in an adaptive, potentially adversarial fashion. The conditionally stochastic outcome $y_h^t$ corresponds to the reward and the next state $(r_h^t, s_{h+1}^t)$, which is sampled independently from the MDP's reward distribution and transition distribution at step $h$, and is conditionally independent given $(s_h^t, a_h^t)$. The learner's objective in the RL framework is to predict a value function $f^t = Q^t$ that minimizes the squared Bellman error, realized by selecting the losses $\ell_{1,h}(x_h^t, f^t) = Q_h^t(s_h^t, a_h^t)$ and $\ell_{2,h}(y_h^t, f^t) = r_h^t + \max_a Q_{h+1}^t(s_{h+1}^t, a)$. The secondary objective is to predict the value function optimistically, with $\ell_3(w^t, f^t) = -\max_a Q(s_1^t, a)$ and $w^t = s_1^t$.

---

**Algorithm 5** Two-Timescale Exponential Weights (adapted from Agarwal and Zhang [2])

1: Initialize $S_0 \leftarrow \varnothing$.
2: **for** $t = 1, 2, \ldots, T$ **do**
3:     For all $f, g \in \mathcal{G}$, define

$$\Delta_h^t(g, f) := \ell_{h,1}(x_h^t, g) - \ell_{h,2}(y_h^t, f),$$

$$q^t(g \mid f) := q^t(g \mid f, S^{t-1}) \propto \exp\left(-\lambda \cdot \frac{1}{H} \sum_{s=1}^{t-1} \sum_{h=1}^{H} \Delta_h^s(g, f)^2\right),$$

$$L^t(f) := \frac{1}{H} \sum_{h=1}^{H} \Delta_h^t(f, f)^2 + \frac{1}{\lambda} \log\left(\mathbb{E}_{g \sim q^t(\cdot | f)}\left[\exp\left(-\lambda \cdot \frac{1}{H} \sum_{h=1}^{H} \Delta_h^t(g, f)^2\right)\right]\right),$$

$$p^t(f) := p^t(f \mid S^{t-1}) \propto \exp\left(-\eta \sum_{s=1}^{t-1} (\beta l_3(w^s, f) + L^s(f))\right).$$

4:     Sample and predict $f^t \sim p^t$.
5:     Observe $\{x_h^t, y_h^t\}_{h \in [H]}$, $w^t$. and update $S_t \leftarrow S_{t-1} \bigcup \{x_h^t, y_h^t\}_{h \in [H]} \bigcup \{w^t\}$.

---

To analyze Algorithm 5, we make a generalized realizability assumption and a generalized completeness assumption; these assumptions abstract away the notions of realizability and completeness in RL.

**Assumption C.1** (Realizability for online learning). *There exists $f^* \in \mathcal{G}$ such that for all $h \in [H]$ and $x_h \in \mathcal{X}_h$, we have*

$$\ell_{h,1}(x_h, f^*) = \mathbb{E}[\ell_{h,2}(y_h, f^*) \mid x_h].$$

**Assumption C.2** (Completeness for online learning). *For any $f \in \mathcal{G}$, there exists $g \in \mathcal{G}$ such that for all $h \in [H]$ and $x_h \in \mathcal{X}_h$, we have*

$$\ell_{h,1}(x_h, g) = \mathbb{E}[\ell_{h,2}(y_h, f) \mid x_h].$$

*For any $f \in \mathcal{G}$, we denote the corresponding $g \in \mathcal{G}$ satisfying this property by $g = \mathcal{T}f$.*

For functions $g$ and $f$ and outcome $x_h$, define

$$\mathcal{E}_h(g, f, x_h) := \ell_{h,1}(x_h, g) - \mathbb{E}[\ell_{h,2}(y_h, f) \mid x_h];$$

this quantity generalizes the notion of Bellman error for reinforcement learning. Recalling that $\Delta_h^t(f, g) := \ell_{h,1}(x_h^t, f) - \ell_{h,2}(y_h^t, g)$, it follows immediately

$$\mathbb{E}[\Delta_h^t(g, f) \mid x_h^t] = \mathbb{E}[(\ell_{h,1}(x_h^t, g) - \ell_{h,2}(y_h^t, f)) \mid x_h^t] = \mathcal{E}_h(g, f, x_h^t).$$

In addition, let us define

$$\mathcal{E}(g, f, x_{1:H})^2 := \frac{1}{H} \sum_{h=1}^{H} (\mathcal{E}_h(g, f, x_h))^2,$$

and

$$\iota^t(f) := l_3(w^t, f) - l_3(w^t, f^*).$$

The following result is the main theorem concerning the performance of Algorithm 5.

**Theorem C.1.** *Let $\lambda = 1/8$, $\eta < 1/(2^{16}(\log(|\mathcal{G}|T/\delta) + 1))$, and $0 < \beta < 1$. Under Assumption C.1 and Assumption C.2, for any $\delta \in (0, 1)$, with probability at least $1 - \delta$,*

$$\frac{1}{H} \sum_{t=1}^{T} \sum_{h=1}^{H} \mathbb{E}_{t-1} \left[ \mathbb{E}_{f \sim p^t} \left[ (\ell_{h,1}(x_h^t, f) - \mathbb{E}[\ell_{h,2}(y_h, f) \mid x_h^t])^2 + 12\beta l_3(w^t, f) \right] \mid x_h^t \right]$$

$$- \left( \frac{1}{H} \sum_{t=1}^{T} \sum_{h=1}^{H} (\ell_{h,1}(x_h^t, f^*) - \mathbb{E}[\ell_{h,2}(y_h, f^*) \mid x_h^t])^2 + 12\beta l_3(w^t, f^*) \right)$$

$$= \sum_{t=1}^{T} \mathbb{E}_{t-1} \left[ \mathbb{E}_{f \sim p^t} \left[ \mathcal{E}(f, f, x_{1:H}^t)^2 + 12\beta \iota^t(f) \right] \mid x_{1:H}^t \right]$$

$$\leq 2^{16} (\eta\beta \log(|\mathcal{G}|T/\delta)T + \log(|\mathcal{G}|)/\eta + \beta^2 T).$$

### C.3.1 Proof of Theorem C.1

For our analysis, it will be useful to consider the following offset version of the loss:

$$\delta_h^t(g, f) := \Delta_h^t(g, f)^2 - (\mathbb{E}[\ell_{h,2}(y_h, f) \mid x_h^t] - \ell_{h,2}(y_h^t, f))^2.$$

We use $x_{1:H}$ as a shorthand for $\{x_h\}_{h \in [H]}$ and further define

$$\delta^t(g, f) := \frac{1}{H} \sum_{h=1}^{H} \delta_h^t(g, f),$$

$$\delta^t(f) := \mathbb{E}_{g \sim q^t(\cdot|f)} \delta^t(g, f),$$

$$Z^t(f) := -\frac{1}{\lambda} \log \mathbb{E}_{g \sim q^t(\cdot|f)} \exp(-\lambda\delta^t(g, f)),$$

$$Z^t = -\frac{1}{\eta} \log \mathbb{E}_{f \sim p^t} \exp\left(-\eta \cdot [\beta\iota^t(f) + \delta^t(f, f) - Z^t(f)]\right).$$

Recall from Algorithm 5 that

$$q^t(g \mid f) := q^t(g \mid f, S^{t-1}) \propto \exp\left(-\lambda \cdot \frac{1}{H} \sum_{s=1}^{t-1} \sum_{h=1}^{H} \Delta_h^s(g, f)^2\right),$$

$$p^t(f) := p^t(f \mid S^{t-1}) \propto \exp\left(-\eta \sum_{s=1}^{t-1} (\beta l_3(w^s, f) + L^s(f))\right).$$

Thus, we can verify the following relationships:

$$q^t(g \mid f) = \frac{\exp\left(-\lambda \sum_{s=1}^{t-1} \delta^s(g, f)\right)}{\sum_{g' \in \mathcal{G}} \exp\left(-\lambda \sum_{s=1}^{t-1} \delta^s(g', f)\right)},$$

$$p^t(f) = \frac{\exp\left(-\eta \sum_{s=1}^{t-1} [\beta\iota^s(f) + \delta^s(f, f) - Z^s(f)]\right)}{\sum_{f' \in \mathcal{G}} \exp\left(-\eta \sum_{s=1}^{t-1} [\beta\iota^s(f') + \delta^s(f', f') - Z^s(f')]\right)}, \tag{23}$$

$$q^{t+1}(g \mid f) = q^t(g \mid f) \cdot e^{-\lambda[\delta^t(g,f) - Z^t(f)]},$$

$$p^{t+1}(f) = p^t(f) \cdot e^{-\eta[\beta\iota^t(f) + \delta^t(f,f) - Z^t(f) - Z^t]}.$$

In what follows, we use $\mathbb{E}_{t-1}[\cdot]$ to abbreviate $\mathbb{E}[\cdot \mid \mathfrak{F}^{t-1}]$, where $\mathfrak{F}^{t-1} := \sigma(S^{t-1})$.

**Proof of Theorem C.1.** Under Assumption C.1, $(\ell_{h,1}(x_h^t, f^*) - \mathbb{E}[\ell_{h,2}(y_h, f^*) \mid x_h^t])^2 = 0$. Thus, the first equality holds by definition as

$$\frac{1}{H} \sum_{t=1}^{T} \sum_{h=1}^{H} \mathbb{E}_{t-1}\left[\mathbb{E}_{f \sim p^t}\left[(\ell_{h,1}(x_h^t, f) - \mathbb{E}[\ell_{h,2}(y_h, f) \mid x_h^t])^2 + 12\beta l_3(w^t, f)\right] \mid x_h^t\right]$$

$$- \left(\frac{1}{H} \sum_{t=1}^{T} \sum_{h=1}^{H} (\ell_{h,1}(x_h^t, f^*) - \mathbb{E}[\ell_{h,2}(y_h, f^*) \mid x_h^t])^2 + 12\beta l_3(w^t, f^*)\right)$$

$$= \sum_{t=1}^{T} \mathbb{E}_{t-1}\left[\mathbb{E}_{f \sim p^t}\left(\mathcal{E}(f, f, x_{1:H}^t)^2 + 12\beta\iota^t(f)\right) \mid x_{1:H}^t\right].$$

To prove the result, we appeal to two technical lemmas, Lemma C.4 and Lemma C.5, stated below. Plugging the bound from Lemma C.5 into Lemma C.4, we have

$$\sum_{t=1}^{T} \mathbb{E}_{t-1}\left[\mathbb{E}_{f \sim p^t} \mathcal{E}(f, f, x_{1:H}^t)^2 \mid x_{1:H}^t\right] + 6\beta \sum_{t=1}^{T} \mathbb{E}_{t-1}[\mathbb{E}_{f \sim p^t} \iota(f) \mid x_{1:H}^t]$$

$$\leq 64\left(\frac{1}{128} \sum_{t=1}^{T} \mathbb{E}_{t-1}\left[\mathbb{E}_{f \sim p^t}\left[\mathcal{E}(f, f, x_{1:H}^t)^2\right] \mid x_{1:H}^t\right] + (48\eta^2\beta^2 + 8\eta\beta)\log(|\mathcal{G}|T/\delta)T + 8\eta\beta^2 T + 16\log|\mathcal{G}|\right)$$

$$+ \frac{6}{\eta}\log|\mathcal{G}| + 18\beta^2 T.$$

Rearranging, we obtain

$$\sum_{t=1}^{T} \mathbb{E}_{t-1}\left[\mathbb{E}_{f \sim p^t}\left(\mathcal{E}(f, f, x_{1:H}^t)^2 + 12\beta\iota^t(f)\right) \mid x_{1:H}^t\right] \leq 2^{16}(\eta\beta\log(|\mathcal{G}|T/\delta)T + \log(|\mathcal{G}|)/\eta + \beta^2 T).$$

$\square$

We now state the technical lemmas, Lemma C.4 and Lemma C.5, used in the proof above

**Lemma C.4.** *Under Assumption C.1, for any $0 \leq \eta \leq 1/24$ and $0 < \beta < 1$, we have*

$$\sum_{t=1}^{T} \mathbb{E}_{t-1}\big[\mathbb{E}_{f\sim p^t}\, \mathcal{E}(f,f,x_{1:H}^t)^2 \mid x_{1:H}^t\big] + 6\beta \sum_{t=1}^{T} \mathbb{E}_{t-1}\big[\mathbb{E}_{f\sim p^t}\, \iota^t(f) \mid x_{1:H}^t\big]$$

$$\leq 64 \sum_{t=1}^{T} \mathbb{E}_{t-1}[\mathbb{E}_{f\sim p^t}\, Z^t(f) \mid x_{1:H}^t] + 18\eta\beta^2 T + \frac{6}{\eta}\log|\mathcal{G}|.$$

**Lemma C.5.** *Let $\lambda = 1/8$ and $\eta < 1/(2^{16}(\log(|\mathcal{G}|T/\delta)+1))$. Under Assumption C.2, for any $0 < \delta < 1$, with probability at least $1 - \delta$, we have*

$$\sum_{t=1}^{T} \mathbb{E}_{t-1}[\mathbb{E}_{f\sim p^t}\, Z^t(f) \mid x_{1:H}^t] \leq \frac{1}{128} \sum_{t=1}^{T} \mathbb{E}_{t-1}\big[\mathbb{E}_{f\sim p^t}\, \mathcal{E}(f,f,x_{1:H}^t)^2 \mid x_{1:H}^t\big]$$

$$+ (48\eta^2\beta^2 + 8\eta\beta)\log(|\mathcal{G}|T/\delta)T$$

$$+ 8\eta\beta^2 T + 16\log|\mathcal{G}|.$$

The remainder of the proof is organized as follows.

- Appendix C.3.2 presents basic technical lemmas.

- Appendix C.3.3 presents the proof of Lemma C.4.

- Appendix C.3.4 presents the proof of Lemma C.5.

- Appendices C.3.6 and C.3.7 contain additional technical lemmas used in the proof of Lemma C.5.

### C.3.2 Basic properties

In this section, we present basic technical results that will be used within the proof of Lemma C.5 and Lemma C.4.

In this section, we will present some basic properties of $\delta^t(g,f)$, $\mathcal{E}(g,f,x_{1:H}^t)^2$ and $Z^t(f)$ for any $t \in [T], g, f \in \mathcal{G}, x_{1:H}^t \in \bigcup_{h\in[H]} \mathcal{X}_h$. These properties are mainly due to a sub-Gaussian term in the definition of $\delta^t(g,f)$. Recall

$$\delta^t(g,f) = \frac{1}{H}\sum_{h=1}^{H}(\ell_{h,1}(x_h^t,g) - \ell_{h,2}(y_h^t,f))^2 - (\mathbb{E}[\ell_{h,2}(y_h,f) \mid x_h^t] - \ell_{h,2}(y_h^t,f))^2$$

$$= \frac{1}{H}\sum_{h=1}^{H}((\ell_{h,1}(x_h^t,g) - \mathbb{E}[\ell_{h,2}(y_h,f) \mid x_h^t])^2$$

$$+ 2(\mathbb{E}[\ell_{h,2}(y_h,f) \mid x_h^t] - \ell_{h,2}(y_h^t,f))(\ell_{h,1}(x_h^t,g) - \mathbb{E}[\ell_{h,2}(y_h,f) \mid x_h^t]))$$

$$= \mathcal{E}(g,f,x_{1:H}^t)^2 + \frac{1}{H}\sum_{h=1}^{H} 2(\mathbb{E}[\ell_{h,2}(y_h,f) \mid x_h^t] - \ell_{h,2}(y_h^t,f))\mathcal{E}_h(g,f,x_h^t).$$

Note that the quantity $\mathbb{E}[\ell_{h,2}(y_h,f) \mid x_h^t] - \ell_{h,2}(y_h^t,f)$, conditioned on $x_h^t$, is a mean 0 random variable bounded by $[-1,1]$. Thus, we have

$$\mathbb{E}[\delta^t(g,f) \mid x_{1:H}^t] = \mathbb{E}\big[\mathcal{E}(g,f,x_{1:H}^t)^2 \mid x_{1:H}^t\big]. \tag{24}$$

Furthermore, since $\mathbb{E}[\ell_{h,2}(y_h,f) \mid x_h^t] - \ell_{h,2}(y_h^t,f)$ is sub-Gaussian with variance proxy $1/2$ by Hoeffding's inequality, we have the following three lemmas.

**Lemma C.6.** *For any $t \leq T$, $g, f \in \mathcal{G}$ and $c \in \mathbb{R}$, we have*

$$\mathbb{E}[\exp(-c\delta^t(g,f)) \mid x_{1:H}^t] \leq \mathbb{E}\big[\exp\big(-c(1-2c)\mathcal{E}(g,f,x_{1:H}^t)^2\big) \mid x_{1:H}^t\big].$$

*In addition, for any $0 < c < \frac{1}{2}$, we have $\mathbb{E}[\exp(-c\delta^t(g,f)) \mid x_{1:H}^t] \leq 1$.*

**Proof of Lemma C.6.** By the $1/2$-sub-Gaussianity of $\mathbb{E}[\ell_{h,2}(y_h, f) \mid x_h^t] - \ell_{h,2}(y_h^t, f)$, we have

$$\mathbb{E}[\exp(-c\delta^t(g,f)) \mid x_{1:H}^t]$$

$$= \mathbb{E}\big[\exp(-c\mathcal{E}(g,f,x_{1:H}^t)^2) \mid x_{1:H}^t\big] \mathbb{E}\left[\exp\left(-\frac{2c}{H}\sum_{h=1}^{H}(\mathbb{E}[\ell_{h,2}(y_h,f) \mid x_h^t] - \ell_{h,2}(y_h^t,f))\mathcal{E}_h(g,f,x_h^t)\right) \mid x_{1:H}^t\right]$$

$$\leq \mathbb{E}\big[\exp\big(-c(1-2c/H)\mathcal{E}(g,f,x_{1:H}^t)^2\big) \mid x_{1:H}^t\big]$$

$$\leq \mathbb{E}\big[\exp\big(-c(1-2c)\mathcal{E}(g,f,x_{1:H}^t)^2\big) \mid x_{1:H}^t\big].$$

$\square$

**Lemma C.7.** *For any $t \leq T$, $g$, $f$ and $x_{1:H}^t$, we have* $\mathbb{E}\big[\delta^t(g,f)^2 \mid x_{1:H}^t\big] \leq 5\,\mathbb{E}\big[\mathcal{E}(g,f,x_{1:H}^t)^2 \mid x_{1:H}^t\big].$

**Proof of Lemma C.7.** The result follows by writing

$$\mathbb{E}\big[\delta^t(g,f)^2 \mid x_{1:H}^t\big]$$

$$= \mathbb{E}\left[\left(\mathcal{E}(g,f,x_{1:H}^t)^2 + \frac{1}{H}\sum_{h=1}^{H}2(\mathbb{E}[\ell_{h,2}(y_h,f) \mid x_h^t] - \ell_{h,2}(y_h^t,f))\mathcal{E}_h(g,f,x_h^t)\right)^2 \mid x_{1:H}^t\right]$$

$$= \mathbb{E}\left[\mathcal{E}(g,f,x_{1:H}^t)^4 + \frac{4}{H^2}\left(\sum_{h=1}^{H}(\mathbb{E}[\ell_{h,2}(y_h,f) \mid x_h^t] - \ell_{h,2}(y_h^t,f))\mathcal{E}_h(g,f,x_h^t)\right)^2 \mid x_{1:H}^t\right]$$

$$\leq \mathbb{E}\left[\mathcal{E}(g,f,x_{1:H}^t)^2 + \frac{4}{H^2}\sum_{h=1}^{H}((\mathbb{E}[\ell_{h,2}(y_h,f) \mid x_h^t] - \ell_{h,2}(y_h^t,f))\mathcal{E}_h(g,f,x_h^t))^2 \mid x_{1:H}^t\right]$$

$$\leq 5\,\mathbb{E}\big[\mathcal{E}(g,f,x_{1:H}^t)^2 \mid x_{1:H}^t\big],$$

where the first equality is by defintion, the second equality is by expanding the terms and notice that the cross terms have zero mean, the first inequality is obtained using the fact that $|\mathcal{E}_h(g,f,x_h)| \leq 1$ and that the cross terms of the expansion of the second term have zero mean and the final inequality is by $|\mathbb{E}[\ell_{h,2}(y_h,f) \mid x_h^t] - \ell_{h,2}(y_h^t,f)| \leq 1$. $\square$

**Lemma C.8.** *For all $t \leq T$, as long as $0 \leq \lambda \leq 1/8$, we have that for all $f$,*

$$4\,\mathbb{E}_{t-1}[Z^t(f) \mid x_{1:H}^t] \geq \mathbb{E}_{g\sim q^t(\cdot|f,S^{t-1})}\big[\mathcal{E}(g,f,x_{1:H}^t)^2\big] \geq 0$$

*almost surely. In particular, $\mathbb{E}_{t-1}[(Z^t(f^*)) \mid x_{1:H}^t] \geq 0$.*

**Proof of Lemma C.8.** Recall $Z^t(f) = -\frac{1}{\lambda}\log\mathbb{E}_{g\sim q^t(\cdot|f)}\exp(-\lambda\delta^t(g,f))$, thus we have

$$-\lambda\,\mathbb{E}_{t-1}[Z^t(f) \mid x_{1:H}^t] = \mathbb{E}_{t-1}[\log\mathbb{E}_{g\sim q^t(\cdot|f)}\exp(-\lambda\delta^t(g,f)) \mid x_{1:H}^t]$$

$$\leq \log\mathbb{E}_{t-1}[\mathbb{E}_{g\sim q^t(\cdot|f)}\exp(-\lambda\delta^t(g,f)) \mid x_{1:H}^t] \qquad\text{(Jensen)}$$

$$\leq \log\mathbb{E}_{t-1}\big[\mathbb{E}_{g\sim q^t(\cdot|f)}\exp\big(-\lambda(1-2\lambda)\mathcal{E}(g,f,x_{1:H}^t)^2\big) \mid x_{1:H}^t\big]$$

$$\qquad\qquad\qquad\qquad\text{(Lemma C.6)}$$

$$\leq \log\mathbb{E}_{t-1}\left[\mathbb{E}_{g\sim q^t(\cdot|f)}\left(1 - \frac{1}{2}\lambda(1-2\lambda)\mathcal{E}(g,f,x_{1:H}^t)^2\right) \mid x_{1:H}^t\right]$$

$$\qquad\qquad\qquad\qquad\text{(Lemma C.1)}$$

$$\leq -\frac{\lambda}{4}\,\mathbb{E}_{t-1}\big[\mathbb{E}_{g\sim q^t(\cdot|f)}\mathcal{E}(g,f,x_{1:H}^t)^2 \mid x_{1:H}^t\big] = -\frac{\lambda}{4}\,\mathbb{E}_{g\sim q^t(\cdot|f)}\mathcal{E}(g,f,x_{1:H}^t)^2.$$

$\square$

Recalling that $Z^t(f) = -\frac{1}{\lambda}\log\mathbb{E}_{g\sim q^t(\cdot|f)}\exp(-\lambda\delta^t(g,f))$ and $\delta^t(f) = \mathbb{E}_{g\sim q^t(\cdot|f)}\delta^t(g,f)$, we obtain the following relationship via Jensen's inequality and second order expansion of the exponential function.

**Lemma C.9.** *For any $t \leq T$ and realization of $S^{t-1}$, we have that as long as $\lambda < 1/8$, $Z^t(f) \leq \delta^t(f)$ and $|Z^t(f)| \leq \frac{9}{8}\mathbb{E}_{g\sim q^t(\cdot|f)}|\delta^t(g,f)|$ for all $g, f \in \mathcal{G}$.*

**Proof of Lemma C.9.** By Jensen's inequality,

$$Z^t(f) = -\frac{1}{\lambda} \log \mathbb{E}_{g \sim q^t(\cdot|f)} \exp(-\lambda \delta^t(g,f)) \leq \mathbb{E}_{g \sim q^t(\cdot|f)} \delta^t(g,f) = \delta^t(f).$$

On the other hand, applying Lemma C.2, we have

$$-\lambda Z^t(f) = \log \mathbb{E}_{g \sim q^t(\cdot|f)} \exp(-\lambda \delta^t(g,f))$$
$$\leq \mathbb{E}_{g \sim q^t(\cdot|f)} \big( -\lambda \delta^t(g,f) + \lambda^2 \delta^t(g,f)^2 \big).$$

Thus

$$|Z^t(f)| \leq |\delta^t(f)| + \lambda \mathbb{E}_{g \sim q^t(\cdot|f)} \delta^t(g,f)^2 \leq \frac{9}{8} \mathbb{E}_{g \sim q^t(\cdot|f)} |\delta^t(g,f)|.$$

$\square$

Another important fact we will use is that $f^*$ satisfies $\delta^t(f^*, f^*) = 0$. Furthermore, for any $f \in \mathcal{G}$, the pair $\mathcal{T}f, f$ always has $\delta^t(\mathcal{T}f, f) = 0$ for all $t \in [T]$.

**Lemma C.10.** *For any $t \leq T$, under Assumption C.1 we have $\delta^t(f^*, f^*) = 0$. In addition, if Assumption C.2 is satisfied, then for any $f$, $\delta^t(\mathcal{T}f, f) = 0$.*

**Proof of Lemma C.10.** By the definition of $f^*$,

$$\delta^t(f^*, f^*) = \frac{1}{H} \sum_{h=1}^{H} (\ell_{h,1}(x_h^t, f^*) - \ell_{h,2}(y_h^t, f^*))^2 - (\mathbb{E}[\ell_{h,2}(y_h, f^*) \mid x_h^t] - \ell_{h,2}(y_h^t, f^*))^2 = 0.$$

Likewise, by the definition of $\mathcal{T}f$,

$$\delta^t(\mathcal{T}f, f) = \frac{1}{H} \sum_{h=1}^{H} (\ell_{h,1}(x_h^t, \mathcal{T}f) - \ell_{h,2}(y_h^t, f))^2 - (\mathbb{E}[\ell_{h,2}(y_h, f) \mid x_h^t] - \ell_{h,2}(y_h^t, f))^2 = 0.$$

$\square$

### C.3.3  Proof of Lemma C.4

In this section, we prove Lemma C.4, which gives a guarantee for the outer exponential weights update used within Algorithm 5.

**Proof of Lemma C.4.** The definition of the exponential weights update (in particular, (23)) implies that

$$-\eta \cdot \min_f \left( \sum_{t=1}^{T} \beta \iota^t(f) + \delta^t(f,f) - Z^t(f) \right) - \log |\mathcal{G}|$$

$$\leq \sum_{t=1}^{T} \log(\mathbb{E}_{f \sim p^t} \exp(-\eta(\beta \iota^t(f) + \delta^t(f,f) - Z^t(f))))$$

$$\leq \frac{1}{3} \sum_{t=1}^{T} \log(\mathbb{E}_{f \sim p^t} \exp(-3\eta \beta \iota^t(f))) + \frac{1}{3} \sum_{t=1}^{T} \log(\mathbb{E}_{f \sim p^t} \exp(-3\eta \delta^t(f,f)))$$

$$+ \frac{1}{3} \sum_{t=1}^{T} \log(\mathbb{E}_{f \sim p^t} \exp(3\eta Z^t(f))),$$

where the final inequality holds due to the fact that $\mathbb{E}[XYZ] \leq \sqrt[3]{\mathbb{E}[X^3]\,\mathbb{E}[Y^3]\,\mathbb{E}[Z^3]}$ for positive random variables $X, Y, Z$ (which in turn can be shown via a repeated application of Hölder's inequality). We further have

$$\mathbb{E}_{t-1}[\log(\mathbb{E}_{f \sim p^t} \exp(-3\eta \delta^t(f,f))) \mid x_{1:H}^t]$$
$$\leq \log(\mathbb{E}_{t-1}[\mathbb{E}_{f \sim p^t} \exp(-3\eta \delta^t(f,f)) \mid x_{1:H}^t]) \hspace{2cm} \text{(Jensen)}$$
$$\leq \log\big(\mathbb{E}_{t-1}\big[\mathbb{E}_{f \sim p^t} \exp(-\eta(1-6\eta)\mathcal{E}(f,f,x_{1:H}^t)^2) \mid x_{1:H}^t\big]\big) \hspace{0.5cm} \text{(Lemma C.6)}$$
$$\leq \log\big(\mathbb{E}_{t-1}\big[\mathbb{E}_{f \sim p^t}(1 - \eta(1-6\eta)\mathcal{E}(f,f,x_{1:H}^t)^2) \mid x_{1:H}^t\big]\big) \hspace{0.5cm} \text{(Lemma C.1)}$$
$$\leq -\frac{\eta}{2} \mathbb{E}_{t-1}\big[\mathbb{E}_{f \sim p^t} \mathcal{E}(f,f,x_{1:H}^t)^2 \mid x_{1:H}^t\big].$$

Meanwhile, using Lemma C.2, we get

$$\log(\mathbb{E}_{t-1}[\mathbb{E}_{f \sim p^t} \exp(-3\eta\beta\iota^t(f)) \mid x^t_{1:H}]) \leq \mathbb{E}_{t-1}\big[\mathbb{E}_{f \sim p^t}(-3\eta\beta\iota^t(f) + 9\eta^2\beta^2(\iota^t(f))^2) \mid x^t_{1:H}\big].$$

Moreover, under Assumption C.1, the benchmark term is negative by

$$\mathbb{E}_{t-1}\left[\min_f\left(\sum_{t=1}^T \delta^t(f,f) - Z^t(f) + \beta\iota^t(f)\right) \mid x^t_{1:H}\right]$$

$$\leq \mathbb{E}_{t-1}\left[\sum_{t=1}^T \delta^t(f^*,f^*) - Z^t(f^*) + \beta\iota^t(f^*) \mid x^t_{1:H}\right]$$

$$= -\sum_{t=1}^T \mathbb{E}_{t-1}[Z^t(f^*) \mid x^t_{1:H}] \quad \text{(Lemma C.10 and the definition of } \iota^t)$$

$$\leq 0. \quad \text{(Lemma C.8)}$$

Furthermore, the log-exponential term is—up to a constant—bounded by its first order expansion:

$$\sum_{t=1}^T \log(\mathbb{E}_{t-1}[\mathbb{E}_{f \sim p_t} \exp(3\eta Z^t(f)) \mid x^t_{1:H}])$$

$$\leq \sum_{t=1}^T \log(\mathbb{E}_{t-1}[\mathbb{E}_{f \sim p_t} \exp(3\eta\delta^t(f)) \mid x^t_{1:H}]) \quad \text{(Lemma C.9)}$$

$$= \sum_{t=1}^T \log\big(\mathbb{E}_{t-1}\big[\mathbb{E}_{f \sim p^t} \exp\big(3\eta\,\mathbb{E}_{g \sim q^t(\cdot|f)}\,\delta^t(g,f)\big) \mid x^t_{1:H}\big]\big)$$

$$\leq \sum_{t=1}^T \log\big(\mathbb{E}_{t-1}\big[\mathbb{E}_{f \sim p^t} \mathbb{E}_{g \sim q^t(\cdot|f)} \exp(3\eta\delta^t(g,f)) \mid x^t_{1:H}\big]\big) \quad \text{(by Jensen)}$$

$$\leq \sum_{t=1}^T \log\big(\mathbb{E}_{t-1}\big[\mathbb{E}_{f \sim p^t} \mathbb{E}_{g \sim q^t(\cdot|f)} \exp\big(3\eta(1+6\eta)\mathcal{E}(g,f,x^t_{1:H})^2\big) \mid x^t_{1:H}\big]\big)$$

$$\text{(Lemma C.6)}$$

$$\leq \sum_{t=1}^T \mathbb{E}_{t-1}\big[\mathbb{E}_{f \sim p^t} \mathbb{E}_{g \sim q^t(\cdot|f)} 6\eta(1+6\eta)\mathcal{E}(g,f,x^t_{1:H})^2 \mid x^t_{1:H}\big] \quad \text{(Lemma C.1)}$$

$$\leq 32\eta \sum_{t=1}^T \mathbb{E}_{t-1}[\mathbb{E}_{f \sim p^t} Z^t(f) \mid x^t_{1:H}]. \quad \text{(Lemma C.8)}$$

By using the fact that $|\iota^t(f)| \leq 1$ and combining the above displays, we obtain that

$$-\log|\mathcal{G}| \leq -\frac{\eta}{6}\,\mathbb{E}_{t-1}\left[\mathbb{E}_{f \sim p^t}\,\mathcal{E}(f,f,x^t_{1:H})^2 \mid x^t_{1:H}\right]$$

$$+ \frac{32\eta}{3}\sum_{t=1}^T \mathbb{E}_{t-1}[\mathbb{E}_{f \sim p^t} Z^t(f) \mid x^t_{1:H}]$$

$$+ 3\eta^2\beta^2 - \mathbb{E}_{t-1}[\mathbb{E}_{f \sim p^t}[\eta\beta\iota^t(f)]].$$

Rearranging yields the desired statement. □

### C.3.4 Proof of Lemma C.5

We state three technical lemmas, Lemmas C.11 to C.13, which are proven in subsequent subsections, then prove Lemma C.5 as a consequence.

**Lemma C.11.** *Almost surely with respect to the draw of $S^T$, we have*

$$\sum_{t=1}^{T} \mathbb{E}_{t-1}\big[\mathbb{E}_{f\sim p^t(\cdot|S^{t-1})}\, Z^t(f) \mid x_{1:H}^t\big]$$

$$\leq \frac{1}{\lambda}\underbrace{\sum_{t=1}^{T}\mathbb{E}_{t-1}\bigg[\mathbb{E}_{p^t}\bigg[\Big(e^{-\eta[\beta\iota^t(f)+\delta^t(f,f)-Z^t(f)-Z^t]}-1\Big)\log\frac{1}{q^t(\mathcal{T}f\mid f)}\bigg]\mid x_{1:H}^t\bigg]}_{\text{Term I}}$$

$$-\underbrace{\sum_{t=1}^{T}\mathbb{E}_{t-1}\bigg[\mathbb{E}_{p^t}\bigg[\Big(e^{-\eta[\beta\iota^t(f)+\delta^t(f,f)-Z^t(f)-Z^t]}-1\Big)Z^t(f)\bigg]\mid x_{1:H}^t\bigg]}_{\text{Term II}}+\frac{1}{\lambda}\log|\mathcal{G}|.$$

**Lemma C.12.** *Under Assumption C.2, for any $0 < \delta < 1$, with probability at least $1-\delta$ over the draw of $S^T$, we have for all $t \leq T$,*

$$\mathbb{E}_{t-1}\bigg[\mathbb{E}_{f\sim p^t}\Big(e^{-\eta[\beta\iota^t(f)+\delta^t(f,f)-Z^t(f)-Z^t]}-1\Big)\log\frac{1}{q^t(\mathcal{T}f\mid f)}\mid x_{1:H}^t\bigg]$$

$$\leq 32\eta\,\mathbb{E}_{t-1}\big[\mathbb{E}_{f\sim p^t}\big[\big(Z^t(f)+\mathcal{E}(f,f,x_{1:H}^t)^2\big)\log(|\mathcal{G}|T/\delta)\big]\mid x_{1:H}^t\big]$$
$$+(24\eta^2\beta^2+4\eta\beta)\log(|\mathcal{G}|T/\delta).$$

**Lemma C.13.** *For any $t \leq T$, the following bound holds almost surely with respect to the draw of $S^t$:*

$$\mathbb{E}_{t-1}\bigg[\mathbb{E}_{f\sim p^t}\bigg[\Big(e^{-\eta[\beta\iota^t(f)+\delta^t(f,f)-Z^t(f)-Z^t]}-1\Big)Z^t(f)\bigg]\mid x_{1:H}^t\bigg]$$

$$\leq \eta\,\mathbb{E}_{t-1}\big[\mathbb{E}_{f\sim p^t}\big[15\mathcal{E}(f,f,x_{1:H}^t)^2+280Z^t(f)\big]\mid x_{1:H}^t\big]+4\eta\beta^2.$$

**Proof of Lemma C.5.** By Lemma C.11, we have

$$\sum_{t=1}^{T}\mathbb{E}_{t-1}[\mathbb{E}_{f\sim p^t}\, Z^t(f)\mid x_{1:H}^t]\leq \frac{1}{\lambda}\cdot(\text{Term I})-(\text{Term II})+\frac{1}{\lambda}\log|\mathcal{G}|.$$

Thus under Assumption C.2, using the bound in Lemma C.12 for Term I and using the bound in Lemma C.13 for Term II, we have with probability at least $1-\delta$,

$$\sum_{t=1}^{T}\mathbb{E}_{t-1}[\mathbb{E}_{f\sim p^t}\, Z^t(f)\mid x_{1:H}^t]$$

$$\leq \frac{1}{\lambda}\cdot\sum_{t=1}^{T}32\eta\,\mathbb{E}_{t-1}\big[\mathbb{E}_{f\sim p^t}\big[\big(Z^t(f)+\mathcal{E}(f,f,x_{1:H}^t)^2\big)\log(|\mathcal{G}|T/\delta)\big]\mid x_{1:H}^t\big]$$

$$+(24\eta^2\beta^2+4\eta\beta)\log(|\mathcal{G}|T/\delta)T$$

$$+\eta\sum_{t=1}^{T}\mathbb{E}_{t-1}\big[\mathbb{E}_{f\sim p^t}\big[15|\mathcal{E}(f,f,x_{1:H}^t)|^2+280Z^t(f)\big]\mid x_{1:H}^t\big]$$

$$+4\eta\beta^2T+\frac{1}{\lambda}\log|\mathcal{G}|.$$

Reorganizing the terms, we have

$$\bigg(1-\frac{32\eta}{\lambda}\log(|\mathcal{G}|T/\delta)-280\eta\bigg)\sum_{t=1}^{T}\mathbb{E}_{t-1}[\mathbb{E}_{f\sim p^t}\, Z^t(f)\mid x_{1:H}^t]$$

$$\leq\bigg(\frac{32\eta}{\lambda}\log(|\mathcal{G}|T/\delta)+15\eta\bigg)\sum_{t=1}^{T}\mathbb{E}_{t-1}\big[\mathbb{E}_{f\sim p^t}\,\mathcal{E}(f,f,x_{1:H}^t)^2\mid x_{1:H}^t\big]$$

$$+(24\eta^2\beta^2+4\eta\beta)\log(|\mathcal{G}|T/\delta)T+4\eta\beta^2T+\frac{1}{\lambda}\log|\mathcal{G}|.$$

Using the assumed bound on $\eta$ and $\lambda$ gives

$$\frac{1}{2}\sum_{t=1}^{T}\mathbb{E}_{t-1}[\mathbb{E}_{f\sim p^t}\,Z^t(f)\mid x_{1:H}^t]\leq\frac{1}{2^8}\sum_{t=1}^{T}\mathbb{E}_{t-1}\left[\mathbb{E}_{f\sim p^t}\left[\mathcal{E}(f,f,x_{1:H}^t)^2\right]\mid x_{1:H}^t\right]$$
$$+(24\eta^2\beta^2+4\eta\beta)\log(|\mathcal{G}|T/\delta)T+4\eta\beta^2 T+8\log|\mathcal{G}|.$$

$\square$

### C.3.5 Proof of Lemma C.11 (Error Decomposition)

**Proof of Lemma C.11.** Recalling the relationships $q^{t+1}(g\mid f)=q^t(g\mid f)\cdot e^{-\lambda[\delta^t(g,f)-Z^t(f)]}$ and $p^{t+1}(f)=p^t(f)\cdot e^{-\eta[\beta\iota^t(f)+\delta^t(f,f)-Z^t(f)-Z^t]}$, the proof begins with the following manipulation:

$$\mathbb{E}_{f\sim p^{t+1}(\cdot|S^t)}\left[\log\frac{1}{q^{t+1}(\mathcal{T}f\mid f)}\right]-\mathbb{E}_{f\sim p^t(\cdot|S^{t-1})}\left[\log\frac{1}{q^t(\mathcal{T}f\mid f)}\right]$$
$$=\mathbb{E}_{p^{t+1}-p_t}\left[\log\frac{1}{q^t(\mathcal{T}f\mid f)}\right]+\mathbb{E}_{p^{t+1}-p_t}\left[\log\frac{q^t(\mathcal{T}f\mid f)}{q^{t+1}(\mathcal{T}f\mid f)}\right]$$
$$+\mathbb{E}_{p_t}\left[\log\frac{q^t(\mathcal{T}f\mid f)}{q^{t+1}(\mathcal{T}f\mid f)}\right]$$
$$=\mathbb{E}_{p^t}\left[\left(e^{-\eta[\beta\iota^t(f)+\delta^t(f,f)-Z^t(f)-Z^t]}-1\right)\log\frac{1}{q^t(\mathcal{T}f\mid f)}\right]$$
$$+\mathbb{E}_{p_t}\left[\left(e^{-\eta[\beta\iota^t(f)+\delta^t(f,f)-Z^t(f)-Z^t]}-1\right)\cdot\lambda\cdot(\delta^t(\mathcal{T}f,f)-Z^t(f))\right]$$
$$+\lambda\cdot\mathbb{E}_{p_t}[\delta^t(\mathcal{T}f,f)-Z^t(f)].$$

Thus, taking expectation with respect to $\mathbb{E}_{t-1}[\cdot]$, dividing by $\lambda$, rearranging, summing over $t$ on both sides, and taking advatnage of telescoping, we obtain

$$\sum_{t=1}^{T}\mathbb{E}_{t-1}\left[\mathbb{E}_{f\sim p^t(\cdot|S^{t-1})}\,Z^t(f)\mid x_{1:H}^t\right]$$
$$\leq\frac{1}{\lambda}\sum_{t=1}^{T}\mathbb{E}_{t-1}\left[\mathbb{E}_{p^t}\left[\left(e^{-\eta[\beta\iota^t(f)+\delta^t(f,f)-Z^t(f)-Z^t]}-1\right)\log\frac{1}{q^t(\mathcal{T}f\mid f)}\right]\mid x_{1:H}^t\right]$$
$$+\sum_{t=1}^{T}\mathbb{E}_{t-1}\left[\mathbb{E}_{p^t}\left[\left(e^{-\eta[\beta\iota^t(f)+\delta^t(f,f)-Z^t(f)-Z^t]}-1\right)\cdot(\delta^t(\mathcal{T}f,f)-Z^t(f))\right]\mid x_{1:H}^t\right]$$
$$+\sum_{t=1}^{T}\mathbb{E}_{t-1}[\mathbb{E}_{p_t}[\delta^t(\mathcal{T}f,f)]\mid x_{1:H}^t]$$
$$+\frac{1}{\lambda}\left(\mathbb{E}_0\left[\mathbb{E}_{f\sim p^1}\log\frac{1}{q_1(\mathcal{T}f\mid f)}\mid x_{1:H}^1\right]-\mathbb{E}_T\left[\mathbb{E}_{f\sim p^{T+1}}\log\frac{1}{q_{T+1}(\mathcal{T}f\mid f)}\mid x_{1:H}^{T+1}\right]\right).$$

Finally, we use that $\delta^t(\mathcal{T}f,f)=0$, $\log\frac{1}{q_{T+1}(\mathcal{T}f|f)}\geq 0$, and $\log\frac{1}{q_1(\mathcal{T}f|f)}=\log|\mathcal{G}|$.

$\square$

### C.3.6 Proof of Lemma C.12 (Bound on Term I)

Toward proving Lemma C.12, we state and prove a series of technical lemmas, Lemmas C.14 to C.17.

**Lemma C.14.** *Under Assumption C.2, for any $0<\delta<1$, the event $A^t$ defined below holds simultaneously for all $t\leq T$ with probability at least $1-\delta$ over the draw of $S^T$:*

$$A^t=\left\{\sup_{f\in\mathcal{G}}\,\log\frac{1}{q^{t+1}(\mathcal{T}f\mid f)}\leq 2\log(|\mathcal{G}|T/\delta)\right\}.$$

**Proof of Lemma C.14.** Recalling that $q^{t+1}(g \mid f) = \dfrac{\exp\left(-\lambda \sum_{s=1}^{t} \delta^s(g,f)\right)}{\sum_{g' \in \mathcal{G}} \exp\left(-\lambda \sum_{s=1}^{t} \delta^s(g',f)\right)}$, we have

$$\log\left(\mathbb{E}_{S^t}\left[\sup_f \frac{1}{q^{t+1}(\mathcal{T}f \mid f)}\right]\right) = \log\left(\mathbb{E}_{S^t}\left[\sup_f \sum_g \exp\left(-\lambda \sum_{s=1}^{t} \delta^s(g,f)\right)\right]\right) \quad \text{(Lemma C.10)}$$

$$\leq \log\left(\mathbb{E}_{S^t}\left[\sum_{f,g} \exp\left(-\lambda \sum_{s=1}^{t} \delta^s(g,f)\right)\right]\right)$$

$$= \log\left(\sum_{f,g} \mathbb{E}_{S^t}\left[\exp\left(-\lambda \sum_{s=1}^{t} \delta^s(g,f)\right)\right]\right)$$

$$\leq 2\log|\mathcal{G}|, \quad \text{(Lemma C.6)}$$

where the final inequality uses $0 < \lambda < 1/2$ as well as the fact that for any fixed choice of $f, g$, conditioned on $x_{1:H}^s$, $s \leq t$, the random variables $\delta^s(g, f)$ (for $s \leq t$) are independent. Then, by Markov's inequality and the union bound, we have the desired result. $\qquad \square$

**Lemma C.15.** *For any $t \leq T$, $0 \leq \lambda \leq 1/8$, $f$, almost surely with respect to the draw of $S^t$ and $x_{1:H}^t$, we have*

$$\mathbb{E}_{t-1}\left[|Z^t(f)|^2 \mid x_{1:H}^t\right] \leq 40 \cdot \mathbb{E}_{t-1}[Z^t(f) \mid x_{1:H}^t].$$

**Proof of Lemma C.15.** Using Lemma C.9, we get

$$\mathbb{E}_{t-1}\left[|Z^t(f)|^2 \mid x_{1:H}^t\right] \leq 2\,\mathbb{E}_{t-1}\left[\left(\mathbb{E}_{g \sim q^t(\cdot|f)}|\delta^t(g,f)|\right)^2 \mid x_{1:H}^t\right]$$

$$\leq 2\,\mathbb{E}_{t-1}\left[\mathbb{E}_{g \sim q^t(\cdot|f)}|\delta^t(g,f)|^2 \mid x_{1:H}^t\right] \quad \text{(by Jensen)}$$

$$\leq 10\,\mathbb{E}_{t-1}\left[\mathbb{E}_{g \sim q^t(\cdot|f)}\,\mathcal{E}(g,f,x_{1:H}^t)^2 \mid x_{1:H}^t\right] \quad \text{(Lemma C.7)}$$

$$\leq 40\,\mathbb{E}_{t-1}[Z^t(f) \mid x_{1:H}^t]. \quad \text{(Lemma C.8)}$$

$\qquad \square$

**Lemma C.16.** *For any $t \leq T$, almost surely with respect to the draw of $S^t$ and $x_{1:H}^t$, we have*

$$\mathbb{E}_{t-1}\left[|Z^t|^2 \mid x_{1:H}^t\right] \leq \mathbb{E}_{t-1}\left[\mathbb{E}_{f \sim p^t(\cdot|S^{t-1})}[10\mathcal{E}(f,f,x_{1:H}^t)^2 + 80Z^t(f)] \mid x_{1:H}^t\right] + 3\beta^2.$$

**Proof of Lemma C.16.** Using Jensen's inequality, we have

$$|Z^t| = \left|-\frac{1}{\eta}\log(\mathbb{E}_{f \sim p^t}\exp(-\eta[\beta\iota^t(f) + \delta^t(f,f) - Z^t(f)]))\right|$$

$$\leq \mathbb{E}_{f \sim p^t}[|\beta\iota^t(f)| + |\delta^t(f,f)| + |Z^t(f)|].$$

Thus by Lemma C.7 and Lemma C.15, we have

$$\mathbb{E}_{t-1}\left[|Z^t|^2 \mid x_{1:H}^t\right] \leq 3\,\mathbb{E}_{t-1}\left[\mathbb{E}_{f \sim p^t}\left[|\beta\iota^t(f)|^2 + |\delta^t(f,f)|^2 + |Z^t(f)|^2\right] \mid x_{1:H}^t\right]$$

$$\leq \mathbb{E}_{t-1}\left[\mathbb{E}_{f \sim p^t}\left[10\mathcal{E}(f,f,x_{1:H}^t)^2 + 80Z^t(f)\right] \mid x_{1:H}^t\right] + 3\beta^2.$$

$\qquad \square$

**Lemma C.17.** *For any $0 \leq \eta \leq 1/240$, $f$, almost surely with respect to the draw of $S^t$ and $x_{1:H}^t$, we have*

$$\mathbb{E}_{t-1}\left[e^{-\eta[\beta\iota^t(f) + \delta^t(f,f) - Z^t(f) - Z^t]} - 1 \mid x_{1:H}^t\right]$$

$$\leq \mathbb{E}_{t-1}\left[16\eta Z^t(f) + 4\eta\,\mathbb{E}_{f' \sim p^t(\cdot|S^{t-1})}\,\mathcal{E}(f',f,x_{1:H}^t)^2 \mid x_{1:H}^t\right]$$

$$+ 12\eta^2\beta^2 + 2\eta\beta.$$

**Proof of Lemma C.17.** Jensen's inequality gives that

$$Z^t \leq \mathbb{E}_{f' \sim p^t} \left[ \beta \iota^t(f') + \delta^t(f', f') - Z^t(f') \right]$$
$$Z^t(f) \leq \mathbb{E}_{g \sim q^t(\cdot | f)} \left[ \delta^t(g, f) \right] = \delta^t(f).$$

We may then write

$$\mathbb{E}_{t-1} \left[ e^{-\eta \left[ \beta \iota^t(f) + \delta^t(f,f) - Z^t(f) - Z^t \right]} - 1 \mid x_{1:H}^t \right]$$

$$\leq \mathbb{E}_{t-1} \left[ e^{-\eta \beta \iota^t(f) - \eta \delta^t(f,f) + \eta \delta^t(f) + \eta \mathbb{E}_{f' \sim p^t} [\beta \iota^t(f') + \delta^t(f',f') - Z^t(f')]} - 1 \mid x_{1:H}^t \right]$$

$$\leq \frac{1}{6} \mathbb{E}_{t-1} \left[ \mathbb{E}_{f' \sim p^t} \left[ e^{-6\eta \beta \iota^t(f)} + e^{-6\eta \delta^t(f,f)} + e^{6\eta \delta^t(f)} + e^{6\eta \beta \iota^t(f')} + e^{6\eta \delta^t(f',f')} + e^{-6\eta Z^t(f')} - 6 \right] \mid x_{1:H}^t \right],$$

where the second equality uses Jensen's inequality to pull the $\mathbb{E}_{f' \sim p^t}[\cdot]$ outside of the exponential and then the fact that $a_1 \cdots a_6 \leq \frac{1}{6} \sum_{i=1}^{6} a_i^6$ for real numbers $a_1, \ldots, a_6$. We control the six terms on the right hand side separately as follows:

- Term (a): Using Lemma C.2 we get

$$\mathbb{E}_{t-1} \left[ e^{-6\eta \beta \iota^t(f)} \mid x_{1:H}^t \right] \leq 1 + 6\eta\beta + 36\eta^2 \beta^2.$$

- Term (b): Using Lemma C.6 we get

$$\mathbb{E}_{t-1} \left[ e^{-6\eta \delta^t(f,f)} \mid x_{1:H}^t \right] \leq 1.$$

- Term (c):

$$\begin{aligned}
\mathbb{E}_{t-1} \left[ e^{6\eta \delta^t(f)} \mid x_{1:H}^t \right] &= \mathbb{E}_{t-1} \left[ \exp(6\eta \, \mathbb{E}_{g \sim q^t(\cdot|f)} \delta^t(g,f)) \mid x_{1:H}^t \right] \\
&\leq \mathbb{E}_{t-1} \left[ \mathbb{E}_{g \sim q^t(\cdot|f)} \exp(6\eta \delta^t(g,f)) \mid x_{1:H}^t \right] \\
&\leq \mathbb{E}_{t-1} \left[ \mathbb{E}_{g \sim q^t(\cdot|f)} \exp\big(6\eta(1+12\eta)\mathcal{E}(g,f,x_{1:H}^t)^2\big) \mid x_{1:H}^t \right] \\
&\qquad\qquad\qquad\qquad\qquad\qquad\qquad\qquad\qquad\qquad\text{(Lemma C.6)} \\
&\leq 1 + 16\eta \, \mathbb{E}_{t-1} \left[ \mathbb{E}_{g \sim q^t(\cdot|f)} \mathcal{E}(g,f,x_{1:H}^t)^2 \mid x_{1:H}^t \right] \quad \text{(Lemma C.1)} \\
&\leq 1 + 64\eta \, \mathbb{E}_{t-1}[Z^t(f) \mid x_{1:H}^t]. \qquad\qquad\qquad\quad \text{(Lemma C.8)}
\end{aligned}$$

- Term (d): Using Lemma C.2, we have

$$\mathbb{E}_{t-1} \left[ \mathbb{E}_{f' \sim p^t} e^{6\eta \beta \iota^t(f')} \mid x_{1:H}^t \right] \leq 1 + 6\eta\beta + 36\eta^2 \beta^2.$$

- Term (e):

$$\begin{aligned}
\mathbb{E}_{t-1} &\left[ \mathbb{E}_{f' \sim p^t} e^{6\eta \delta^t(f',f')} \mid x_{1:H}^t \right] \\
&\leq \mathbb{E}_{t-1} \left[ \mathbb{E}_{f' \sim p^t} \exp\big(6\eta(1+12\eta)\mathcal{E}(f',f',x_{1:H}^t)^2\big) \mid x_{1:H}^t \right] \quad \text{(Lemma C.6)} \\
&\leq 1 + 16\eta \, \mathbb{E}_{t-1} \left[ \mathbb{E}_{f' \sim p^t} \mathcal{E}(f',f',x_{1:H}^t)^2 \mid x_{1:H}^t \right]. \qquad\qquad \text{(Lemma C.1)}
\end{aligned}$$

- Term (f):

$$\begin{aligned}
\mathbb{E}_{t-1} &\left[ \mathbb{E}_{f' \sim p^t} e^{-6\eta Z^t(f')} \mid x_{1:H}^t \right] \\
&\leq \mathbb{E}_{t-1} \left[ \mathbb{E}_{f' \sim p^t} 1 - 6\eta Z^t(f') + 36\eta^2 (Z^t(f'))^2 \mid x_{1:H}^t \right] \quad \text{(Lemma C.2)} \\
&\leq 1 + (-6\eta + 1440\eta^2) \, \mathbb{E}_{t-1}[\mathbb{E}_{f' \sim p^t} Z^t(f') \mid x_{1:H}^t] \qquad\qquad \text{(Lemma C.15)} \\
&\leq 1. \qquad\qquad\qquad\qquad\qquad\qquad\qquad\qquad\qquad\qquad\qquad \text{(Lemma C.8)}
\end{aligned}$$

$\square$

**Proof of Lemma C.12.** Combining the preceding lemmas, we have with probability at least $1 - \delta$ over the draw of $S^T$, the event $A^t$ defined in Lemma C.14 holds for all $t \leq T$ simultaneously, and thus

$$\mathbb{E}_{t-1}\left[\mathbb{E}_{f \sim p^t}\left(e^{-\eta\left[\beta\iota^t(f)+\delta^t(f,f)-Z^t(f)-Z^t\right]} - 1\right)\log\frac{1}{q^t(\mathcal{T}f \mid f)} \mid x^t_{1:H}\right]$$

$$\leq \mathbb{E}_{f \sim p^t}\left[\left(16\eta\,\mathbb{E}_{t-1}\left[Z^t(f) + \mathbb{E}_{f' \sim p^t}\mathcal{E}(f',f',x^t_{1:H})^2 \mid x^t_{1:H}\right] + 12\eta^2\beta^2 + 2\eta\beta\right)\log\frac{1}{q^t(\mathcal{T}f \mid f)}\right]$$

$$\leq 16\eta\,\mathbb{E}_{t-1}\left[\mathbb{E}_{f \sim p^t}\left[\left(Z^t(f) + \mathcal{E}(f,f,x^t_{1:H})^2\right) \cdot 2\log(|\mathcal{G}|T/\delta)\right] \mid x^t_{1:H}\right]$$
$$+ (24\eta^2\beta^2 + 4\eta\beta)\log(|\mathcal{G}|T/\delta),$$

where the first inequality is derived from Lemma C.17, and the second inequality follows from Lemma C.8 and Lemma C.14. $\qquad\square$

### C.3.7  Proof of Lemma C.13 (Bound on Term II)

**Proof of Lemma C.13.** We compute

$$\mathbb{E}_{t-1}\left[\mathbb{E}_{f \sim p^t}\left[\left(e^{-\eta\left[\beta\iota^t(f)+\delta^t(f,f)-Z^t(f)-Z^t\right]} - 1\right)Z^t(f)\right] \mid x^t_{1:H}\right]$$

$$\leq \mathbb{E}_{t-1}\left[\mathbb{E}_{f \sim p^t}\left[\left(e^{\eta\left(\beta|\iota^t(f)|+|\delta^t(f,f)|+|Z^t(f)|+|Z^t|\right)} - 1\right)|Z^t(f)|\right] \mid x^t_{1:H}\right]$$

$$\leq 2\eta\,\mathbb{E}_{t-1}[\mathbb{E}_{f \sim p^t}[[\beta|\iota^t(f)| + |\delta^t(f,f)| + |Z^t(f)| + |Z^t|]|Z^t(f)|] \mid x^t_{1:H}]$$

$$\leq \eta\,\mathbb{E}_{t-1}\left[\mathbb{E}_{f \sim p^t}\left[|\delta^t(f,f)|^2 + 5|Z^t(f)|^2 + |Z^t|^2\right] \mid x^t_{1:H}\right] + \eta\beta^2$$

$$\leq \eta\,\mathbb{E}_{t-1}\left[\mathbb{E}_{f \sim p^t}\left[15\mathcal{E}(f,f,x^t_{1:H})^2 + 280Z^t(f)\right] \mid x^t_{1:H}\right] + 4\eta\beta^2,$$

where the second inequality applies Lemma C.1 (together with the fact that $\eta \in (0, 1/240)$), the third inequality uses Young's inequality and the fact that $|\iota^t(f)| \leq 1$ for all $f$, and the final inequality applies Lemma C.15 (to bound $\mathbb{E}_{t-1}[|Z^t(f)|^2]$), Lemma C.16 (to bound $\mathbb{E}_{t-1}[|Z^t|^2]$), and Lemma C.7 (to bound $\mathbb{E}_{t-1}[\delta^t(f,f) \mid x^t_{1:H}]$) . $\qquad\square$

## D  Proofs from Section 2 and Section 3

### D.1  Proofs from Section 2

**Proof of Theorem 2.1 and Theorem B.1.** We prove Theorem B.1; Theorem 2.1 is the special case of this result in which $n = 1$. Observe that for Algorithm 3, we have

$$\mathbf{Reg}_{\mathsf{DM}} = n \cdot \sum_{k=1}^{K}\mathbb{E}_{\pi^k \sim p^k}\left[f^{M^\star}(\pi_{M^\star}) - f^{M^\star}(\pi^k)\right].$$

We can rewrite this sum as

$$\sum_{k=1}^{K}\mathbb{E}_{\pi^k \sim p^k}\left[f^{M^\star}(\pi_{M^\star}) - f^{M^\star}(\pi^k)\right]$$

$$= \sum_{k=1}^{K}\mathbb{E}_{\pi^k \sim p^k}\mathbb{E}_{\widehat{\psi}^k \sim \mu^k}\left[f^{\widehat{\psi}^k}(\pi_{\widehat{\psi}^k}) - f^{M^\star}(\pi^k)\right] + \mathbb{E}_{\widehat{\psi}^k \sim \mu^k}\left[\left(f^{M^\star}(\pi_{M^\star}) - f^{\widehat{\psi}^k}(\pi_{\widehat{\psi}^k})\right)\right]$$

$$= \sum_{k=1}^{K}\mathbb{E}_{\pi^k \sim p^k}\mathbb{E}_{\widehat{\psi}^k \sim \mu^k}\left[f^{\widehat{\psi}^k}(\pi_{\widehat{\psi}^k}) - f^{M^\star}(\pi^k) - \gamma \cdot D^{\pi^k}\left(\widehat{\psi}^k \parallel M^\star\right)\right]$$
$$+ \mathbb{E}_{\widehat{\psi}^k \sim \mu^k}\left[\gamma \cdot D^{\pi^k}\left(\widehat{\psi}^k \parallel M^\star\right) + (f^{M^\star}(\pi_{M^\star}) - f^{\widehat{\psi}^k}(\pi_{\widehat{\psi}^k}))\right]$$

$$= \sum_{k=1}^{K}\mathbb{E}_{\pi^k \sim p^k}\mathbb{E}_{\widehat{\psi}^k \sim \mu^k}\left[f^{\widehat{\psi}^k}(\pi_{\widehat{\psi}^k}) - f^{M^\star}(\pi^k) - \gamma \cdot D^{\pi^k}\left(\widehat{\psi}^k \parallel M^\star\right)\right] + \gamma \cdot \mathbf{OptEst}_\gamma^D.$$

For each step $k$, by the choice of $p^k$, we have

$$\mathbb{E}_{\pi^k \sim p^k} \mathbb{E}_{\widehat{\psi}^k \sim \mu^k} \left[ f^{\widehat{\psi}^k}(\pi_{\widehat{\psi}^k}) - f^{M^\star}(\pi^k) - \gamma \cdot D^{\pi^k}\left(\widehat{\psi}^k \parallel M^\star\right) \right]$$

$$\leq \sup_{M \in \mathcal{M}} \mathbb{E}_{\pi^k \sim p^k} \mathbb{E}_{\widehat{\psi}^k \sim \mu^k} \left[ f^{\widehat{\psi}^k}(\pi_{\widehat{\psi}^k}) - f^M(\pi^k) - \gamma \cdot D^{\pi^k}\left(\widehat{\psi}^k \parallel M\right) \right]$$

$$= \inf_{p \in \Delta(\Pi)} \sup_{M \in \mathcal{M}} \mathbb{E}_{\pi \sim p} \mathbb{E}_{\widehat{\psi} \sim \mu^k} \left[ f^{\widehat{\psi}}(\pi_{\widehat{\psi}}) - f^M(\pi) - \gamma \cdot D^{\pi}\left(\widehat{\psi} \parallel M\right) \right]$$

$$= \mathsf{o\text{-}dec}_\gamma^D(\mathcal{M}, \mu^k) \leq \mathsf{o\text{-}dec}_\gamma^D(\mathcal{M}).$$

Finally, we use that probability at least $1 - \delta$, $\mathbf{OptEst}_\gamma^D \leq \mathbf{OptEst}_\gamma^D(K, n, \delta)$.

$\square$

We next prove Proposition 2.1, which gives a bound on

$$\mathbf{OptEst}_\gamma^{\mathsf{bi}} = \sum_{k=1}^{K} \mathbb{E}_{\pi^k \sim p^k} \mathbb{E}_{\widehat{M}^k \sim \mu^k} \left[ \sum_{h=1}^{H} \left( \mathbb{E}^{M^\star, \pi} \left[ \ell_h^{\mathrm{est}}(\widehat{Q}^k; z_h) \right] \right)^2 + \gamma^{-1}(f^{M^\star}(\pi_{M^\star}) - f^{\widehat{M}^t}(\pi_{\widehat{M}^k})) \right].$$

---

**Algorithm 6** Optimistic Estimation for Bilinear Classes

1: **parameters**:
- • Number of rounds $T$, Batch size $n$
- • Learning rate $\eta > 0$.
- • Discrepancy function $\ell_h^{\mathrm{est}}(Q; z_h^{k,l})$ Exploration parameter $\gamma > 0$.

2: Let $K = T/n$ and $B^0 := \varnothing$.

3: **for** $k = 1, 2, \cdots, K$ **do**

4:     Form the randomized estimator $\mu^k$ via $\mu^k(Q) \propto \exp\left(-\eta \sum_{i < k} \ell^i(Q)\right)$, where

$$\ell^i(Q) := \sum_{h=1}^{H} \left( \frac{1}{n} \sum_{l=1}^{n} \ell_h^{\mathrm{est}}(Q; z_h^{i,l}) \right)^2 - \frac{1}{8\gamma} \cdot f^Q(\pi_Q)$$

5:     Receive batch of samples $B^k = \{(\pi^{k,l}, r^{k,l}, o^{k,l})\}_{l=1}^{n}$ where $\pi^{k,l} \sim p^k$ and $(r^{k,l}, o^{k,l}) \sim M^\star(\pi^{k,l})$.

                    // $p^k$ **is the decision distribution produced by E2D.Opt (cf. Algorithm 3).**

6:     Let $z_h^{k,l} = (s_h^{k,l}, a_h^{k,l}, r_h^{k,l}, s_{h+1}^{k,l})$, where we recall $o^{k,l} = (s_1^{k,l}, a_1^{k,1}, r_1^{k,l}), \ldots, (s_H^{k,l}, a_H^{k,1}, r_H^{k,l})$.

---

**Proof of Proposition 2.1.** Throughout this proof, we use the batched estimation notation from Appendix B.1. Let $n$ be fixed, and let $K := T/n$ be the number of epochs. Recall that for each step $k \in [K]$, the estimation oracle is given a batch of examples $B^k = \{(\pi^{k,l}, r^{k,l}, o^{k,l})\}_{l=1}^{n}$ where $\pi^{k,l} \sim p^k$ and $(r^{k,l}, o^{k,l}) \sim M^\star(\pi^{k,l})$. Each observation (trajectory) takes the form $o^{k,l} = (s_1^{k,l}, a_1^{k,l}, r_1^{k,l}), \ldots, (s_H^{k,l}, a_H^{k,l}, r_H^{k,l})$. Throughout the proof, we use the notation

$$\mathcal{E}_h^k(Q) = \mathbb{E}^{M^\star, \pi^k} \left[ \ell_h^{\mathrm{est}}(Q; z_h) \right],$$

and abbreviate $Q^\star = Q^{M^\star, \star}$.

**Estimation algorithm.** Define

$$\widehat{\mathcal{E}}_h^k(Q) = \frac{1}{n} \sum_{l=1}^{n} \ell_h^{\mathrm{est}}(Q; z_h^{k,l}),$$

where $z_h^{k,l} := (s_h^{k,l}, a_h^{k,l}, r_h^{k,l}, s_{h+1}^{k,l})$. Let $\eta > 0$ and $\alpha > 0$ be parameters whose values will be chosen at the end of the proof. Defining

$$\ell^k(Q) := \sum_{h=1}^{H} (\widehat{\mathcal{E}}_h^k(Q))^2 - \alpha f^Q(\pi_Q),$$

[Algorithm 6](#) chooses

$$\mu^k(Q) \propto \exp\left(-\eta \sum_{i<k} \ell^i(Q)\right)$$

as the randomized estimator for epoch $k$.

**Estimation error bound.** Let us abbreviate $L = L_{\mathrm{bi}}(\mathcal{Q}; M^\star)$. Observe that for all $k$, $|\ell^k(Q)| \leq R := HL^2 + \alpha$ almost surely. Hence, [Lemma C.3](#) implies that as long as $\eta \leq 1/2R$, we have

$$\sum_{k=1}^{K} \mathbb{E}_{Q\sim\mu^k}[\ell^k(Q)] - \sum_{k=1}^{K} \ell^k(Q^\star) \leq 4\eta \sum_{k=1}^{K} \mathbb{E}_{Q\sim\mu^k}\left[(\ell^k(Q))^2\right] + \frac{\log|\mathcal{Q}|}{\eta} \qquad (25)$$

For each $k \in [K]$, we have that for all $Q \in \mathcal{Q}$,

$$(\ell^k(Q))^2 \leq 2\left(\sum_{h=1}^{H}(\widehat{\mathcal{E}}_h^k(Q))^2\right)^2 + 2\alpha^2 \leq 2HL^2 \sum_{h=1}^{H}(\widehat{\mathcal{E}}_h^k(Q))^2 + 2\alpha^2.$$

As a result, [(25)](#) implies that

$$\sum_{k=1}^{K}\sum_{h=1}^{H} \mathbb{E}_{Q\sim\mu^k}\left[\left(\widehat{\mathcal{E}}_h^k(Q)\right)^2\right] + \alpha \sum_{k=1}^{K} \mathbb{E}_{Q\sim\mu^k}\left[(f^{M^\star}(\pi_{M^\star}) - f^Q(\pi_Q)\right] \qquad (26)$$

$$\leq \sum_{k=1}^{K}\sum_{h=1}^{H}\left(\widehat{\mathcal{E}}_h^k(Q^\star)\right)^2 + 8\eta HL^2 \sum_{k=1}^{K}\sum_{h=1}^{H} \mathbb{E}_{Q\sim\mu^k}\left[\left(\widehat{\mathcal{E}}_h^k(Q)\right)^2\right] + 8\eta\alpha^2 K + \frac{\log|\mathcal{Q}|}{\eta}. \qquad (27)$$

Whenever $\eta \leq \frac{1}{16HL^2}$, rearranging gives

$$\frac{1}{2}\sum_{k=1}^{K}\sum_{h=1}^{H} \mathbb{E}_{Q\sim\mu^k}\left[\left(\widehat{\mathcal{E}}_h^k(Q)\right)^2\right] + \alpha \sum_{k=1}^{K} \mathbb{E}_{Q\sim\mu^k}\left[(f^{M^\star}(\pi_{M^\star}) - f^Q(\pi_Q)\right] \qquad (28)$$

$$\leq \sum_{k=1}^{K}\sum_{h=1}^{H}\left(\widehat{\mathcal{E}}_h^k(Q^\star)\right)^2 + 8\eta\alpha^2 K + \frac{\log|\mathcal{Q}|}{\eta}. \qquad (29)$$

We now appeal to the following lemma.

**Lemma D.1.** *With probability at least $1 - \delta$, it holds that for all $k \in [K]$, $h \in [H]$, and $Q \in \mathcal{Q}$,*

$$\frac{1}{2}(\mathcal{E}_h^k(Q))^2 - \varepsilon_{\mathrm{conc}}^2(n) \leq (\widehat{\mathcal{E}}_h^k(Q))^2 \leq 2(\mathcal{E}_h^k(Q))^2 + 2\varepsilon_{\mathrm{conc}}^2(n), \qquad (30)$$

*where $\varepsilon_{\mathrm{conc}}(n) := L\sqrt{\frac{2\log(|\mathcal{Q}|KH/\delta)}{n}}$.*

Going forward, we condition on the event in [Lemma D.1](#). Applying the inequality [(30)](#) within [(29)](#), we have

$$\frac{1}{4}\sum_{k=1}^{K}\sum_{h=1}^{H} \mathbb{E}_{Q\sim\mu^k}\left[(\mathcal{E}_h^k(Q))^2\right] + \alpha \sum_{k=1}^{K} \mathbb{E}_{Q\sim\mu^k}\left[(f^{M^\star}(\pi_{M^\star}) - f^Q(\pi_Q)\right] \qquad (31)$$

$$\leq 2\sum_{k=1}^{K}\sum_{h=1}^{H}(\mathcal{E}_h^k(Q^\star))^2 + 8\eta\alpha^2 K + \frac{\log|\mathcal{Q}|}{\eta} + 3HK\varepsilon_{\mathrm{conc}}^2(n)$$

$$= 8\eta\alpha^2 K + \frac{\log|\mathcal{Q}|}{\eta} + 3HK\varepsilon_{\mathrm{conc}}^2(n), \qquad (32)$$

where the last equality uses that $\mathcal{E}_h^k(Q^\star) = 0$ for all $h \in [H]$, and $k \in [K]$, which is a consequence of [Definition 2.1](#). Next, we recall that

$$\sum_{k=1}^{K}\sum_{h=1}^{H} \mathbb{E}_{Q\sim\mu^k}\left[(\mathcal{E}_h^k(Q))^2\right] = \sum_{k=1}^{K} \mathbb{E}_{Q\sim\mu^k}\left[D_{\mathrm{bi}}^{\pi^k}(Q \parallel M^\star)\right].$$

A standard application of Freedman's inequality (c.f Lemma A.3 in Foster et al. [12]) implies that with probability at least $1 - \delta$,

$$\sum_{k=1}^{K} \mathbb{E}_{Q \sim \mu^k} \left[ D_{\mathsf{bi}}^{\pi^k}(Q \parallel M^\star) \right] \geq \frac{1}{2} \sum_{k=1}^{K} \mathbb{E}_{\pi \sim p^k} \mathbb{E}_{Q \sim \mu^k} \left[ D_{\mathsf{bi}}^{\pi^k}(Q \parallel M^\star) \right] - O(HL^2 \log(H/\delta)).$$

Putting everything together, we have that

$$\frac{1}{8} \sum_{k=1}^{K} \mathbb{E}_{\pi^k \sim p^k} \mathbb{E}_{Q \sim \mu^k} \left[ D_{\mathsf{bi}}^{\pi^k}(Q \parallel M^\star) + 8\alpha(f^{M^\star}(\pi_{M^\star}) - f^Q(\pi_Q)) \right]$$

$$\lesssim \eta \alpha^2 K + \frac{\log|\mathcal{Q}|}{\eta} + HK\varepsilon_{\mathrm{conc}}^2(n) + HL^2 \log(H/\delta).$$

Choosing $\alpha = \frac{1}{8\gamma}$, this gives

$$\mathbf{OptEst}_\gamma^{\mathsf{bi}} \lesssim \eta \alpha^2 K + \frac{\log|\mathcal{Q}|}{\eta} + HK\varepsilon_{\mathrm{conc}}^2(n) + HL^2 \log(H/\delta)$$

$$\lesssim \eta \alpha^2 K + \frac{\log|\mathcal{Q}|}{\eta} + HL^2 \log(|\mathcal{Q}|KH/\delta)\left(1 + \frac{K}{n}\right)$$

We choose $\eta = \sqrt{\frac{\log|\mathcal{Q}|}{\alpha^2 K}} \wedge \frac{1}{16R}$, which satisfies the constraints described earlier in the proof, and gives

$$\mathbf{OptEst}_\gamma^{\mathsf{bi}} \lesssim \frac{\sqrt{K \log|\mathcal{Q}|}}{\gamma} + R\log|\mathcal{Q}| + HL^2 \log(|\mathcal{Q}|KH/\delta)\left(1 + \frac{K}{n}\right)$$

$$\lesssim \frac{\sqrt{K \log|\mathcal{Q}|}}{\gamma} + HL^2 \log(|\mathcal{Q}|KH/\delta)\left(1 + \frac{1}{\gamma} + \frac{K}{n}\right).$$

$\square$

**Proof of Lemma D.1.** For any fixed $k \in [K]$ and $Q \in \mathcal{Q}$, Hoeffding's inequality implies that with probability at least $1 - \delta$,

$$\left| \mathcal{E}_h^k(Q) - \widehat{\mathcal{E}}_h^k(Q) \right| \leq L\sqrt{\frac{2\log(1/\delta)}{n}}.$$

By a standard union bound it follows that with probability at least $1 - \delta$, for all $k \in [K]$, $h \in [H]$, and $Q \in \mathcal{Q}$ simultaneously,

$$\left| \mathcal{E}_h^k(Q) - \widehat{\mathcal{E}}_h^k(Q) \right| \leq L\sqrt{\frac{2\log(|\mathcal{Q}|KH/\delta)}{n}} =: \varepsilon_{\mathrm{conc}}(n).$$

Whenever this event occurs, the AM-GM inequality implies that

$$(\mathcal{E}_h^k(Q))^2 \leq 2(\widehat{\mathcal{E}}_h^k(Q))^2 + 2\varepsilon_{\mathrm{conc}}^2(n)$$

and likewise $(\widehat{\mathcal{E}}_h^k(Q))^2 \leq 2(\mathcal{E}_h^k(Q))^2 + 2\varepsilon_{\mathrm{conc}}^2(n)$. $\square$

**Proof of Proposition 2.2.** Let $\mu \in \Delta(\mathcal{Q})$ be fixed. Fix $\alpha \in (0,1)$, and let $\pi_Q^\alpha$ be the randomized policy that—for each $h$—independently plays $\pi_{Q,h}$ with probability $1 - \alpha/H$ and $\pi_{Q,h}^{\mathrm{est}}$ with probability $\alpha/H$. Let $p \in \Delta(\Pi)$ be the distribution induced by sampling $Q \sim \mu$ and playing $\pi_Q^\alpha$. Translated to our notation, Foster et al. [12] (cf. Proof of Theorem 7.1, Eq. (152)) shows that for all MDPs $M$, this strategy guarantees that for all $\eta > 0$,

$$\mathbb{E}_{\pi \sim p} \mathbb{E}_{Q \sim \mu} \left[ f^Q(\pi_Q) - f^M(\pi) \right] \leq \alpha + \frac{H \cdot d_{\mathsf{bi}}(\mathcal{Q}; M)}{2\eta} + \frac{\eta}{2} \sum_{h=1}^{H} \mathbb{E}_{Q,Q' \sim \mu} \left[ \langle X_h(Q; M), W_h(Q'; M) \rangle^2 \right]$$

$$= \alpha + \frac{H \cdot d_{\mathsf{bi}}(\mathcal{Q}; M)}{2\eta} + \frac{\eta}{2} \sum_{h=1}^{H} \mathbb{E}_{Q,Q' \sim \mu} \left[ \left( \mathbb{E}^{M, \pi_Q \circ_h \pi_Q^{\mathrm{est}}} \left[ \ell_h^{\mathrm{est}}(Q'; z_h) \right] \right)^2 \right].$$

In the on-policy case in which $\pi_Q^{\text{est}} = \pi_Q$, it suffices to set $\alpha = 0$, which gives

$$\sum_{h=1}^{H} \mathbb{E}_{Q,Q'\sim\mu}\left[\left(\mathbb{E}^{M,\pi_Q \circ_h \pi_Q^{\text{est}}}\left[\ell_h^{\text{est}}(Q';z_h)\right]\right)^2\right]$$

$$= \sum_{h=1}^{H} \mathbb{E}_{Q,Q'\sim\mu}\left[\left(\mathbb{E}^{M,\pi_Q}\left[\ell_h^{\text{est}}(Q';z_h)\right]\right)^2\right]$$

$$= \sum_{h=1}^{H} \mathbb{E}_{\pi\sim p}\,\mathbb{E}_{Q'\sim\mu}\left[\left(\mathbb{E}^{M,\pi}\left[\ell_h^{\text{est}}(Q';z_h)\right]\right)^2\right]$$

$$= \mathbb{E}_{\pi\sim p}\,\mathbb{E}_{Q\sim\mu}[D_{\mathsf{bi}}^{\pi}(Q \parallel M)],$$

where the last equality relabels $Q' \leftarrow Q$. Setting $\eta = 2\gamma$ yields the result.

In the general case where $\pi_Q^{\text{est}} \neq \pi_Q$, we have that for all $Q' \in \mathcal{Q}$,[9]

$$\mathbb{E}_{\pi\sim p}\left[\left(\mathbb{E}^{M,\pi}\left[\ell_h^{\text{est}}(Q';z_h)\right]\right)^2\right] \geq \frac{\alpha}{H}(1 - \alpha/H)^{H-1}\,\mathbb{E}_{Q\sim\mu}\left[\left(\mathbb{E}^{M,\pi_Q \circ_h \pi_Q^{\text{est}}}\left[\ell_h^{\text{est}}(Q';z_h)\right]\right)^2\right].$$

We have $\frac{\alpha}{H}(1 - \alpha/H)^{H-1} \geq \frac{\alpha}{2H}$ whenever $\alpha \leq 1/2$, which gives

$$\mathbb{E}_{\pi\sim p}\,\mathbb{E}_{Q\sim\mu}\left[f^Q(\pi_Q) - f^M(\pi)\right] \leq \alpha + \frac{H \cdot d_{\mathsf{bi}}(\mathcal{Q};M)}{2\eta} + \frac{\eta H}{\alpha} \cdot \mathbb{E}_{\pi\sim p}\,\mathbb{E}_{Q\sim\mu}[D_{\mathsf{bi}}^{\pi}(Q \parallel M)]$$

$$= \alpha + \frac{H^2 \cdot d_{\mathsf{bi}}(\mathcal{Q};M)}{2\gamma\alpha} + \gamma \cdot \mathbb{E}_{\pi\sim p}\,\mathbb{E}_{Q\sim\mu}[D_{\mathsf{bi}}^{\pi}(Q \parallel M)],$$

where the last line chooses $\eta = \gamma\alpha/H$. To conclude, we set $\alpha = \sqrt{H^2 d_{\mathsf{bi}}(\mathcal{Q};\mathcal{M})/4\gamma}$, which is admissible whenever $\gamma \geq H^2 d_{\mathsf{bi}}(\mathcal{Q};\mathcal{M})$.

$\square$

**Proof of Corollary 2.1.** In both of the cases in the theorem statement (on-policy and off-policy), whenever $\gamma \geq 1$ and $n \leq \sqrt{T}$ (to ensure that $K/n \geq 1$), combining Theorem B.1 and Proposition 2.1 gives

$$\mathbf{Reg_{DM}} \lesssim \mathsf{o\text{-}dec}_\gamma^{\mathsf{bi}}(\mathcal{M}) \cdot T + \gamma HL^2 \log(|\mathcal{Q}|KH/\delta) \cdot \frac{T}{n} + \sqrt{nT \log|\mathcal{Q}|}.$$

We choose $n = \sqrt{T}$, which is admissible whenever $T$ is sufficiently large, and gives

$$\mathbf{Reg_{DM}} \lesssim \mathsf{o\text{-}dec}_\gamma^{\mathsf{bi}}(\mathcal{M}) \cdot T + \gamma HL^2 \log(|\mathcal{Q}|KH/\delta)\sqrt{T} + \sqrt{\log|\mathcal{Q}|} \cdot T^{3/4}.$$

In the on-policy case, we have, from Proposition 2.2, that

$$\mathbf{Reg_{DM}} \lesssim \frac{HdT}{\gamma} + \gamma HL^2 \log(|\mathcal{Q}|KH/\delta)\sqrt{T} + \sqrt{\log|\mathcal{Q}|} \cdot T^{3/4},$$

and in the off-policy case, we have

$$\mathbf{Reg_{DM}} \lesssim \sqrt{\frac{H^2 d}{\gamma}}T + \gamma HL^2 \log(|\mathcal{Q}|KH/\delta)\sqrt{T} + \sqrt{\log|\mathcal{Q}|} \cdot T^{3/4}.$$

Choosing $\gamma$ to balance yields the result. $\square$

---

[9]Note that this result uses that the quantity $\mathbb{E}^{M,\pi}\left[\ell_h^{\text{est}}(Q';z_h)\right]$ only depends on the policy $\pi$ through $a_1, \ldots, a_h$.

### D.2  Proofs from Section 3

**Proof of Proposition 3.1.** We begin with the upper bound. First, note that by Assumption 3.1 and the AM-GM inequality, we have

$$\mathsf{o\text{-}dec}_\gamma^D(\mathcal{M}) = \sup_{\mu \in \Delta(\mathcal{M})} \inf_{p \in \Delta(\Pi)} \sup_{M \in \mathcal{M}} \mathbb{E}_{\pi \sim p} \mathbb{E}_{\overline{M} \sim \mu}\big[f^{\overline{M}}(\pi_{\overline{M}}) - f^M(\pi) - \gamma \cdot D^\pi(\overline{M} \parallel M)\big]$$

$$\leq \sup_{\mu \in \Delta(\mathcal{M})} \inf_{p \in \Delta(\Pi)} \sup_{M \in \mathcal{M}} \mathbb{E}_{\pi \sim p} \mathbb{E}_{\overline{M} \sim \mu}\Big[f^{\overline{M}}(\pi_{\overline{M}}) - f^{\overline{M}}(\pi) - \frac{\gamma}{2} \cdot D^\pi(\overline{M} \parallel M)\Big] + \frac{L_{\mathrm{lip}}^2}{2\gamma}.$$

$$(33)$$

Consider any fixed choice for $\mu \in \Delta(\mathcal{M})$. By Sion's minimax theorem (see Foster et al. [12] for details), we have

$$\inf_{p \in \Delta(\Pi)} \sup_{M \in \mathcal{M}} \mathbb{E}_{\pi \sim p} \mathbb{E}_{\overline{M} \sim \mu}\Big[f^{\overline{M}}(\pi_{\overline{M}}) - f^{\overline{M}}(\pi) - \frac{\gamma}{2} \cdot D^\pi(\overline{M} \parallel M)\Big]$$

$$= \inf_{p \in \Delta(\Pi)} \sup_{\nu \in \Delta(\mathcal{M})} \mathbb{E}_{\pi \sim p} \mathbb{E}_{\overline{M} \sim \mu} \mathbb{E}_{M \sim \nu}\Big[f^{\overline{M}}(\pi_{\overline{M}}) - f^{\overline{M}}(\pi) - \frac{\gamma}{2} \cdot D^\pi(\overline{M} \parallel M)\Big]$$

$$= \sup_{\nu \in \Delta(\mathcal{M})} \inf_{p \in \Delta(\Pi)} \mathbb{E}_{\pi \sim p} \mathbb{E}_{\overline{M} \sim \mu} \mathbb{E}_{M \sim \nu}\Big[f^{\overline{M}}(\pi_{\overline{M}}) - f^{\overline{M}}(\pi) - \frac{\gamma}{2} \cdot D^\pi(\overline{M} \parallel M)\Big],$$

so that the main term in (33) is equal to

$$\sup_{\mu \in \Delta(\mathcal{M})} \sup_{\nu \in \Delta(\mathcal{M})} \inf_{p \in \Delta(\Pi)} \mathbb{E}_{\pi \sim p} \mathbb{E}_{\overline{M} \sim \mu} \mathbb{E}_{M \sim \nu}\Big[f^{\overline{M}}(\pi_{\overline{M}}) - f^{\overline{M}}(\pi) - \frac{\gamma}{2} \cdot D^\pi(\overline{M} \parallel M)\Big]$$

$$\leq \sup_{\nu \in \Delta(\mathcal{M})} \inf_{p \in \Delta(\Pi)} \sup_{\mu \in \Delta(\mathcal{M})} \mathbb{E}_{\pi \sim p} \mathbb{E}_{\overline{M} \sim \mu} \mathbb{E}_{M \sim \nu}\Big[f^{\overline{M}}(\pi_{\overline{M}}) - f^{\overline{M}}(\pi) - \frac{\gamma}{2} \cdot D^\pi(\overline{M} \parallel M)\Big]$$

$$= \sup_{\nu \in \Delta(\mathcal{M})} \inf_{p \in \Delta(\Pi)} \sup_{\overline{M} \in \mathcal{M}} \mathbb{E}_{\pi \sim p} \mathbb{E}_{M \sim \nu}\Big[f^{\overline{M}}(\pi_{\overline{M}}) - f^{\overline{M}}(\pi) - \frac{\gamma}{2} \cdot D^\pi(\overline{M} \parallel M)\Big].$$

Relabeling, this is equal to

$$\sup_{\mu \in \Delta(\mathcal{M})} \inf_{p \in \Delta(\Pi)} \sup_{M \in \mathcal{M}} \mathbb{E}_{\pi \sim p} \mathbb{E}_{\overline{M} \sim \mu}\Big[f^M(\pi_M) - f^M(\pi) - \frac{\gamma}{2} \cdot D^\pi(M \parallel \overline{M})\Big] = \sup_{\mu \in \Delta(\mathcal{M})} \underline{\mathsf{dec}}_{\gamma/2}^{\check{D}}(\mathcal{M}, \mu).$$

We now prove the lower bound. Using Assumption 3.1 and the AM-GM inequality once more, we have

$$\mathsf{o\text{-}dec}_\gamma^D(\mathcal{M}) = \sup_{\mu \in \Delta(\mathcal{M})} \inf_{p \in \Delta(\Pi)} \sup_{M \in \mathcal{M}} \mathbb{E}_{\pi \sim p} \mathbb{E}_{\overline{M} \sim \mu}\big[f^{\overline{M}}(\pi_{\overline{M}}) - f^M(\pi) - \gamma \cdot D^\pi(\overline{M} \parallel M)\big]$$

$$\geq \sup_{\mu \in \Delta(\mathcal{M})} \inf_{p \in \Delta(\Pi)} \sup_{M \in \mathcal{M}} \mathbb{E}_{\pi \sim p} \mathbb{E}_{\overline{M} \sim \mu}\Big[f^{\overline{M}}(\pi_{\overline{M}}) - f^{\overline{M}}(\pi) - \frac{3\gamma}{2} \cdot D^\pi(\overline{M} \parallel M)\Big] - \frac{L_{\mathrm{lip}}^2}{2\gamma}.$$

$$\geq \sup_{\mu \in \Delta(\mathcal{M})} \sup_{\nu \in \Delta(\mathcal{M})} \inf_{p \in \Delta(\Pi)} \mathbb{E}_{\pi \sim p} \mathbb{E}_{\overline{M} \sim \mu} \mathbb{E}_{M \sim \nu}\Big[f^{\overline{M}}(\pi_{\overline{M}}) - f^{\overline{M}}(\pi) - \frac{3\gamma}{2} \cdot D^\pi(\overline{M} \parallel M)\Big] - \frac{L_{\mathrm{lip}}^2}{2\gamma}.$$

$$(34)$$

Using the minimax theorem in the same fashion as before, the main term in (34) is equal to

$$\sup_{\nu \in \Delta(\mathcal{M})} \sup_{\mu \in \Delta(\mathcal{M})} \inf_{p \in \Delta(\Pi)} \mathbb{E}_{\pi \sim p} \mathbb{E}_{\overline{M} \sim \mu} \mathbb{E}_{M \sim \nu}\Big[f^{\overline{M}}(\pi_{\overline{M}}) - f^{\overline{M}}(\pi) - \frac{3\gamma}{2} \cdot D^\pi(\overline{M} \parallel M)\Big]$$

$$= \sup_{\nu \in \Delta(\mathcal{M})} \inf_{p \in \Delta(\Pi)} \sup_{\mu \in \Delta(\mathcal{M})} \mathbb{E}_{\pi \sim p} \mathbb{E}_{\overline{M} \sim \mu} \mathbb{E}_{M \sim \nu}\Big[f^{\overline{M}}(\pi_{\overline{M}}) - f^{\overline{M}}(\pi) - \frac{3\gamma}{2} \cdot D^\pi(\overline{M} \parallel M)\Big]$$

$$= \sup_{\nu \in \Delta(\mathcal{M})} \inf_{p \in \Delta(\Pi)} \sup_{\overline{M} \in \mathcal{M}} \mathbb{E}_{\pi \sim p} \mathbb{E}_{M \sim \nu}\Big[f^{\overline{M}}(\pi_{\overline{M}}) - f^{\overline{M}}(\pi) - \frac{3\gamma}{2} \cdot D^\pi(\overline{M} \parallel M)\Big].$$

Relabeling, this is equal to

$$\sup_{\mu \in \Delta(\mathcal{M})} \inf_{p \in \Delta(\Pi)} \sup_{M \in \mathcal{M}} \mathbb{E}_{\pi \sim p} \mathbb{E}_{\overline{M} \sim \mu}\Big[f^M(\pi_M) - f^M(\pi) - \frac{3\gamma}{2} \cdot D^\pi(M \parallel \overline{M})\Big] = \sup_{\mu \in \Delta(\mathcal{M})} \underline{\mathsf{dec}}_{3\gamma/2}^{\check{D}}(\mathcal{M}, \mu).$$

$\square$

**Proof of Proposition 3.2.** Let $\overline{M}$ be arbitrary. By Sion's minimax theorem (see Foster et al. [12] for details), we have

$$\mathsf{dec}_\gamma^D(\mathcal{M}, \overline{M}) = \sup_{\mu \in \Delta(\mathcal{M})} \inf_{p \in \Delta(\Pi)} \mathbb{E}_{\pi \sim p, M \sim \mu}\big[f^M(\pi_M) - f^M(\pi) - \gamma \cdot D^\pi\big(\overline{M} \parallel M\big)\big].$$

By the assumed triangle inequality for $D$, we have that for all $\pi \in \Pi$,

$$\mathbb{E}_{M,M' \sim \mu}[D^\pi(M \parallel M')] \le C\,\mathbb{E}_{M \sim \mu}\big[D^\pi\big(\overline{M} \parallel M\big)\big] + C\,\mathbb{E}_{M' \sim \mu}\big[D^\pi\big(\overline{M} \parallel M'\big)\big]$$
$$= 2C\,\mathbb{E}_{M \sim \mu}\big[D^\pi\big(\overline{M} \parallel M\big)\big].$$

Applying this bound above, we have that

$$\mathsf{dec}_\gamma^D(\mathcal{M}, \overline{M}) \le \sup_{\mu \in \Delta(\mathcal{M})} \inf_{p \in \Delta(\Pi)} \mathbb{E}_{\pi \sim p, M \sim \mu}\Big[f^M(\pi_M) - f^M(\pi) - \frac{\gamma}{2C} \cdot \mathbb{E}_{M' \sim \mu}[D^\pi(M' \parallel M)]\Big]$$

$$\le \sup_{\nu \in \Delta(\mathcal{M})} \sup_{\mu \in \Delta(\mathcal{M})} \inf_{p \in \Delta(\Pi)} \mathbb{E}_{\pi \sim p, M \sim \mu}\Big[f^M(\pi_M) - f^M(\pi) - \frac{\gamma}{2C} \cdot \mathbb{E}_{M' \sim \nu}[D^\pi(M' \parallel M)]\Big]$$

$$\le \sup_{\nu \in \Delta(\mathcal{M})} \inf_{p \in \Delta(\Pi)} \sup_{M \in \mathcal{M}} \mathbb{E}_{\pi \sim p}\Big[f^M(\pi_M) - f^M(\pi) - \frac{\gamma}{2C} \cdot \mathbb{E}_{M' \sim \nu}[D^\pi(M' \parallel M)]\Big]$$

$$= \sup_{\nu \in \Delta(\mathcal{M})} \underline{\mathsf{dec}}_{\gamma/(2C)}^D(\mathcal{M}, \nu).$$

$\square$

**Proof of Proposition 3.3.** The upper bound on $\mathsf{o\text{-}dec}_\gamma^{\mathsf{bi}}(\mathcal{M})$ follows from Proposition 2.2, so it remains to prove the lower bound on $\mathsf{dec}_\gamma^{\mathsf{bi}}(\mathcal{M})$.

For a model $M$, define the Bellman operator $\mathcal{T}_h^M$ via

$$[\mathcal{T}_h^M Q](s,a) = \mathbb{E}_{s_{h+1} \sim P_h^M(\cdot|s,a),\, r_h \sim R_h^M(\cdot|s,a)}\Big[r_h + \max_{a' \in \mathcal{A}} Q(s_{h+1}, a')\Big]. \tag{35}$$

Let

$$D_{\mathsf{sbe}}^\pi(Q \parallel M) := \sum_{h=1}^H \mathbb{E}^{M,\pi}\Big[(Q_h(s_h, a_h) - [\mathcal{T}_h^M Q_{h+1}](s_h, a_h))^2\Big]$$

and $D_{\mathsf{sbe}}^\pi\big(\overline{M} \parallel M\big) := D_{\mathsf{sbe}}^\pi(Q^{\overline{M},\star} \parallel M)$. By Jensen's inequality, it suffices to lower bound $\mathsf{dec}_\gamma^{D_{\mathsf{sbe}}}(\mathcal{M})$.

Let $\mathcal{S} = \{\mathfrak{s}, \mathfrak{t}\}$ and $\mathcal{A} = \{\mathfrak{a}, \mathfrak{b}\}$. We consider a sub-family $\mathcal{M}' \subset \mathcal{M}$ of deterministic combination lock MDPs parameterized by $\vec{a} = (\vec{a}_1, \ldots, \vec{a}_H) \in \mathcal{A}^H$, with $M_{\vec{a}}$ defined as follows.

- The initial state is $s_1 = \mathfrak{s}$.

- For each $h = 1, \ldots, H-1$, if $s_h = \mathfrak{s}$, then selecting $a_h = \vec{a}_h$ transitions to $s_{h+1} = \mathfrak{s}$, and selecting $a_h \ne \vec{a}_h$ transitions to $s_{h+1} = \mathfrak{t}$. $\mathfrak{t}$ is a self-looping terminal state: if $s_h = \mathfrak{t}$, then $s_{h+1} = \mathfrak{t}$ regardless of the action taken.

- If $s_H = \mathfrak{s}$ and $a_H = \vec{a}_H$, then $r_H = \Delta > 0$; all other state-action tuples have zero reward.

We choose $\overline{M}$ such that $Q_h^{\overline{M},\star}(s,a) = 0$ for all $(h, s, a)$. We calculate that for all $\vec{a} \in \mathcal{A}^H$, and all policies $\pi \in \Pi_{\mathrm{RNS}}$,

- $f^{M_{\vec{a}}}\big(\pi_{M_{\vec{a}}}\big) - f^{M_{\vec{a}}}(\pi) = \Delta \cdot \mathbb{P}^{M_{\vec{a}},\pi}(a_{1:H} \ne \vec{a})$.

- $D_{\mathsf{sbe}}^\pi\big(Q^{\overline{M},\star} \parallel M_{\vec{a}}\big) = \Delta^2 \cdot \mathbb{P}^{M_{\vec{a}},\pi}(a_{1:H} = \vec{a})$.

It follows that

$$\mathsf{dec}_\gamma^{D_{\mathsf{sbe}}}(\mathcal{M}, \overline{M}) \ge \inf_{p \in \Delta(\Pi)} \max_{\vec{a} \in \mathcal{A}^H} \mathbb{E}_{\pi \sim p}\big[\Delta \cdot \mathbb{P}^{M_{\vec{a}},\pi}(a_{1:H} \ne \vec{a}) - \gamma \cdot \Delta^2 \cdot \mathbb{P}^{M_{\vec{a}},\pi}(a_{1:H} = \vec{a})\big]$$

$$\ge \inf_{p \in \Delta(\Pi)} \mathbb{E}_{\vec{a} \sim \mathrm{Unif}(\mathcal{A}^H)} \mathbb{E}_{\pi \sim p}\big[\Delta \cdot \mathbb{P}^{M_{\vec{a}},\pi}(a_{1:H} \ne \vec{a}) - \gamma \cdot \Delta^2 \cdot \mathbb{P}^{M_{\vec{a}},\pi}(a_{1:H} = \vec{a})\big].$$

It is straightforward to see by induction that for all $\pi \in \Pi_{\mathrm{RNS}}$, $\mathbb{E}_{\vec{a} \sim \mathrm{Unif}(\mathcal{A}^H)}[\mathbb{P}^{M_{\vec{a}}, \pi}(a_{1:H} = \vec{a})] \leq 2^{-H}$ and $\mathbb{E}_{\vec{a} \sim \mathrm{Unif}(\mathcal{A}^H)}[\mathbb{P}^{M_{\vec{a}}, \pi}(a_{1:H} \neq \vec{a})] \geq \frac{1}{2}$, so we have

$$\mathsf{dec}_\gamma^{D_{\mathrm{sbe}}}(\mathcal{M}, \overline{M}) \geq \frac{\Delta}{2} - \gamma \frac{\Delta^2}{2^H}.$$

The result follows by choosing $\Delta$ appropriately.

$\square$

**Proof of Proposition 3.4.** Let $H$ be given, and assume without loss of generality that $S = 2^{H-1} + 2^{H-2}$. Let $N := 2^{H-2}$. We construct a family of MDPs $\mathcal{M} = \{M_1, \ldots, M_N\}$ with deterministic rewards and transitions as follows.

- $\mathcal{A} = \{\mathfrak{a}, \mathfrak{b}\}$.
- The state space is $\mathcal{S} = \mathcal{T} \cup \mathcal{E} \cup \{\mathfrak{s}\}$. $\mathfrak{s}$ is a deterministic initial state which appears only in layer 1. $\mathcal{T}$ represents a depth $H - 2$ binary tree (with the root at layer $h = 2$), which has $N = 2^{H-2}$ leaf states in layer $H$, labeled by $\{1, \ldots, N\}$, and $2^{H-1} - 1$ states in total. $\mathcal{E} = \{N + 1, \ldots, 2N\}$ is an auxiliary collection of self-looping terminal states, each of which can appears in layers $h = 2, \ldots, H$. In total, we have $|\mathcal{S}| = 2^{H-1} + 2^{H-2}$.

The dynamics and rewards for MDP $M_i$ are as follows.

- At layer $h = 1$, $s_1 = \mathfrak{s}$ deterministically.
- If $a_1 = \mathfrak{a}$, we transition to $s_2 = N + i \in \mathcal{E}$, and if $a_1 = \mathfrak{b}$, we transition to the root state for the tree $\mathcal{T}$. We receive no reward.
- For $h \geq 2$, all states $s \in \mathcal{E}$ are self-looping and have no reward (i.e. $s_{h+1} = s_h$ if $s_h \in \mathcal{E}$ for $h > 1$).
- For $h \geq 2$, states in $\mathcal{T}$ follow a standard deterministic binary tree structure (e.g., [24, 8]). Beginning from the root node at $h = 2$, action $\mathfrak{a}$ transitions to the left successor, while action $\mathfrak{b}$ transitions to the right successor. For $M_i$, we receive reward 1 for reaching the leaf node $i \in \mathcal{T}$ at layer $H$, and receive zero reward for all other states. Note that the transition probabilities for this portion of the MDP do not depend on $i$.

**Online estimation.** We first construct an online estimation algorithm for the class $\mathcal{M}$. Recall that we adopt Hellinger distance, given by $D_{\mathsf{H}}^\pi(\widehat{M}, M) = D_{\mathsf{H}}^2(\widehat{M}(\pi), M(\pi))$.

We first note that since all $M \in \mathcal{M}$ have $f^M(\pi_M) = 1$, the optimistic estimation error is equal to the (non-optimistic) estimation error

$$\mathbf{Est}^{\mathsf{H}} = \sum_{t=1}^T \mathbb{E}_{\pi^t \sim p^t} \mathbb{E}_{\widehat{M}^t \sim \mu^t}\left[ D_{\mathsf{H}}^2\left(\widehat{M}^t(\pi^t), M^\star(\pi^t)\right)\right].$$

Observe that since all MDPs $M \in \mathcal{M}$ have deterministic rewards and transitions, there exists a function $o^M(\pi)$ such that $o \sim M(\pi)$ has $o = o^M(\pi)$ almost surely. It follows that

$$D_{\mathsf{H}}^2(M(\pi), M'(\pi)) = 2\mathbb{I}\{o^M(\pi) \neq o^{M'}(\pi)\}$$

for all $M, M' \in \mathcal{M}$, so we have

$$\mathbf{Est}^{\mathsf{H}} = \sum_{t=1}^T \mathbb{E}_{\pi^t \sim p^t} \mathbb{E}_{\widehat{M}^t \sim \mu^t}\left[ D_{\mathsf{H}}^2\left(\widehat{M}^t(\pi^t), M^\star(\pi^t)\right)\right] = 2\sum_{t=1}^T \mathbb{E}_{\pi^t \sim p^t} \mathbb{E}_{\widehat{M}^t \sim \mu^t}\left[\mathbb{I}\{o^{\widehat{M}^t}(\pi^t) \neq o^{M^\star}(\pi^t)\}\right].$$

For the estimation algorithm, we choose

$$\mu^t(M) \propto \exp\left(-\sum_{i<t}\mathbb{I}\{o^M(\pi^i) \neq o^i\}\right).$$

Lemma C.3 implies that with probability 1, the sequence $\pi^1, \ldots, \pi^T$ satisfies

$$\sum_{t=1}^T \mathbb{E}_{\widehat{M}^t \sim \mu^t}\left[\mathbb{I}\{o^{\widehat{M}^t}(\pi^t) \neq o^t\}\right] - \sum_{t=1}^T \mathbb{I}\{o^{M^\star}(\pi^t) \neq o^t\} \leq \frac{1}{2}\sum_{t=1}^T \mathbb{E}_{\widehat{M}^t \sim \mu^t}\left[\mathbb{I}\{o^{\widehat{M}^t}(\pi^t) \neq o^t\}\right] + \log|\mathcal{M}|.$$

Since $o^{M^\star}(\pi^t) = o^t$, rearranging yields

$$\sum_{t=1}^{T} \mathbb{E}_{\widehat{M}^t \sim \mu^t}\left[\mathbb{I}\{o^{\widehat{M}^t}(\pi^t) \neq o^t\}\right] \leq 2\log|\mathcal{M}|.$$

From here, a standard application of Freedman's inequality [16] implies that with probability at least $1-\delta$, $\sum_{t=1}^{T} \mathbb{E}_{\pi^t \sim p^t} \mathbb{E}_{\widehat{M}^t \sim \mu^t}\left[\mathbb{I}\{o^{\widehat{M}^t}(\pi^t) \neq o^t\}\right] \lesssim \log(|\mathcal{M}|/\delta) \lesssim \log(S/\delta)$.

**Lower bound for posterior sampling.** Observe that for all $M_i \in \mathcal{M}$, the unique optimal policy has $\pi_{M_i}(\mathfrak{s}) = \mathfrak{b}$ at $h = 1$. Thus, the posterior sampling algorithm, which chooses $p^t(\pi) = \mu^t(\{M \in \mathcal{M} \mid \pi_M = \pi\})$, will never play $a_1 = \mathfrak{a}$, and will never encounter states in $\mathcal{E}$. As a result, the problem is equivalent (for this algorithm) to a multi-armed bandit problem with $N$ arms and noiseless binary rewards, which requires $\mathbb{E}[\mathbf{Reg}_{\mathsf{DM}}] \gtrsim N \gtrsim S$ in the worst-case [21].

**Upper bound for Algorithm 2.** We first bound the Optimistic DEC for $\mathcal{M}$. For any $\mu \in \Delta(\mathcal{M})$, we can write

$$\mathsf{o\text{-}dec}_\gamma^{\mathsf{H}}(\mathcal{M}, \mu) = \inf_{p \in \Delta(\Pi)} \sup_{M \in \mathcal{M}} \mathbb{E}_{\pi \sim p} \mathbb{E}_{\overline{M} \sim \mu}\left[f^{\overline{M}}(\pi_{\overline{M}}) - f^M(\pi) - \gamma \cdot D\big(\overline{M}(\pi) \parallel M(\pi)\big)\right]$$

$$= \inf_{p \in \Delta(\Pi)} \sup_{M \in \mathcal{M}} \mathbb{E}_{\pi \sim p} \mathbb{E}_{\overline{M} \sim \mu}\left[f^{\overline{M}}(\pi_{\overline{M}}) - f^M(\pi) - 2\gamma \cdot \mathbb{I}\{o^{\overline{M}}(\pi) \neq o^M(\pi)\}\right]$$

$$= \inf_{p \in \Delta(\Pi)} \sup_{M \in \mathcal{M}} \mathbb{E}_{\pi \sim p} \mathbb{E}_{\overline{M} \sim \mu}\left[f^M(\pi_M) - f^M(\pi) - 2\gamma \cdot \mathbb{I}\{o^{\overline{M}}(\pi) \neq o^M(\pi)\}\right],$$

where the last equality uses that $f^M(\pi_M) = 1$ for all $M \in \mathcal{M}$. We choose $p = (1 - \varepsilon)\mu(\{M \in \mathcal{M} \mid \pi_M = \cdot\}) + \varepsilon \pi_{\mathfrak{a}}$, where $\pi_{\mathfrak{a}}$ is the policy that plays action $\mathfrak{a}$ deterministically.

Now, let $M \in \mathcal{M}$ be fixed. Since each model $M_i$ transitions to $s_2 = N + i$ deterministically when $a_1 = \mathfrak{a}$, we

$$\mathbb{E}_{\pi \sim p} \mathbb{I}\{o^{\overline{M}}(\pi) \neq o^M(\pi)\} \geq \varepsilon \cdot \mathbb{I}\{o^{\overline{M}}(\pi_{\mathfrak{a}}) \neq o^M(\pi_{\mathfrak{a}})\} = \varepsilon \cdot \mathbb{I}\{M \neq \overline{M}\}$$

and

$$\mathbb{E}_{\overline{M} \sim \mu} \mathbb{E}_{\pi \sim p} \mathbb{I}\{o^{\overline{M}}(\pi) \neq o^M(\pi)\} \geq \varepsilon \mu(\mathcal{M} \setminus \{M\}).$$

Similarly, we have

$$\mathbb{E}_{\pi \sim p}[f^M(\pi_M) - f^M(\pi)] = \varepsilon + (1 - \varepsilon)\mu(\mathcal{M} \setminus \{M\}).$$

By choosing $\varepsilon = \gamma^{-1}$, which is admissible whenever $\gamma \geq 1$, we have

$$\mathbb{E}_{\pi \sim p} \mathbb{E}_{\overline{M} \sim \mu}\left[f^M(\pi_M) - f^M(\pi) - 2\gamma \cdot \mathbb{I}\{o^{\overline{M}}(\pi) \neq o^M(\pi)\}\right] \leq \varepsilon + \mu(\mathcal{M} \setminus \{M\}) - 2\gamma\varepsilon\mu(\mathcal{M} \setminus \{M\}) \leq \frac{1}{\gamma}.$$

This establishes that

$$\mathsf{o\text{-}dec}_\gamma^{\mathsf{H}}(\mathcal{M}) \leq \frac{1}{\gamma}$$

for all $\gamma \geq 1$. A regret bound of the form $\mathbb{E}[\mathbf{Reg}_{\mathsf{DM}}] \leq \widetilde{O}(\sqrt{T\log(S)})$ now follows by invoking Theorem 2.1 with the estimation guarantee in the prequel and choosing $\gamma$ appropriately.

$\square$

