# OpenReview forum: "Model-Free Reinforcement Learning with the Decision-Estimation Coefficient"
_NeurIPS.cc/2023/Conference — NeurIPS 2023 poster_

### Official Review · Reviewer_DHV3 · 2023-07-05

**Soundness:** 3 good
**Presentation:** 3 good
**Contribution:** 3 good
**Rating:** 6
**Confidence:** 3

**Summary:**

The paper studies model-free RL algorithm using DEC as the complexity measure.

**Strengths:**

Extending previous model-based algorithm to model-free algorithm is a nice contribution.

**Weaknesses:**

I believe this paper provides a nice contribution for the DEC family of papers. But I would like the authors to discuss more on the implication side. Why we care about extending model-free algorithms to bilinear classes? There are too many complexity measures for RL function approximation that are proposed recently so I am concerned about how useful those theories are for practice.

The algorithm for Bellman-Eluder dimension is model-free. Then can I ask why we need a new algorithm?

I hope the authors could pay more attention to the computational side. So far, all the lines of DEC works are not computationally efficient and even lack any experiments.


**Questions:**

See above.

**Limitations:**

1. No discussion or solution on how to implement the algorithm.
2. The analysis seems to build on two existing works: Zhang [32] and Foster et al. [13]. Not very exciting about the technique.

---

> ### Author Rebuttal · Authors · 2023-08-09
>
> Thank you for the review and useful suggestions!
>
> > I believe this paper provides a nice contribution for the DEC family of papers. But I would like the authors to discuss more on the implication side. Why we care about extending model-free algorithms to bilinear classes? There are too many complexity measures for RL function approximation that are proposed recently so I am concerned about how useful those theories are for practice. The algorithm for Bellman-Eluder dimension is model-free. Then can I ask why we need a new algorithm?
>
>
> The goal of this work is not to obtain a new algorithm for certain model classes, but rather to show for the first time that the DEC and estimation-to-decisions framework for proving regret upper bounds leads to meaningful guarantees in the model-free setting. Prior to our work, the DEC/estimation-to-decisions framework (Foster et al. 2021) offered the most general framework for RL with general function approximation, as well as the only complexity measure that leads to both *upper* and *lower* bounds on regret. However, the upper bounds in Foster et al. (2021) are limited to model-based settings. On the other hand, complexity measures such as bilinear classes and bellman-eluder dimension apply to model-free settings, but are less general than the DEC (and only give upper bounds, not lower bounds). Our work gives a unifying perspective, and enjoys the generality of the DEC framework, yet also gives meaningful guarantees for model-free settings.
>
> A secondary advantage of the estimation-to-decisions framework is that it yields a modular algorithm, with a conceptual separation between the estimation and decision-making components. In contrast, the GOLF algorithm (for classes with bounded Bellman-Eluder dimension) is more ad-hoc in nature, employing global optimism to select a value function and its associated greedy policy, which is optimistic and consistent with all prior observations. The estimation and decision-making components of GOLF are thus tightly coupled. Our work shows that these components can be decoupled while still obtaining similar guarantees, which we believe constitutes a significant conceptual contribution.
>
> Let us mention in passing that we consider bilinear classes only as a stylized example to illustrate our techniques. Bilinear classes capture many known generalizations of tabular MDPs, including linear MDPs, linear mixture MDPs, MDPs with low Bellman rank, feature selection in low-rank MDPs, and many more. Note, however, that our techniques are not limited to this setting.
>
> > I hope the authors could pay more attention to the computational side. So far, all the lines of DEC works are not computationally efficient and even lack any experiments.
>
> We emphasize that focusing on statistical complexity as opposed to computational efficiency is a common theme in the line of research on RL with general function approximation, and is not exclusive to the DEC (e.g., existing works on bellman rank, bilinear classes, bellman-eluder dimension, and so on are also inefficient). Understanding how to design computationally efficient algorithms for RL with general function approximation is a fascinating direction for future research, but the first step, which our paper and the papers above works towards, is to understand what is even possible statistically. Nonetheless, we hope that by placing RL theory on solid statistical foundations, our work can serve as a starting point for future work on efficient algorithms.

---

### Official Review · Reviewer_kpYc · 2023-07-09

**Soundness:** 4 excellent
**Presentation:** 4 excellent
**Contribution:** 3 good
**Rating:** 6
**Confidence:** 4

**Summary:**

This paper studies a variant of the recently proposed framework of Foster et al. for sequential decision making, based on a concept called "decision estimation coefficient" (DEC). The variant proposed here is based on enhancing the estimation step in the "estimation-to-decisions" (E2D) with an optimistic bias inspired by the recent work of Zhang. The authors show that this biased estimation scheme, coupled with an appropriately adjusted decision-making rule, can satisfy very similar regret guarantees for a wide range of sequential decision problems, and can provide improvements in certain settings. In particular, the authors show that their approach can handle a large class of tractable models for reinforcement learning called "bilinear classes".

**Strengths:**

The paper is well-written and the technical content is of excellent quality. The proposed approach is justified and explained well, and its analysis is also presented in an accessible way (at least on a high level). While the regret bounds for bilinear MDP classes is not novel in the sense that there exist other algorithms that achieve the same guarantees, I appreciate the conceptual contribution of showing that the DEC framework is also capable of tackling these (relatively) challenging problems.

I appreciated the very careful comparison between all the relevant DEC variants in Section 3: the authors didn't just propose a technique and proved some bounds about it, but also explored a range of other opportunities and explained the differences between them in an accessible manner.

**Weaknesses:**

On the negative side, the rates that the authors derive are not particularly great: the scaling goes from $T^{2/3}$  in the setting with the most stringent assumptions all the way to $T^{5/6}$ as more and more assumptions are dropped. While the authors discuss this limitation quite openly, I would have appreciated some more discussion as to where this relatively poor scaling comes from. One contributing factor is certainly the use of batched estimation steps. My understanding is that some further looseness may come from the optimistic bonuses added to the estimation procedure, which makes the total estimation error grow polynomially with the number of updates (as opposed to logarithmically, which would allow getting sqrt{T} rates after putting everything together). I wonder though if this intuition is correct, and I would appreciate it if the authors could clarify what rate they would get if they could afford to set n=1 (without paying for it). Altogether, it would have been nice if the to compare the various notions of estimation error with the same care as what the DEC variants have received.

Overall, I am leaning towards suggesting acceptance, but I would feel more strongly about my support if the authors were able to address my questions above in a satisfying way.

**Questions:**

See above.

---

> ### Author Rebuttal · Authors · 2023-08-09
>
> Thank you for the positive comments and helpful suggestions!
>
> > While the authors discuss this limitation quite openly, I would have appreciated some more discussion as to where this relatively poor scaling comes from. One contributing factor is certainly the use of batched estimation steps. My understanding is that some further looseness may come from the optimistic bonuses added to the estimation procedure, which makes the total estimation error grow polynomially with the number of updates (as opposed to logarithmically, which would allow getting sqrt{T} rates after putting everything together). I wonder though if this intuition is correct, and I would appreciate it if the authors could clarify what rate they would get if they could afford to set n=1 (without paying for it).
>
> This is a great question. There is indeed room for further work along these lines. The reason for our suboptimal rates differs slightly in the two settings we consider: (1) Corollary 2.1, which makes no Bellman completeness assumption, and (2) Corollary B.1, which assumes Bellman completeness. We discuss both cases separately below.
>
> **Without the completeness assumption (Proposition 2.1 and Corollary 2.1):** For our results that do not make use of completeness, we use a divergence based on average bellman error. Batching is necessary for algorithms based on average bellman error (Prop 2.1 and Corollary 2.1) due to the following well-known technical issue regarding average bellman error: Because the quantity D_{bi} that we are interested in is the *square of an average* as opposed to an *average of a square*, we cannot take advantage of concentration across rounds/iterations, and multiple samples are needed to estimate this quantity well for each iteration (note that the guarantee in Proposition 2.1 is vacuous if there is n=1 sample per batch). This issue is precisely why well-known algorithms based on average bellman error such as OLIVE (Jiang et al. 2017) and BiLinUCB (Du et al. 2021) require multiple samples per iteration, and is one of the main reasons why these algorithms are analyzed in the PAC framework instead of regret.
>
> Without bellman completeness, we obtain T^{3/4} regret in the on-policy case and T^{5/6} for the off-policy case. The reason that these rates are worse than \sqrt{T} is due to the batching issue above (as well as the additional issue of going off policy in the latter case), and is not related to the use of optimistic estimation. If one is interested in PAC guarantees instead of regret, we expect that our analysis can be adapted to provide tighter guarantees.
>
> **With the completeness assumption:** Corollary B.1 gets a rate of T^{2/3}. The algorithm uses a small batch size of n=H, which causes no degradation in rates, and is only needed to ensure that certain conditional independence assumptions required by Theorem C.1 are satisfied; batching is required in Agarwal & Zhang (2022) for the same reason. The improvement in rate that we achieve in the bellman-complete setting (compared to Corollary 2.1) is due to the fact that the batch size does not grow with T, which is facilitated by the two-timescale exponential weights method from Agarwal and Zhang (2022), which takes advantage of the bellman-completeness assumption. In fact, as stated, their result their result would seem to a imply bound on the estimation error in Proposition B.1 that scales as $\log|\mathcal{Q}| + \sqrt{\log|\mathcal{Q}|*T}/\gamma$, which would lead to a \sqrt{T} regret bound for decision making. Unfortunately, there is a gap in their proof (reference [1] in our paper) which is due to the optimistic bonuses added to the estimation procedure. Our self-contained proof fixes this gap, but this leads to a degradation in the rate. We suspect that there exists an estimation algorithm that can obtain the estimation bound claimed above, which would to \sqrt{T} regret, but this likely requires new algorithmic ideas and is out of scope for the current paper.
>
> We mention in passing that compared to other model-free algorithms based on completeness (e.g., GOLF), the reason for the technical difficulties around estimation above is that our results require *online* estimation guarantees rather than *offline* estimation guarantees. This is a more stringent requirement, but is necessary to take advantage of the DEC.
>
>
> > Altogether, it would have been nice if the to compare the various notions of estimation error with the same care as what the DEC variants have received.
>
> Thank you for the useful suggestion! We will add discussion around this point in the final version.

---

> > ### Comment · Reviewer_kpYc · 2023-08-15
> >
> > Thank you for the response! I really appreciate the clarification. Perhaps it would be useful to include (at least a version) of this text in the final version of the paper. I especially wonder about the nature of the gap in the analysis of Agarwal and Zhang --- has this issue been publicly known? If so, an explicit pointer in the present submission would be appreciated. If not, it could make sense to provide a short description of the issue somewhere in the present submission (of course only after discussing with the original authors to make sure that they are fine with it).

---

> > > ### Author Response · Authors · 2023-08-15
> > >
> > > Thank you for the suggestion! We have been in contact with the authors of Agarwal and Zhang (2022) and they are planning to update their paper with a fix under additional structural assumptions. We will be sure to include a pointer once the updated version is available.

---

### Official Review · Reviewer_ooU7 · 2023-07-27

**Soundness:** 4 excellent
**Presentation:** 3 good
**Contribution:** 3 good
**Rating:** 6
**Confidence:** 3

**Summary:**

This paper contributes to a line of research on decision-estimation coefficients (and related algorithms). An optimistic variant of the original DEC is introduced and it is shown that this variant can lead to a new related meta-algorithm with optimism. This new structure and algorithm are shown to provide non-trivial regret bounds for model-free RL for bilinear classes (a known framework for sample efficient learning). It is also shown that the original DEC is insufficient to achieve the same. Finally it is shown that posterior sampling does not solve all the problems that E2D.Opt can solve.

**Strengths:**

Overall this is a good paper that is presented clearly and makes an interesting contribution to a growing line of research. The application of E2D-style algorithms to model-free settings is important, as well as the ability to incorporate classes that were previously not handled.

The discussion of where advantages exist for optimistic DEC are thorough and helpful. The additional results also shed light on the problems that can be handled by both coefficients and other algorithms.

**Weaknesses:**

The new coefficient is used to reproduce an existing result. This leads to a natural new meta-algorithm and theorem, but since they are only shown in one specialized case, it’s unclear whether it’s useful beyond this one setting as a “meta-algorithm.” While it is apparent that one could swap out different divergences and oracles, it may be described better as a single algorithm, inspired by the original E2D.

The regret bound appears to have a slightly worse rate than what would be achievable with the original BiLin-UCB, which I believe has T^{2 / 3} if converted properly to regret.

It’s not clear how one would implement such an algorithm in practice.

**Questions:**

Does optimistic DEC subsume all of the original DEC results? Are there any downsides compared to the original E2D?

What benefit does the batching offer if the regret bounds still scale as poly(K)? Why not just use one sample per batch?

Prop 3.4: shows that PS does not cover all the problems that E2D.Opt can solve. How about the other way?

Prop 3.4: What is the expectation over? Is this an average regret over MDPs in the class and if so, is posterior sampling misspecified?

Suggestions:

(14) does not seem to point to an equation in the main paper.

Lines 94-95: It may be more clear to say that “there exist algorithms/oracles” that do this.


**Limitations:**

No, but this work is theoretical. The limitations are either apparent from the assumptions or discussed adequately. A conclusion to summarize would be helpful though.

---

> ### Author Rebuttal · Authors · 2023-08-09
>
> Thank you for the positive comments and helpful suggestions!
>
> > Does optimistic DEC subsume all of the original DEC results? Are there any downsides compared to the original E2D?
>
> Yes, the optimistic DEC—when equipped with Hellinger distance—recovers all of the main guarantees based on the DEC in Foster et al. (2021). By accommodating more general divergences, we generalize these results. The only possible downside we are currently aware of concerns computation: For some classes (e.g., linear models), there are computationally efficient algorithms for non-optimistic online estimation, but it is unclear whether *optimistic* online estimation can also be performed in a computationally efficient fashion.
>
> > What benefit does the batching offer if the regret bounds still scale as poly(K)? Why not just use one sample per batch?
>
> Batching is necessary for algorithms based on average bellman error (Prop 2.1 and Corollary 2.1) due to the following well-known technical issue regarding average bellman error: Because the quantity D_{bi} that we are interested in is the *square of an average* as opposed to an *average of a square*, we cannot take advantage of concentration across rounds/iterations, and multiple samples are needed to estimate this quantity well for each iteration (note that the guarantee in Proposition 2.1 is vacuous if there is n=1 sample per batch). This issue is precisely why well-known algorithms based on average bellman error such as OLIVE (Jiang et al. 2017) and BiLinUCB (Du et al. 2021) require multiple samples per iteration, and is one of the main reasons why these algorithms are analyzed in the PAC framework instead of regret.
>
> > Prop 3.4: shows that PS does not cover all the problems that E2D.Opt can solve. How about the other way?
>
> E2D.Opt subsumes all results we are aware of based on the frequentist posterior sampling framework used in Zhang (2021) and subsequent work (e.g., Agarwal and Zhang (2022), Zhong et al. (2022)). In addition, it can be shown that the DEC is bounded whenever the information ratio of Russo and Van Roy, which is commonly used to analyze posterior sampling and related algorithms, is bounded; see discussion in Foster et al (2021,2022). Understanding connections between the DEC and more general posterior sampling-like algorithms is an interesting question for future research.
>
> > Prop 3.4: What is the expectation over? Is this an average regret over MDPs in the class and if so, is posterior sampling misspecified?
>
> Note that Proposition 3.4 concerns a frequentist setting, and “Posterior Sampling” in this context refers to the frequentist posterior sampling approach based on optimistic estimation given by Zhang (2022): $\mu^{t}$ is a randomized estimator (distribution over models) produced by an optimistic estimation algorithm, and the frequentist posterior sampling scheme samples a model from this distribution and plays the optimal decision for it. Since we are in a frequentist setting, the expectation only considers the randomness of the algorithm and the randomness of the samples drawn from the model itself.
>
> We mention in passing that while the purpose of this example was to compare to the frequentist posterior sampling approach of Zhang et al. (2022), it is straightforward to show that our lower bound construction also applies to classical posterior sampling in the Bayesian framework with a well-specified prior. We are happy to include this result if it is of interest to the reviewer.
>
> > The new coefficient is used to reproduce an existing result. This leads to a natural new meta-algorithm and theorem, but since they are only shown in one specialized case, it’s unclear whether it’s useful beyond this one setting as a “meta-algorithm.” While it is apparent that one could swap out different divergences and oracles, it may be described better as a single algorithm, inspired by the original E2D.
>
> The goal of this work is not to obtain a new algorithm for certain model classes, but rather to show for the first time that the DEC and estimation-to-decisions framework for proving regret upper bounds leads to meaningful guarantees in the model-free setting. Prior to our work, the DEC/estimation-to-decisions framework (Foster et al. 2021) offered the most general framework for RL with general function approximation, as well as the only complexity measure that leads to both upper and lower bounds on regret. However, the upper bounds in Foster et al. (2021) are limited to model-based settings. On the other hand, complexity measures such as bilinear classes and bellman-eluder dimension apply to model-free settings, but are less general than the DEC (and only give upper bounds, not lower bounds). Our work gives a unifying perspective, and enjoys the generality of the DEC framework, yet also gives meaningful guarantees for model-free settings.
>
> > The regret bound appears to have a slightly worse rate than what would be achievable with the original BiLin-UCB, which I believe has T^{2 / 3} if converted properly to regret.
>
> Please refer to our response to Reviewer kpYc for detailed discussion around this issue.
>
> Thanks again for the suggestions!

---

> > ### Comment · Reviewer_ooU7 · 2023-08-18
> > **Response**
> >
> > Thank you for the helpful response!
> >
> > > it is straightforward to show that our lower bound construction also applies to classical posterior sampling in the Bayesian framework with a well-specified prior. We are happy to include this result if it is of interest to the reviewer.
> >
> > It may be worth mentioning in passing, but I see now it is clear from the original proof.

---

### Official Review · Reviewer_nAEr · 2023-07-31

**Soundness:** 3 good
**Presentation:** 3 good
**Contribution:** 3 good
**Rating:** 6
**Confidence:** 3

**Summary:**

This paper adapted the DEC framework and E2D algorithm to incorporate model-free reinforcement learning. Specifically, this paper proposed the optimistic DEC and proved that it appears in the regret bound for the optimistic E2D algorithm with an optimistic estimation oracle.  The example of bilinear class is provided. The paper discussed when the optimistic E2D algorithm will have benefit than E2D in its original form.

**Strengths:**

This paper is well-written and provides concrete steps toward understanding the general framework of DEC. The original DEC framework is mostly model-based decision making, whereas the new framework can handle model-free RL.


**Weaknesses:**

This paper is largely built upon [13] and [32]. The additional results over [13] and [32] are a bit incremental. While using the DEC framework to provide a regret bound for bilinear class is nice, this does not new/improved regret bound for bilinear class.

In Appendix A, the authors discussed the relationship with constrained DEC [16], and mentioned that it would be interesting to explore whether optimistic estimation can be combined with the constrained DEC techniques. To make this paper stronger, one possible direction is to discuss the relationship with constrained DEC in details (either provide some results in this direction, or point out the technical difficulty to the combination).


**Questions:**

What is the general picture of the lower bound side of optimistic DEC? Am I correct that oDEC is always smaller than DEC, and DEC serves as a lower bound, so it is not meaningful to discuss oDEC as a lower bound?

---

> ### Author Rebuttal · Authors · 2023-08-09
>
> Thank you for your positive comments!
>
> > Am I correct that oDEC is always smaller than DEC, and DEC serves as a lower bound, so it is not meaningful to discuss oDEC as a lower bound?
>
> The relation between the oDEC and the DEC depends on the choice of divergence and the model class. As shown by Corollary 3.1, for Hellinger divergence, the two are equivalent up to constants, which means that our results are always tighter than the guarantees in Foster et al. (2021). As shown by Proposition 3.3, there is a model class for which the oDEC with bilinear divergence is bounded, but the DEC with bilinear divergence can be exponential. In general though, the tightest relationship between oDEC and DEC that we are aware of is Proposition 3.1, which shows that the oDEC with divergence D is never larger than the DEC with the “flipped” version of D.
>
> > What is the general picture of the lower bound side of optimistic DEC?
>
> The DEC defined with Hellinger divergence serves as a lower bound for expected regret. As mentioned before, when defined with respect to Hellinger divergence, the DEC and oDEC are equivalent. Hence, the oDEC with Hellinger divergence also serves as a lower bound on expected regret. Note that the regret upper bound we have obtained for bilinear classes is in terms of the oDEC with squared Bellman error, which in general is not equivalent to the (o)DEC with Hellinger divergence. Proving lower bounds based on variants of the DEC with alternative divergences D such as squared Bellman error is a fascinating question for future research, but is quite subtle, as the choice of D seems to be tied to the model class under consideration (i.e., depending on the choice of D, the oDEC may lead to tight results for some model classes but not others).
>
> Let us mention, however, that the goal of this work is not to obtain matching upper and lower bounds on regret in terms of a variant of the DEC, but rather to show that the estimation-to-decision framework for proving regret upper bounds extends to the model-free setting. The question of proving lower bounds is somewhat orthogonal to our focus, though we hope that our results can inspire future work along this direction.
>
> > This paper is largely built upon [13] and [32]. The additional results over [13] and [32] are a bit incremental. While using the DEC framework to provide a regret bound for bilinear class is nice, this does not new/improved regret bound for bilinear class.
>
> The goal of this work is not to obtain a new algorithm for certain model classes, but rather to show for the first time that the DEC and estimation-to-decisions framework for proving regret upper bounds leads to meaningful guarantees in the model-free setting. Prior to our work, the DEC/estimation-to-decisions framework (Foster et al. 2021) offered the most general framework for RL with general function approximation. However, the upper bounds in Foster et al. (2021) are limited to model-based settings. On the other hand, complexity measures such as bilinear classes and bellman-eluder dimension apply to model-free settings, but are less general than the DEC. Our work gives a unifying perspective, and enjoys the generality of the DEC framework, yet also gives meaningful guarantees for model-free settings.
>
> A secondary advantage of the estimation-to-decisions framework is that it yields a modular algorithm, with a conceptual separation between the estimation and decision-making components. In contrast, the GOLF algorithm (for classes with bounded Bellman-Eluder dimension) is more ad-hoc in nature, employing global optimism to select a value function and its associated greedy policy, which is optimistic and consistent with all prior observations. The estimation and decision-making components of GOLF are thus tightly coupled. Our work shows that these components can be decoupled while still obtaining similar guarantees, which we believe constitutes a significant conceptual contribution.
>
> We consider bilinear classes only as a stylized example to illustrate our techniques. Bilinear classes capture many known generalizations of tabular MDPs, including linear MDPs, linear mixture MDPs, MDPs with low Bellman rank, feature selection in low-rank MDPs, and many more. Note, however, that our techniques are not limited to this setting.

---

> > ### Comment · Reviewer_nAEr · 2023-08-17
> > **Thanks for clarification**
> >
> > Thanks to the authors for clarification. I am wondering if the authors can also respond to the question:
> > > In Appendix A, the authors discussed the relationship with constrained DEC [16], and mentioned that it would be interesting to explore whether optimistic estimation can be combined with the constrained DEC techniques. To make this paper stronger, one possible direction is to discuss the relationship with constrained DEC in details (either provide some results in this direction, or point out the technical difficulty to the combination).
> >
> > I agree that oDEC and constrained DEC are two orthogonal directions. The question is whether they can be naturally combined. If they can simply be combined, maybe these can be added to this paper. If not, some thoughtful discussions in this paper will be helpful.

---

> > > ### Author Response · Authors · 2023-08-20
> > >
> > > Thank you for the interest! Recall that Foster et al. (2023) give variants of the constrained DEC for the PAC setting and the regret setting. For the PAC setting, we are optimistic that our techniques can be combined with those of Foster et al. (2023) to derive guarantees based on an optimistic variant of the constrained DEC (however, as shown in Foster et al. (2023), the constrained and offset DEC for PAC coincide up to lower-order terms, so the value of such an extension is unclear). For the regret setting, the algorithms in Foster et al. (2023) are quite tailored to hellinger distance, and make heavy use of the fact that it satisfies the triangle inequality and symmetry. Adapting their techniques to general divergences, even without the use of optimistic estimation, appears to be quite non-trivial, and is likely out of scope for this paper. Nonetheless, we are happy to include additional discussion around these issues in the final version of the paper.

---

### Decision · Program_Chairs · 2023-09-21

**Decision:**

Accept (poster)

**Comment:**

All the reviewers are supportive of the substance of the paper's contributions. Most of the suggestions for improvement are centered around discussion and contextualization (for example, the possibility of combining optimism with constrained DEC; contextualization around optimality in the regret rates; and the primary benefit of the DEC framework being statistical rather than computational). I highly recommend that the authors include this discussion and contextualization in the camera-ready version.